



# Understanding the Gangotri glacier dynamics: Implications from
# a fully distributed inversion of equivalent water-volume change
Anikul Islam[1*], Divyesh Varade[1*], Aliva Nanda[2], Somil Swarnkar[3], Rajiv Sinha[4]
[1.] *Department of Civil Engineering, Indian Institute of Technology, Jammu, India*
[2.] *School of Civil and Environmental Engineering, Indian Institute of Technology, Mandi,*
*India*
[3.] *Department of Earth and Environmental Sciences, Indian Institute of Science,*
*Education & Research, Bhopal, India*
[4.] *Department of Earth Sciences, Indian Institute of Technology Kanpur, India*
* Corresponding author(s)[1] and emails: 2024rce2034@iitjammu.ac.in;
divyesh.varade@iitjammu.ac.in
**Abstract:**
The Gangotri Glacier is scientifically controversial regarding its dynamics, ice thickness,
volume, and mass balance due to the lack of field data. Evidence of rapidly increasing
temperatures with climate change is clearly visible in the concerning mass changes of
Himalayan glaciers. Subsequently, monitoring glacier volumes is critical for managing
regional water resources and predicting glacier dynamics. The Gangotri glacier, a significant
water resource for northern India, is experiencing significant changes due to climate change.
This study emphasizes its dynamic nature from 2016 to 2023. Ice thickness distribution of
Gangotri glacier estimated using velocity and shear stress-based approach. Sentinel-2 multi-
spectral imagery is used to estimate glacier velocity with three different approaches for
comparative assessment of the ice dynamics based on pixel-wise cross-correlation. A laminar
flow-based approach is applied to determine the thickness of the Gangotri Glacier. The
thickness change of the study period is used to estimate the mass balance and equivalent water
volume change of the glacier. The observed velocity ranged from $31 \pm 5.8$ - $81 \pm 15.12$ ma$^{-1}$ in
the accumulation area to $15 \pm 2.8$ - $28 \pm 5.23$ ma$^{-1}$ near the snout, and the thickness varied from
$580 \pm 74.47$ m in the upper reaches to $70 \pm 9$ - $115 \pm 14.77$ m near the snout. Through this
study, we found that the mass wastage of the glaciated ice during the study period was $-1.01 \pm$
0.403 m w.e. a$^{-1}$ (meter water equivalent)$^,$ and the mean glaciated ice volume was $19.70 \pm 2.64$
km$^3$. We observed the volumetric change is a declining pattern of the study period 2017 to
2023 gradually. The climatic parameters observed an increasing trend over the last two
decades. We also found that the Apparent Thermal Inertia (ATI) increased which determined
the debris accumulation over the ablation zone significantly from the side wall of the glacier
due to fluctuation of the temperature differences (Thaw-freezing). These changes denote a
significant reduction in the water storage capacity of the Gangotri Glacier.



**Keywords:** Apparent Thermal Inertia; Gangotri; Glacier velocity; ice-thickness; mass balance;
glacier ice equivalent water volume; ice thinning rate; laminar flow; ice flux divergence

## 1. Introduction

Water resources are critical for sustaining life, ecosystems, and socio-economic activities
globally. The Himalayas, also referred to as the "Water Tower of Asia," are home to one of the
largest mountain glacier networks on Earth (Bolch et al., 2012a). The HKH has a total of 54,252
glaciers occupying an area of 60,054 km$^2$ and an estimated ice reserve of 6,127 km$^3$
(Bajracharya et al., 2015). However, there is a largely significant variation in the size and shape
of glaciers between different river basins. Notably, the largest glaciated areas are found in the
Indus, Brahmaputra and Ganga Basins (Mukherji et al., 2015). The effects of rapid climate
change have significantly altered glacier dynamics, impacting mass balance, ice flow, and
discharge rates (Laurent et al., 2020). These changes pose substantial challenges to water
security, particularly in regions dependent on glacial meltwater during dry seasons (Rabatel et
al., 2013a). The glaciers below 5700m elevation are particularly sensitive to climate change,
particularly when they are not covered by thick debris and are directly exposed (Bajracharya
et al., 2015).
Owing to climate change, most Himalayan glaciers have been retreating at a rate of 16–35
meters per year over the past century (Bhambri & Bolch, 2009; Prasad et al., 2009). This
ongoing melting and stagnation of mountain glaciers pose a threat to local populations by
diminishing the year-round availability of water for various purposes and increasing the risk of
Glacial Lake Outburst Floods (GLOFs) and other mountainous hazards (Bhambri et al., 2020).
Furthermore, the critical links between meltwater runoff and climatic factors have been
identified (Salim & Pandey, 2021).
Glacier velocity is one of the key indicators of glacier dynamics, which assists in
assessing ice flow, ice flux divergence, mass balance, and glacier behavior under climate
change (Benn et al., 2012; Cogley, 2011). It provides insights into glacier retreat or advance,
often acts as a key factor in predicting Glacial Lake Outburst Flood (GLOF) risks (Bhambri et
al., 2020), and enhances the understanding of the glacier's response to climatic factors
(Scherler et al., 2011). Monitoring the glacier velocity trends also aids in estimating ice
discharge, projecting future glacier evolution, and evaluating potential sea-level rise impacts
(Huss & Hock, 2018; Jacob et al., 2012). Overall, glacier velocity measurements are essential
for understanding glacier stability, health, and their role in regional and global climate systems
(Immerzeel et al., 2010a).
Several studies have demonstrated the utilization of optical imagery for the estimation
of glacier velocity, particularly based on feature tracking, including COSI-Corr, ImGRAFT,
CARST, and AutoRIFT. COSI-Corr is an IDL-based tool for optical feature tracking, initially
used for coseismic deformation (Leprince et al., 2007). ImGRAFT, a MATLAB-based toolbox,
is designed for georectifying and tracking features in both ground-based and satellite imagery
(Messerli & Grinsted, 2015a). CARST, which combines Python and Bash scripts, is used for
monitoring glacier changes, including feature tracking (Willis et al., 2018; Zheng et al., 2019).
AutoRIFT, a Python-based algorithm, focuses on microwave imagery for feature tracking (A.
S. Gardner et al., 2018). This study evaluates various velocity estimation approaches for the



Gangotri Glacier to compare their robustness, employing Glacier Image Velocimetry (Van
Wyk De Vries & Wickert, 2021), COSI-Corr (Leprince et al., 2007), and ImGRAFT
techniques, focusing on this benchmark glacier in the Garhwal Himalaya. The precise
estimation of glacier velocity and at better spatial resolution is critical in the inversion of glacier
ice-thickness.

Multiple methods have been utilized to estimate glacier thickness by integrating mass-
balance modelling with ice dynamics (Van Wyk De Vries et al., 2022). Two popular
approaches based on the basal sliding and velocity based inversion have been widely used to
study the glacier ice-thickness (Sinha et al., 2024). It has been generally observed that some
models overestimate and others underestimate the glacier ice-thickness. Subsequently,
ensemble modelling has gained wider popularity in the inversion of glacier ice thickness. In
practical applications, these modelling techniques effectively determine ice thickness for most
glaciers, though they tend to exhibit greater uncertainties when applied to small glaciers with
gentle topography (Linsbauer et al., 2012a; Rabatel et al., 2018). Moreover, conducting in situ
observations using intrusive or extrusive methods, such as hot water drilling, seismic or radar
measurements, and gravimetry, poses significant challenges on glaciers with rugged terrain and
is often impractical for complete glacier surfaces. Consequently, extrapolation techniques are
employed to estimate ice thickness by analyzing traverse profiles of glacier surfaces (Fischer,
2009). The World Glacier Monitoring Service offers point-based ice thickness measurements
for 2,000 glaciers, facilitating the calibration of model parameters (Welty et al., 2020). The
International Association of Cryospheric Sciences (IACS) has also undertaken an impressive
project called the Ice-thickness Models Intercomparison eXperiment (ITMIX) as (Farinotti et
al., 2017) in which they compared 17 different approaches to estimating ice thickness based on
artificial neural networks, mass conservation, mass balance, ice flow velocity, and basal shear
stress, among other methods. Using an ensemble of five models based on ice flow dynamics,
(Farinotti et al., 2019) estimated the ice thickness distribution of approximately 215,000
glaciers outside Greenland and Antarctica, revealing a total volume of $158 \pm 41 \times 10^3$ km³, with
High Mountain Asia hosting about 27% less glacier ice.
Some studies of Himalayan glacier ice thickness estimation have been conducted based
on optical and microwave remote sensing such as interferometric SAR (InSAR) for glacier
mass change and thickness change rate (Bandyopadhyay et al., 2019; Ramsankaran et al.,
2018a; V. B. Singh et al., 2018). In order to estimate the thickness of glacial ice, (Farinotti et
al., 2009) created a model based on mass conservation of ice that uses digital elevation data,
glacier boundaries, and boundaries of ice flow catchments. Some other models for ice thickness
are based on ice surface slope (Haeberli & Hoelzle, 1995a), shear stress (Linsbauer et al.,
2012a) and glacier flow-based ice thickness estimation (Gantayat et al., 2017). The Gangotri
glacier has also been widely studied using the different methods for ice-thickness inversion
(Gantayat et al., 2014).

This study aims to conduct a comprehensive investigation on understanding the ice-
dynamics of the Gangotri glacier through a comparative model-based analysis. The ice-
thickness is modelled using both velocity and shear stress-based approaches with the purpose
to identify the optimal method that aligns theoretical knowledge with modelled results. We
have also investigated the annual ice-thinning rates compared with the geodetic method and



their implications in retrieving the equivalent water volume variations. Unlike the Accumulation Area Ratio (AAR) and Volume-Area (V-A) scaling method which utilizes an empirical relation that may not often fit for every glacier, we utilize the volumetric change-based approach for estimating the mass balance and further study the correspondence with AAR based results. In summary, the current study aims to:

1) evaluate different ice-velocity determination methods using time-series and multi-date satellite imagery;
2) compare the ice-velocity, basal shear stress-based, and geodetic estimation of ice-thickness;
3) estimate the rate of glacier ice thinning rate based on the thickness change;
4) estimate the equivalent water volume changes of the Gangotri glacier;
5) estimate the mass balance of the Gangotri glacier using ice thickness change of the study period.
6) assess the influence of different climatic parameters on glacier dynamics and surface mass balance.

## 2. Study area and datasets

### 2.1. Study Area

The Gangotri Glacier is typical valley-type glacier of the Garhwal Himalaya in the upper Bhagirathi catchment (Fig. 1). Situated in Uttarakhand, India, the Gangotri Glacier spans latitudes 30°43′22″N to 30°55′49″N and longitudes 79°4′41″E to 79°16′34″E, with an approximate length of 30 km and an area of 137 km² (as per this study). The glacier's surface elevation varies significantly, ranging from around 3970 m to 7000 m. The equilibrium line altitude (ELA) for the Gangotri Glacier and its tributaries is estimated at approximately 5100 m, as determined by (Bhushan et al., 2017) based on field observations from the Dokrani Glacier. The major tributaries of Gangotri Glacier are Kirti, Chaturangi, and Raktvarna. Ganga which is the largest river of India flows from the Gangotri glacier. The melting period of the glacier is May to October (Dobhal et al., 2013). The temperature of the Gangotri glacier region and its surroundings varies between 5 °C to 15 °C with a humidity level of ~ 68 %. Solid and liquid precipitation is between 131 mm and 395 mm (Salim & Pandey, 2021; P. Singh & Singh, 2001).





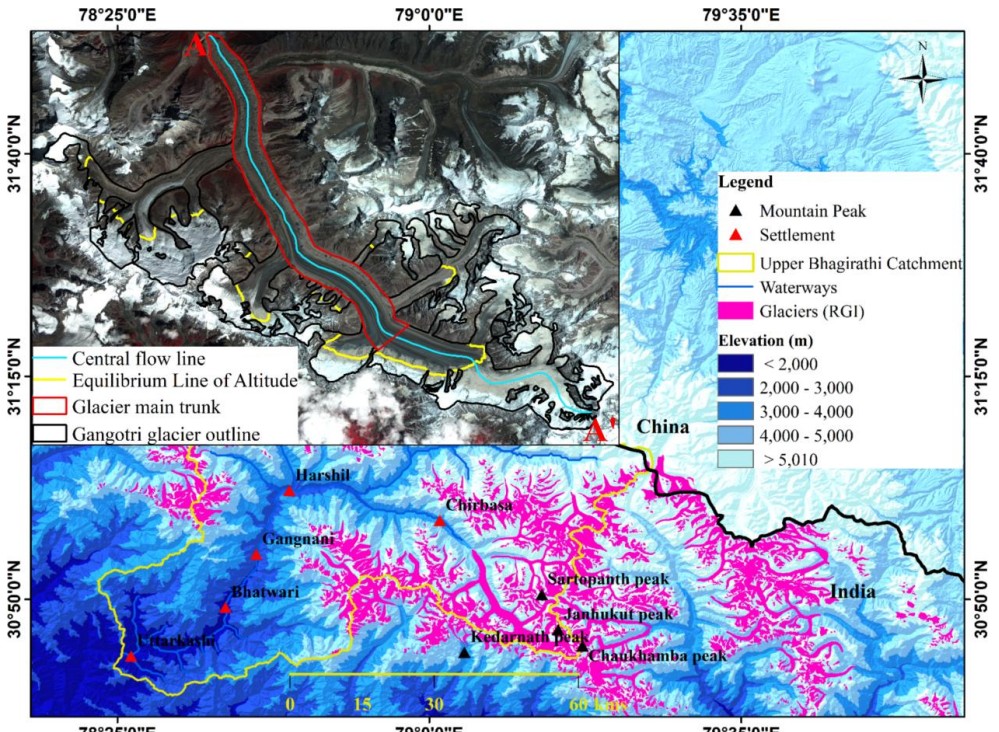

**Figure 1:** Gangotri glacier with glacier boundaries modified from the Randolf Glacier Inventory v6.0, where the snout is shown at A (red bold font), the ELA for the main lobe and the side lobes are shown in yellow, and the central flow line is shown in cyan (A-A').

### *2.2. Datasets*

The datasets used in this study primarily include remote sensing satellite optical imagery, particularly Sentinel-2 RGB and Near-Infrared band imagery (2016 – 2023) with a spatial resolution of 10 meters, which was essential for estimating glacier velocity and capturing temporal trends in glacier motion. For deriving the slope of the glacier surface, stereo pair-based derived digital elevation Cartosat-1 DEM (Elevation / vertical accuracy 8m) (data year 2018) (Talchabhadel et al., 2021), with a resolution of 30 meters, was utilized. The SRTM DEM used in this study (data year 2000), with a spatial resolution of 30 meters, was derived from C-band radar interferometry data and has an accuracy of ±16 meters (Farr et al., 2007). Its vertical accuracy has been reported as less than 9 meters (Rodríguez et al., 2006), and more specifically, 4.31 meters (±14.09 meters) in mountainous regions (Kolecka & Kozak, 2014). The Copernicus DEM (data year 2015) Absolute vertical accuracy < 4m (European Space Agency & Airbus, 2022) (90% linear error), also with a 30-meter resolution, was used for differencing with the SRTM data to analyze elevation changes on the Gangotri Glacier's ice surface over time. The RGI 6.0 (RGI Consortium, 2017) dataset was used for glacier ice masking, providing an accurate framework for delineating the glacier boundaries which is modified using high-resolution satellite imagery (e.g., WorldView, Quickbird, and IKONOS) in Google Earth Pro of the snow-free period by the on-screen digitisation. MODIS Land



Surface Temperature (LST) data (2016 – 2023), at a 500-meter resolution from Aqua and Terra
satellites, was applied to derive ice surface temperature. ITS LIVE velocity data (A. S. Gardner
et al., 2018), with a resolution of 120 meters, was incorporated for correlation with velocity
model outputs, while the widely used ice thickness dataset by Farinotti et al., (2019), with a
50-meter resolution, was used for comparative analysis with thickness model outputs. Table 1.
summarizes the various datasets used in this study.
**Table 1.** Satellite and other ancillary data used in this study.

| Data | Time period | Spatial Resolution | Purpose | Source |
|---|---|---|---|---|
| Sentinel – 2 | 2016 - 2023 | 10 m | Glacier velocity estimation | (European Space Agency, 2018) |
| Cartosat – 1 DEM | 2018 | 30 m | Slope and thickness estimation | (Muralikrishnan et al., 2013) |
| SRTM DEM | 2000 | 30 m | DEM differencing | (OpenTopography, 2013) |
| Copernicus DEM | 2015 | 30 m | DEM differencing | (European Space Agency & Airbus, 2022) |
| RGI 6.0 | 2017 | | Glacier ice masking | (RGI Consortium, 2017) |
| MODIS LST (MOD11A1) (Aqua and Terra dataset) | 2016 - 2023 | 500 m | Derived ice surface temperature | (Yu et al., 2022) |
| ITS_LIVE | 2016 - 2023 | 120 m | Correlation with velocity model output | (A. Gardner et al., 2022) |
| Global Ice Thickness Dataset | 2019 | 50 m | Correlation with thickness model output | (Farinotti et al., 2019) |
| Google Earth Pro | | | Glacier outline modification | - |
| Landsat – 8 | 2022 | 30 m | LST determination for validation of MODIS derived LST | (Earth Resources Observation and Science (EROS) Center, 2013) |
| MODIS (MCD43A3) | 2000 - 2023 | 500 m | Ice Surface Albedo | (Schaaf & Wang, 2021) |
| Terra-climate | 2000 - 2023 | 4.56 km | Max. temperature, Precipitation, Runoff | (Abatzoglou et al., 2018) |
| ERA 5 Land | 2000 - 2023 | 9 km | Snowfall estimation | (C3S, 2018) |



| MCD19A2 | 2000 - 2024 | 500 m | Aerosol Optical Depth (AOD) estimation | (Lyapustin & Wang, 2018) |
| GWRPM25 | 1998 - 2021 | 1.13 km | PM 2.5 trend analysis | (Van Donkelaar et al., 2021) |


## 3. Methodology

The framework for the estimation of the ice thinning rate, ice volume, mass balance and equivalent water volume change of the Gangotri Glacier involves integrating multiple datasets and methods using glacier surface topography, glacier ice movement, glacier shape, glacier elevation change, and basal shear stress of glacier surface as depicted in Fig. 2. The various steps further detailed in the forthcoming subsections in the methodology involve (a) Image processing and ice masking, (b) Ice velocity calculation based on feature tracking, (c) Glacier ice thickness estimation using glacier velocity and shear stress, (d) Glacier elevation change detection by DEM differencing with 15 years' temporal gap, (e) Ice volume estimation with piece-wise aggregation and general mean volume by glacier area and finally (f) Ice thinning rate, mass balance and equivalent water volume change estimation.

195

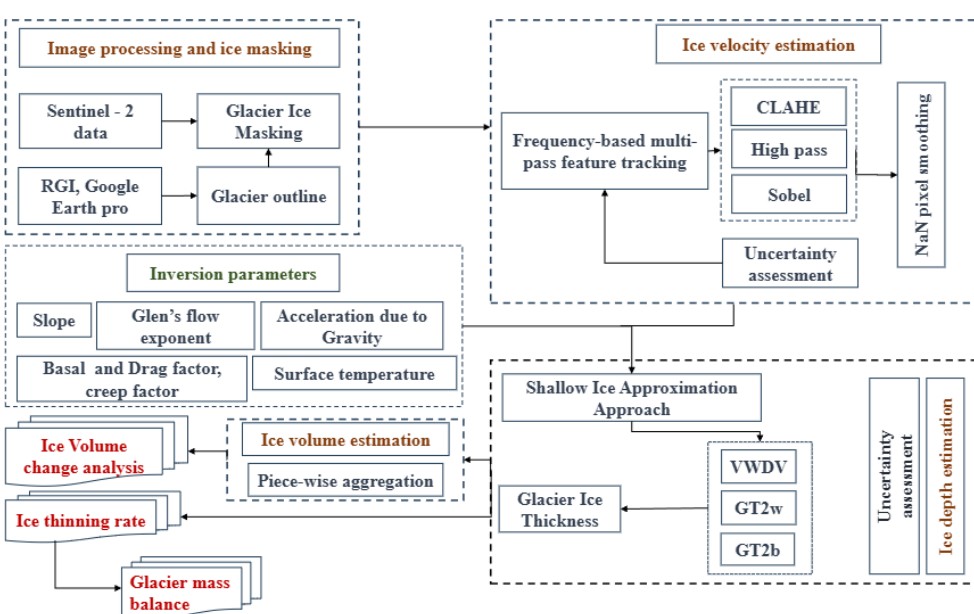

196

**Figure 2.** Workflow for the methodology adopted in this study covering the various steps including primarily the tasks as image processing and ice masking, parametric inversion, estimation of ice surface velocity, modelling of ice thickness and piecewise estimation of glacier ice volume. The individual components are estimated at approximately annual scale to investigate the ice thinning rates and corresponding mass wastage and volume changes.




### 3.1. Image processing and ice masking:

A total of 152 Sentinel-2 RGB images, spanning the years 2016 to 2023, were used for Glacier
Image Velocimetry (GIV)-based velocity estimation. Additionally, for comparison single-band
NIR grayscale images, free of clouds and snow, were utilized for velocity estimation using the
Co-registration of Optically Sensed Images and Correlation (COSI-Corr) technique and Image
georectification and feature tracking toolbox (ImGRAFT) with time-series image pairs from
2016 to 2023. The RGB images were processed in Google Earth Engine (GEE) with a cloud-
masking filter to enhance feature tracking accuracy. Sentinel-2 NIR single-band images,
representing the snow-free months of October and November, were downloaded from the
Sentinel Hub.
The Gangotri Glacier, a valley-type receding glacier located in the Garhwal Himalaya,
exhibits dynamic changes in its outline due to the significant retreat of its snout. To account
for these changes, the glacier outline from the Randolph Glacier Inventory (RGI 6.0) was
updated using high-resolution satellite imagery from Google (Google Earth Pro) and Sentinel-
2 RGB images. All RGB images were processed with a 500-meter buffer around the modified
glacier outline to distinguish ice-covered areas from ice-free terrain, ensuring accurate masking
of the glacier surface. This masking was critical for subsequent GIV-based velocity estimation.
The GIV based velocity map was carefully refined by applying a glacier mask based on the
updated glacier outline, ensuring that the velocity data was confined to the glacier's extent. This
updated outline was derived to accurately represent the dynamic changes in the glacier's
boundaries, allowing for precise spatial alignment and excluding areas outside the glacier.

### 3.2. Glacier ice velocity estimation

Sentinel-2 RGB imageries were used for a pair-wise velocity estimation based on the Glacier
Image Velocimetry (GIV) toolbox using multi pass feature tracking frequency domain image
correlator (Van Wyk De Vries & Wickert, 2021). We utilize image pairs with a minimum
temporal interval of 9 days and a maximum temporal interval of 1 year for the computation of
annual displacement measurements. Total pairs were generated for this glacier with all years
for this study 8-year time period (2016 - 2023).
Displacements between each image pair are determined using a frequency-domain multi-pass
image correlator. A 50% overlap is applied during this iterative refinement, resulting in a final
velocity map resolution of 30 m. The displacement map was generated at final iteration based
on sub-pixel estimator (Van Wyk De Vries & Wickert, 2021) and resulted pair-wise
displacement map converted into final velocity map. Signal to noise ratio lower than 5 and
peak ratio less than 1.3 was considered during multi-pass template matching (Van Wyk De
Vries & Wickert, 2021). For displacement image filtering three filter was utilized contrast
limited histogram equalization size (CLAHE), high pass, and sobel filter. Orientation filter
(Near Anisotropic Orientation Filter) NAOF was used which lead to the best match results
during template matching NAOF (Van Wyk De Vries & Wickert, 2021). To estimate the velocity
of the Gangotri Glacier, a maximum velocity threshold of 200 ma$^{-1}$ was applied. In the final step,
a comprehensive velocity map was created, with velocity values expressed in meters per annum
(ma$^1$).



Sentinel-2 NIR band imagery, primarily acquired during July-October (2016 to 2023)
ensuring minimal cloud cover over the Gangotri Glacier was utilized (Table S1). Using the
freely available COSI-Corr software (Leprince et al., 2007), surface displacement was
estimated for consecutive study intervals (2016–2017, 2017–2018, ..., 2022–2023). The feature
tracking approach was employed within a phase-based correlation framework to achieve
accurate displacement measurements. The specifics of the algorithm can be explored by
(Leprince et al., 2007).
The COSI correlation module provides two correlation algorithms frequency-based and
statistical based respectively. This study used the frequency method, which is better for reliable
results using an optical dataset (Bhushan et al., 2017). The correlation window, commencing
with an initial window dimension of $64 \times 64$, progressively decreases to a final size of $32 \times 32$,
employing a step size of 2 as specified. First, the output is shown at resolution at 160 meters,
containing displacements in the east-west (EWD) and north-south (NSD) direction, as well as
the associated signal-to-noise ratio (SNR). Pixels with low correlation are eliminated by
applying a limit value of SNR < 0.9. Finally, surface displacements were determined by the
Eulerian distance using the two displacement vectors east-west (E - W) and north–south (N -
S). The velocity vector area was verified and synchronized before generating the Velocity map.
The time interval between the two images (about 1 year for each study period) is utilized to
estimate the ice velocity meter per annum ($ma^{-1}$). The velocity ($u$) is estimated from the North-
South (y) component (NSD) and the East-West (x) component (EWD) over the time interval $t$
over which displacements occur as follows.
$$u = \frac{\sqrt{(NSD)^2 + (EWD)^2}}{t} \tag{1}$$


Another method for estimating glacier surface velocity involved using an alternative
feature-tracking algorithm applied to sequential satellite images from 2016 to 2022. Seven
Near-Infrared (NIR) band images acquired in July-October of each year were used as input
(Table S1), providing information for feature tracking. The modified algorithm of ImGRAFT
(Messerli & Grinsted, 2015b) transitioned from a single-pass to a multi-pass tracking
framework, improving displacement detection across multiple image pairs. The algorithm
utilized template matching to detect displacement vectors between corresponding features, and
a regular grid was created with a defined spacing to sample motion within the region of interest.
The displacement vectors were converted into ground units through pixel-to-meter scaling, and
velocity components (x, y) were computed alongside the total surface velocity (u) after
applying a temporal normalization factor.
*3.3. Glacier Ice thickness estimation*
Ice thickness is determined using the glacier surface velocity and the ice surface slope through
the shallow ice approximation (SIA) method (Cuffey & Paterson, 2010; Hutter & Morland, 1984;
Le Meur et al., 2004). The laminar flow-based shallow ice approximation (SIA) method is
commonly used globally for estimating glacier ice thickness. This approach combines the
principles of glacier flow and the assumptions of shallow ice dynamics to provide estimates of
ice thickness across glaciers (Farinotti et al., 2019; Frey et al., 2014; Maussion et al., 2019;



Millan et al., 2022; Nela et al., 2023). In this study, One methods were employed based on
glacier surface velocity (Farinotti et al., 2017; Gantayat et al., 2014; Van Wyk De Vries &
Wickert, 2021), while two methods utilized the basal shear stress approach (Farinotti et al.,
2017, 2019; Haeberli & Hoelzle, 1995b; Kumari et al., 2021; Linsbauer et al., 2012a). These
methods were carefully selected to provide comprehensive estimates of glacier ice thickness
by leveraging different glaciological principles.
### *3.3.1.  Velocity-based ice thickness*
Glaciers primarily move as the ice deforms under the force of gravity. This flow occurs
through three main mechanisms: internal ice deformation, sliding at the base, and deformation
of the subglacial bed (Hambrey & Glasser, 2012). The glacier ice surface velocity $u(H)$ is a
result of two component glacier internal deformation $u_d(H)$ and basal sliding $u_b$ (Cuffey &
Paterson, 2010), as shown in equation 2.
$$u(H) = u_d(H) + u_b \tag{2}$$
where the ice thickness $H$ corresponds to velocities, including both the total velocity and that
due exclusively to internal deformation, assessed at the glacier surface. The deformation
velocity is related to the basal shear stress and the ice thickness as follows (Cuffey & Paterson,
2010).

$$u_d(H) = \frac{2A_c}{n+1}\tau_b^n H \tag{3}$$
where $A_c$ is the Arehenius creep parameter, $\tau_b$ represents basal shear stress, $n$ is the Glen's
flow exponent ($n = 3$). The glacier surface velocity as shown in equation 1 can then be
represented as shown in equation 4 (Glen, 1958).
$$(1 - \beta)u(H) = u(H) = u_b + \frac{2A_c}{n+1}\tau_b^n H \tag{4}$$
where $\beta$ is the basal sliding correction factor (Chandler et al., 2006). The laminar flow law
(King, 1983) accounts for both surface and basal velocities of the glacier. However,
determining basal or sliding velocity through remote sensing is not feasible. For the HMA
region, the basal velocity is assumed to be one-fourth of the surface velocity, as suggested by
(Gantayat et al., 2014) and (Nela et al., 2023). This study utilizes basal shear stress instead of
the full driving stress to model glacier motion, following the shallow-ice approximation.
Arrhenius creep  (Cuffey & Paterson, 2010) constant $A_c$ determinate using temperature based
on the equation 5 (Table S2 – S3 & Fig. S1).
$$A_c = A_c^* \exp\left(\frac{Q_c}{R}\left[\frac{1}{T} - \frac{1}{T^*}\right]\right) \tag{5}$$
where the constants being $A_c^* = 2.4 \cdot 10^{-24}, Q_c = 115$ kJ mol$^{-1}, R \approx 0.0083145$ (the ideal
gas constant), and $T^* = 273$ K  (Cuffey & Paterson, 2010).
Basal shear stress, $\tau_b$ , is expressed in terms of measurable parameters as follows.





$$\tau_b = f\rho_i gH\sin(\alpha) \tag{6}$$


where $f$ is the shape factor (Gantayat et al., 2014; Haeberli & Hoelzle, 1995b), $\rho_i$ is the ice
density (850 kg/m³), $g$ is acceleration due to gravity (9.8 m/s²), $\alpha$ is the ice-surface slope angle
(derived from the DEM) and $H$ is ice-thickness. The final equation derived from substituting
the above (Eq. 2 - 6) equations for velocity-based ice thickness estimation by (Van Wyk De
Vries et al., 2022).
$$H = \left(\frac{n+1}{2(f\rho_i g)^n A_c^* \exp\left(\frac{Q_c}{R}\left[\frac{1}{T} - \frac{1}{T^*}\right]\right)}\right)^{\frac{1}{n+1}} \left(\frac{u(H)(1-\beta)}{\sin(\alpha)^n}\right)^{\frac{1}{n+1}} \tag{7}$$


where the first term of the equation denotes the all constant and parameters and the second term
represents the DEM based estimated slope and GIV derived ice surface velocity.
By this equation we derived the glacier thickness with approach implemented by Van Wyk De
Vries et al., (2022) using the fully distributed two-dimensional ice surface flow speed and
topographic slope field (VWDV model).

### 3.3.2. Stress based ice thickness
Two models, GT2b basal-shear-stress-based basin-divided approach (Linsbauer et al., 2012b)
and GT2w basal-shear-stress-based whole glacier approach (Ramsankaran et al., 2018b) were
created by rewriting equation 6 for the estimate of ice thickness-based basal shear stress which
is the base equation of the GlabTop model (Linsbauer et al., 2012a).

$$H = \frac{\tau_b}{f\rho_t g\sin(\alpha)} \tag{8}$$


where $\tau_b$, which is closely correlated with glacier thickness, is the basal shear stress along the
central flow line. Frey et al., (2014) have provided an empirical relationship between elevation
zone $\Delta z_i$ (km) and basal shear stress $\tau_b$ (kPa) as input of the ice thickness estimation, which is
as follows:

$$\tau_b = \begin{cases} 0.5 + 159.8\Delta z_i - 43.5(\Delta z_i)^2 & \text{if } \Delta z_i \leq 1.6 \\ 150 & \text{if } \Delta z_i > 1.6 \end{cases} \tag{9}$$


### 3.4. Geodetic Approach: DEM co-registration and elevation change estimation
To obtain elevation variations during the study period (2000 - 2015), the SRTM C-band DEM
(version 3), void-filled with a 1 arc second resolution was collected which is freely available
from www.usgs.earthexplorer.com. The SRTM mission, conducted over 11 days in February
2000, used SAR interferometry to map the surface topography of nearly the entire planet,
covering latitudes between 60°N and 56°S (Bhushan et al., 2017). In the past, the SRTM C
Band DEM has been widely utilized to determine the glacial elevation change (Berthier et al.,



2007; Gardelle et al., 2013; Paul et al., 2017; Pieczonka et al., 2013). The Copernicus DEM
2015 was utilized for elevation change estimation with SRTM 2000.
Initially, both DEMs were reprojected into UTM Zone 44N to ensure compatibility
format of the shape, transform, and CRS, with the same spatial resolution for each DEM. In
this study, the (Nuth & Kääb, 2011) co-registration method was employed, which effectively
estimates and corrects for both horizontal and vertical shifts in DEMs. This approach ensures
accurate alignment by identifying systematic offsets between the reference DEM and the DEM
to be aligned. By calculating the necessary transformations, this method adjusts the DEM's
spatial alignment, correcting horizontal displacements and vertical elevation differences
simultaneously (Etzelmüller, 2000). The application of this co-registration technique in this
study eliminates cell misalignment, allowing for precise integration and analysis of DEM
datasets. The expression for elevation difference $dh$ proposed by (Nuth & Kääb, 2011) is as
follows.
$$dh = a \cdot \tan\alpha \cdot \cos(b - \varphi) + dh' \tag{10}$$

where, $dh, \alpha$, and $\varphi$ represent elevation difference at individual pixels, terrain slope, and
aspect, respectively. The terms $a, b, dh'$ denote the magnitude of the horizontal shift, the
direction of the shift vector, and overall elevation bias between two DEMs, respectively. For
penetration depth correction of the SRTM DEM, we apply average C Band penetration values
of $2.3 \pm 0.6$ m for snow/ice as specified by Nuth & Kääb (2011). We also assume that there is
no penetration in the debris-covered sections of the Gangotri Glacier. To distinguish between
debris-covered and glacier ice/snow areas, we used a Landsat image from September 2000,
with the Normalized Difference Snow Index (NDSI) calculated using the green and short-wave
infrared (SWIR) bands and a threshold value of 0.23.
### 3.5. Ice volume and thinning rate estimation
This study utilized two approaches to estimate glacier ice volume. The first method determines
volume by multiplying the glacier area by its ice thickness for each period. The second method
involves aggregating the volumetric contribution of individual pixels, commonly referred to as
the simple area-weighted sum.
$$V_i = H.Ag \tag{11}$$

where, $V_i$ is the glacier ice volume, $H$ is Monte-Carlo-derived mean ice thickness, and Ag
denotes area of the glacier.
$$V_i = \sum_{j=1}^{n_j} \sum_{k=1}^{n_k} \overline{H_{jk}} \Delta x \Delta y \tag{12}$$
where, $\overline{H_{jk}}$ is the Monte-Carlo-derived mean ice thickness at each cell, and $dx$ and $dy$ are the
grid resolution along each axis.
The thinning rate of the Gangotri Glacier was estimated by performing pairwise differencing
of the estimated ice thickness maps for consecutive periods: 2017–2016, 2018–2017, 2019–





2018, 2020–2019, 2021–2020, 2022–2021, and 2023–2022. This approach provided annual
thickness changes for each interval. The mean rate of thickness change per annum was then
determined by averaging these values. Conversely, thickness variation was also derived from
the differential analysis of elevation data obtained from DEMs corresponding to the years 2000
and 2015 for comparison.

### 3.6. Mass balance estimation

We used the ice thinning rate to indirectly estimate the mass balance ($B$) of Gangotri Glacier
following an approach in which thickness change is utilized to derive the change in the ice
volume, which was then converted to the mass balance using a density conversion (Cogley,
2011). The mass balance and the change in the total volume $\Delta V$ was determined by summing
the change in the ice thickness $\Delta h_i$ at an individual pixel $r$ using the equation 13.

$$B = \frac{\Delta V}{S} \cdot \frac{\rho}{\rho_{\text{water}}}; \qquad \Delta V = \sum_{i=1}^{N} r\Delta h_i \qquad (13)$$


where $B$ is the mass balance in m.w.e, $\rho$ is the density of glacier ice, $S$ is the average area of
glacier during the study period. $N$ is the number of pixels covering the glacier ice at its
maximum extent and r is the pixel size.

## 4. Error and uncertainty assessment

### 4.1. Uncertainty of ice velocity

Compared to microwave imagery, the main drawback of optical image-based feature tracking
is that it is limited by cloud boundaries since it lacks the ability to penetrate clouds. Errors in
ice velocity are mostly caused by inadequate contrast in the image and misregistration of image
pixels caused by snow and cloud cover over glacier areas or their vicinity. In this study, to
reduce inaccuracies caused by snow and cloud cover, the image was ensured to have nearly no
cloud cover and minimum snow over the glacierized area.
We have taken into account the stable ground adjacent to the glaciated area that is free of snow
and clouds for the purpose of estimating the uncertainty of ice velocity using the feature
tracking method GIV, COSI-Corr and ImGRAFT. For this measurement, the slope $\leq 25°$ was
considered. For ground stability, the measurement output should be zero.

### 4.2. Uncertainty of Ice Thickness

Statistical propagation of errors was used to calculate the uncertainty in ice-thickness estimates
using the equation 14:

$$\frac{dH}{H} = \sqrt{\left(\frac{1}{4}\frac{dUs}{Us}\right)^2 + \left(\frac{3}{4}\frac{df}{f}\right)^2 + \left(\frac{3}{4}\frac{dA}{A}\right)^2 + \left(\frac{3}{4}\frac{d\rho}{\rho}\right)^2 + \left(\frac{3}{4}\frac{d\sin\alpha}{\sin\alpha}\right)^2 + \left(\frac{3}{4}\frac{d\beta}{\beta}\right)^2} \qquad (14)$$




Velocity uncertainty $Us$ was estimated by stable terrain. The shape factor uncertainty $df$ was
considered ±12 % of the 0.8 shape factor which was taken account the average between ablation
(0.7) and accumulation (0.9) (Linsbauer et al., 2012a). Following Remya et al., (2019), a
density uncertainty $d\rho$ of ±60 kg/m³ and an uncertainty of ± 8.7% sin $\alpha$ were employed in the
current study. The Arrhenius creep constant ($A$) was calculated using varying temperatures
based on a Monte Carlo simulation, with the standard error (SE) of the mean creep constant
accounted for as uncertainty and basal sliding factor varies between 0.1 to 0.4.

### *4.3. Uncertainty of Elevation Change*

Elevation difference values across stable, ice-free terrains with slopes of less than or equal to
30° are taken into consideration in order to assess the relative vertical accuracy (Agarwal et al.,
2017; Pieczonka et al., 2011).The overall uncertainty in elevation change ( $U_{\mathrm{DTM}}$ ) using the
equation 15:
$$U_{\mathrm{DTM}} = \sqrt{(\sigma)^2 + (\Delta p)^2} \tag{15}$$

where, $\sigma$ is the relative vertical accuracy Normalized mean absolute difference (NMAD), $\Delta p$
is the uncertainty in C Band radar penetration correction (±0.6 m). NMAD was taken account
for elevation difference error.

### *4.4. Uncertainty of ice volume*

Uncertainty of ice volume was calculated using an area-weighted sum of the individual
thickness uncertainty (Van Wyk De Vries et al., 2022) of the ice thickness $\overline{\sigma_{jk}}$ at each grid cell
based on the equation 16:
$$\sigma_i = \sum_{j=1}^{n_j} \sum_{k=1}^{n_k} \overline{\sigma_{jk}} \Delta x \Delta y \tag{16}$$


### *4.5. Uncertainty of mass balance*

Uncertainty of mass balance was calculated using equation 17 by substituting the uncertainty
of volume change $d\Delta V$, ice density uncertainty $d\rho$ ±60 kg/m³ and uncertainty of glaciated area
$dS$ estimated followed by (Paul et al., 2017).
$$\frac{dB}{B} = \sqrt{\left(\frac{d\Delta V}{\Delta V}\right)^2 + \left(\frac{d\rho}{\rho}\right)^2 + \left(\frac{dS}{S}\right)^2} \tag{17}$$


## 5.  Results and discussion

### *5.1. Glacier ice velocity estimates*

The Gangotri glacier's velocity pattern is consistent with a valley glacier, as depicted in Fig.
3a-d. The distribution of ice surface velocity across the Gangotri glacier shows significant
variation from the snout to the accumulation area, as well as among its tributaries. During the
study period from 2016 to 2023 (Fig. S2), the mean surface velocity for the entire glacier is



observed to be 34.80 ± 6.5 m a⁻¹ based on the GIV tool (Fig. 3a), while the COSI-Corr-based
results (Fig. 3b) indicated a slightly lower mean velocity of 29.12 ± 3.6 m a⁻¹.
Additionally, velocity estimation using the ImGRAFT (Fig. 3c) algorithm yielded a mean
velocity of 38.26 ± 1.4 m a⁻¹, highlighting some variations across different feature-tracking
methodologies. The glacier exhibited a maximum surface velocity of approximately 100 ±
18.67 m a⁻¹, predominantly observed near the Equilibrium Line Altitude (ELA) at
approximately 5100 m a.s.l (Fig. 3e). In the accumulation area, the velocity ranged from 10 ±
1.86 m a⁻¹ to 100 ± 18.67 m a⁻¹, with the maximum velocities occurring close to the ELA.
Conversely, in the ablation area, the surface velocity ranged from 16 ± 2.98 m a⁻¹ to 99 ± 18.49
m a⁻¹. The main trunk of the glacier exhibited velocities between 16 ± 2.98 m a⁻¹ and 49.99 ±
9.3 m a⁻¹, indicating relatively moderate flow dynamics compared to the accumulation zone.
### *5.2. Glacier ice thickness estimates*
The ice thickness of the Gangotri Glacier varies significantly across its extent, as shown in Fig.
4a, ranging from 50 ± 6.42 m to 493 ± 63.3 m, with an observed mean ice thickness of
approximately 147.32 ± 18.93 m. In the region surrounding the glacier's snout, the ice thickness
ranges between 99.35 ± 12.73 m and 130 ± 16.64 m. The maximum ice thickness is observed
in the upper reaches of the glacier's main trunk, specifically the central part, reaching up to 500
± 64 m. A detailed elevation-wise distribution of the glacier ice thickness is presented in Table
2. The time series estimated ice thickness distribution of the Gangotri glacier in Fig. S3.

**Table 2.** Ice thickness distribution over 500m elevation band

| Elevation range (m a.s.l) | Min. thickness | Max. thickness | Mean thickness |
|---|---|---|---|
| 4000 - 4500 | 99.37 ± 12.8 | 257.97 ± 33.14 | 180 ± 23.12 |
| 4500 - 5000 | 59.84 ± 7.68 | 456.64 ± 58.63 | 212 ± 27.22 |
| 5000 - 5500 | 59.40 ± 7.62 | 493.63 ± 63.38 | 141 ± 18.10 |
| 5500 - 6000 | 58.85 ± 7.55 | 142.22 ± 18.26 | 89.3 ± 11.46 |
| 6000 - 6500 | 58.37 ± 7.49 | 107.72 ± 13.83 | 77.01 ± 9.89 |
| 6500 - 7000 | 58.94 ± 7.56 | 114.33 ± 14.67 | 73.94 ± 9.50 |


The thickness distribution along the central flow line is illustrated in Fig. 4b plotted against
elevation. This graph also highlights the relationship with the ice thickness estimated by
Farinotti et al., (2019) providing a comparative analysis of the two results. The correlation
coefficient between the VWDV-based estimated ice thickness in this study and that from
Farinotti et al., (2019) was observed to be approximately 0.938.



**Figure 3:** Estimated velocity with different approaches. a) Mean velocity by GIV (2016 to 2023) using time-series imagery (152 images); b) Mean velocity by COSI Corr (2016 to 2023), using bi-temporal imagery (8 images); c) Velocity estimated by modified ImGRAFT (2016 to 2023), using multi-date imagery (8 images); d) shows the mean ITS_LIVE velocity dataset, and e) Graph showing the mean velocity and elevation along the central flow line of the glacier (A-A') and dotted red line depicts the ELA.



### 5.3. Glacier ice thinning rate and volumetric change estimates

The changes in ice thickness were determined by comparing ice thickness estimates annually across the years 2016 to 2023. A positive change indicated an increase in thickness at a specific location, likely due to factors such as snowfall, debris deposition due to avalanching, or ice accumulation (Vatsal et al., 2025a). Conversely, a negative change represented a decrease in thickness, which could be attributed to glacier melting, ice loss, or mass movement processes (Castellazzi et al., 2019).

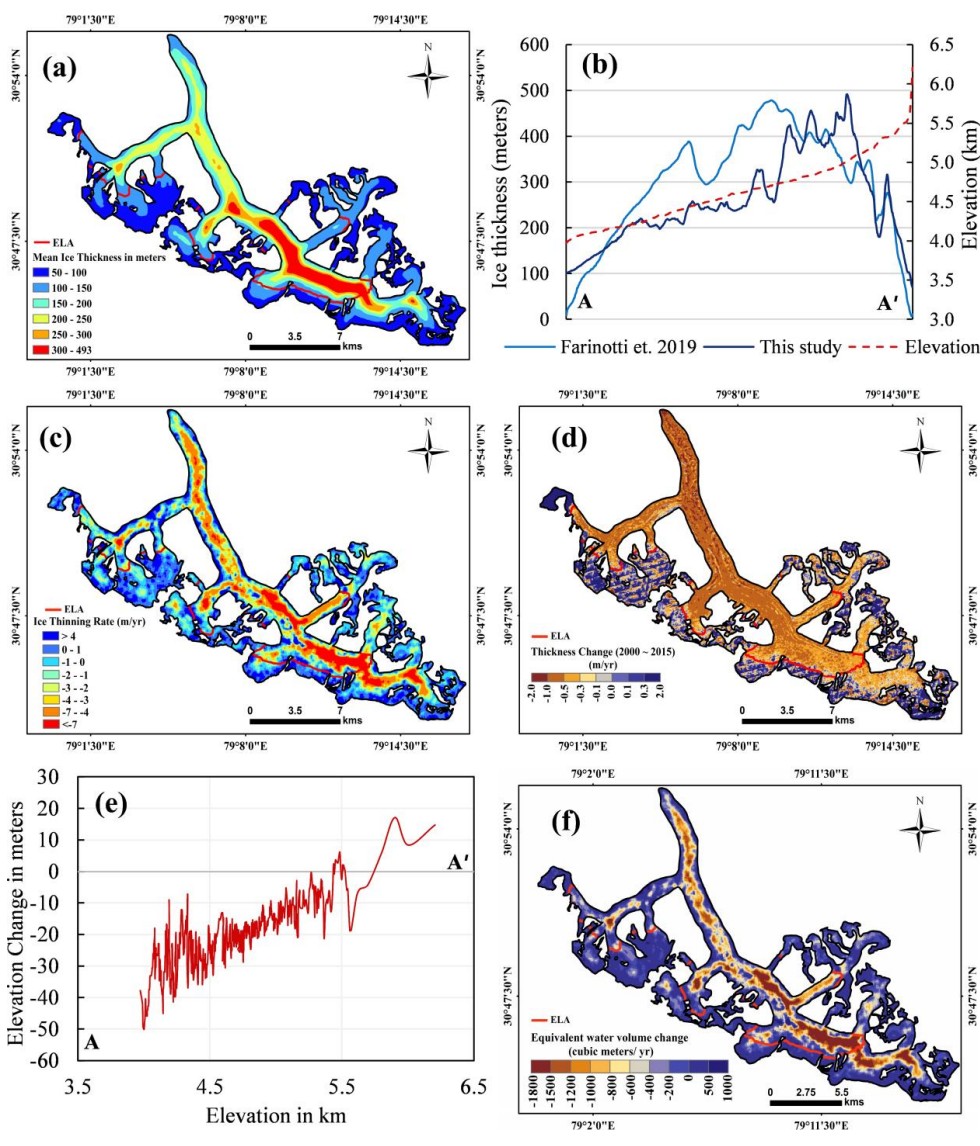

**Figure 4**: a) Mean thickness based on (VWDV) fully distributed velocity-based inversion (Van Wyk De Vries et al., 2022); b) Ice thickness along central flow line of Gangotri glacier (A-A');



c) Ice thinning rate per year; d) Elevation change per year; e) Elevation change along central
flow line (2000 and 2015) (A-A') and f) Equivalent water volume change.

The estimates from the VWDV ice thickness model (Van Wyk De Vries et al., 2022), indicate
that the mean ice thinning rate of the Gangotri Glacier is approximately -1.22 $\pm$ 0.482 m a$^{-1}$
(Fig. 4c). The ablation zone of the lower glacier exhibited a higher thinning rate, ranging
between 0 to -7 $\pm$ 2.76 m a$^{-1}$, compared to the overall glacier. In contrast, the left-hand side
(LHS) and right-hand side (RHS) tributaries displayed a positive thinning rate, varying between
0 to 1.5 $\pm$ 0.592 m a$^{-1}$. The upper accumulation areas of the glacier body and its tributaries
experienced positive thinning rates ranging from 0 to 5 $\pm$ 1.97 m a$^{-1}$, with an average value of
0.27 $\pm$ 0.106 m a$^{-1}$. Furthermore, the elevation change rate of the Gangotri Glacier is estimated
using the geodetic method (2000 ~ 2015), resulting in a rate of -0.50 $\pm$ 0.22 m a$^{-1}$ (Fig. 4d &
4e) with an average elevation change of -8.39 $\pm$ 3.31 m over the period 2000–2015.

**Table 3.** Periodic ice volume of the Gangotri glacier (based on the VWDV model)

| Year | Mean ice thickness (meters) | Ice volume (km$^3$) |
|---|---|---|
| 2016 | 145.45 $\pm$ 17.88 | 20.03 $\pm$ 2.46 |
| 2017 | 164.80 $\pm$ 24.42 | 22.79 $\pm$ 3.38 |
| 2018 | 133.31 $\pm$ 18.04 | 18.36 $\pm$ 1.87 |
| 2019 | 144.32 $\pm$ 18.09 | 19.87 $\pm$ 2.68 |
| 2020 | 136.75 $\pm$ 18.43 | 18.77 $\pm$ 2.53 |
| 2021 | 136.44 $\pm$ 20.16 | 18.89 $\pm$ 2.76 |
| 2022 | 144.28 $\pm$ 21.15 | 19.95 $\pm$ 2.91 |
| 2023 | 136.85 $\pm$ 18.39 | 18.85 $\pm$ 2.53 |


Table 3 provides a detailed overview of the mean ice thickness and volume of the Gangotri
Glacier for each year. These values exhibit considerable temporal variability, reflecting the
dynamic nature of the glacier's mass balance, influenced by factors such as climatic conditions,
snowfall, melting, and ice dynamics. The estimated equivalent water volume change is shown
in Fig. 4f. Over the main tributaries, the equivalent water volume change ranges from ~500 $\pm$
196.40 to -200 $\pm$ 78.55 m³a$^{-1}$, with the upper accumulation area showing a positive mean
equivalent water volume change of approximately 66.44 $\pm$ 26.09 m³a$^{-1}$. The mean equivalent
water volume change for the Gangotri Glacier is approximately -276.23 $\pm$ 108.50 m³a$^{-1}$, while
the main trunk exhibits a change of approximately -450 $\pm$ 176.75 m³a$^{-1}$.
*5.4. Mass balance estimates*
The mass balance of the Gangotri Glacier was assessed using time-series thickness data on the
glacier's thinning rate, as depicted in Fig. 5. The glacier's average mass balance is calculated to
be -1.04 $\pm$ 0.403 m w.e. a$^{-1}$, indicating a persistent negative trend. Fig. 5 presents the mass
balance distributed across 500 m elevation bands, derived from volumetric changes within
these areas. The lower main trunk of the glacier recorded a mass balance rate of -1.96 $\pm$ 0.78
m w.e. a$^{-1}$, while the upper region experienced the highest mass loss at approximately -2.01 $\pm$





0.78 m w.e. a⁻¹. In contrast, the accumulation zone above 5500 m a.s.l. exhibited a positive
mass balance, ranging from 0.192 ± 0.074 to 0.42 ± 0.162 m w.e. a⁻¹.

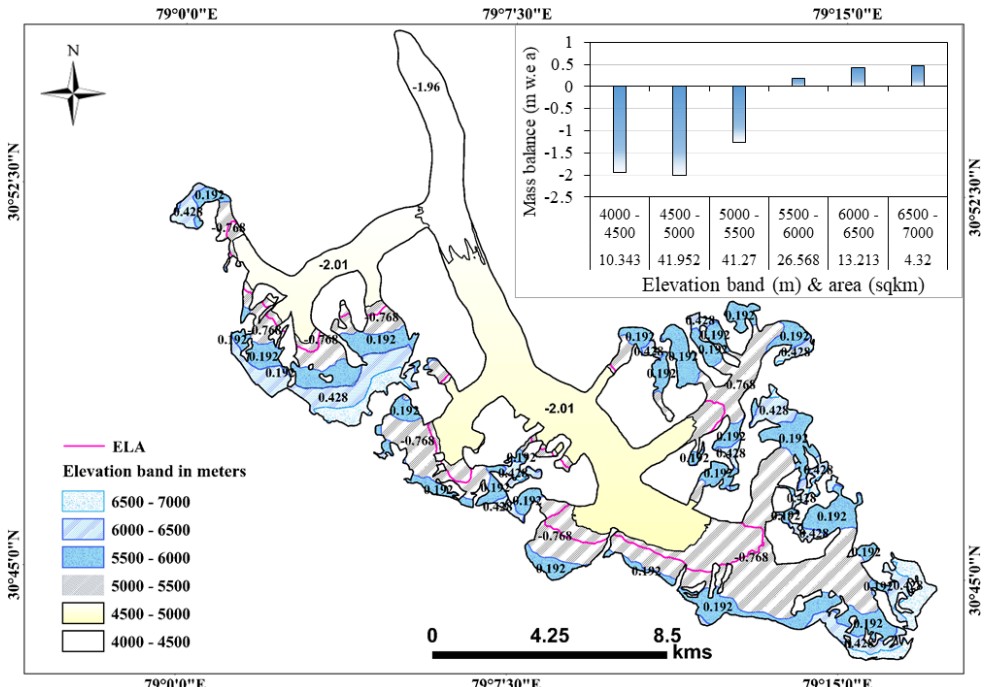

**Figure 5**: Specific mass balance (SMB) distribution of the Gangotri Glacier across 500 m
elevation bands. The graph in the upper right corner illustrates the specific mass balance
variations corresponding to different elevation zones. The map highlights maximum mass loss
in the ablation area, while higher accumulation zones exhibit significant mass gain.
**6. Discussion**
*6.1. Glacier velocities*

*6.1.1. Comparative assessment with ITS_LIVE*

This study investigates the ice surface velocity of the Gangotri Glacier by employing
three different feature-tracking approaches: GIV, COSI-Corr, and ImGRAFT. These methods
were evaluated by comparing their results with velocity estimates from the ITS LIVE dataset
(A. Gardner et al., 2022). Despite studies indicating a wide range of errors in the product, such
as a very low correlation 0.29 (RMSE 53.54) for the Kaskawulsh Glacier and the high
correlation 0.98  (RMSE 109) for the Lowell Glacier in the Yukon region, Canada, between
ITS_LIVE V2 and GPS data from 2017 to 2022), the dataset is widely used (Zhang et al.,
(2024), especially in the absence of ground-based velocity measurements.
The analysis focused on the central flow line and different glacier zones to ensure robust
comparisons by minimizing lateral variability (Fig. 6a). To further assess the reliability of the





findings, recent velocity estimates derived from Sentinel-1 data (Bhattacharjee & Garg, 2024)
were considered. Additionally, previous studies using the feature-tracking method (Gantayat
et al., 2014; Saraswat et al., 2013) were referenced for comparative analysis.

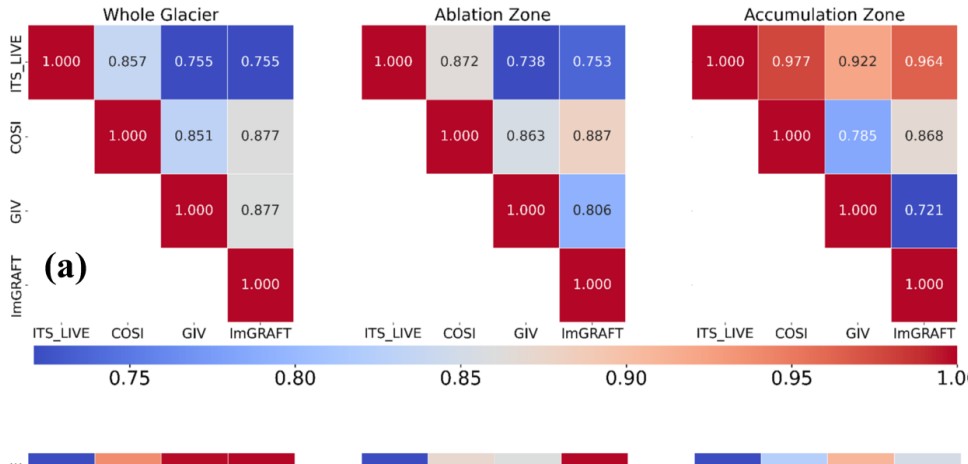

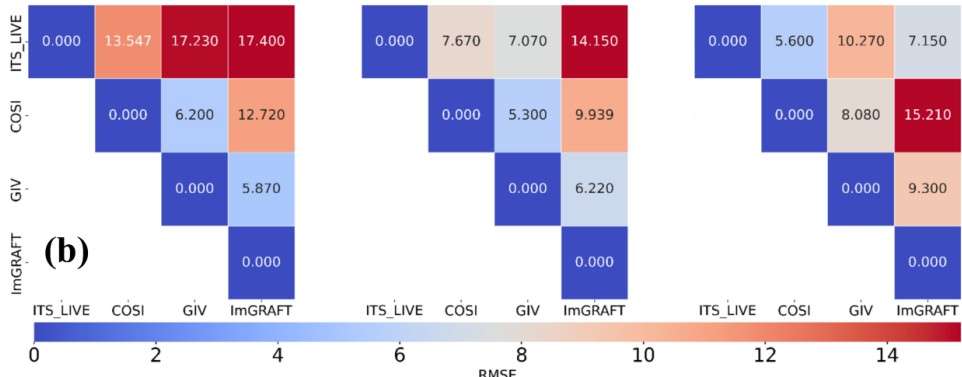

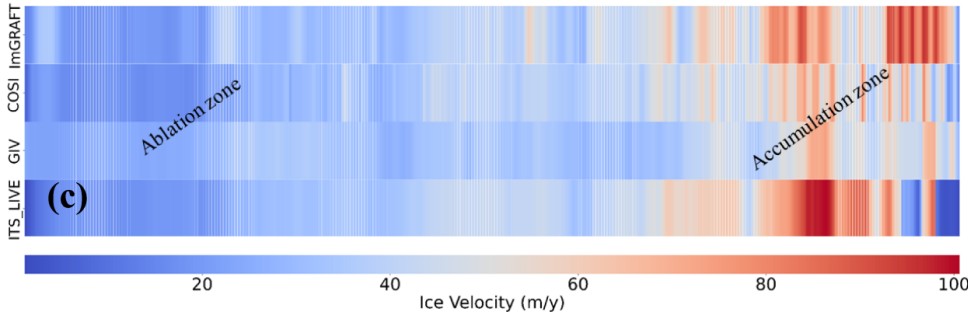


**Figure 6:** (a) Illustrates the correlation between different glacier velocity estimation
approaches for the Gangotri Glacier across various zones. (b) Root Mean Square Error (EMSE)
of the ice velocity between different approach and (c) Depicts the distribution of glacier





velocity along the central flow line, extending from the glacier terminus to the accumulation
peak (A–A').
Fig. 6 illustrates the statistical comparison between different glacier velocity estimation
methods across the Gangotri Glacier, with a specific focus on ITS LIVE as a reference. The
comparison follows a Least Absolute Residuals (LAR) fit approach with 95% prediction
bounds to ensure robust statistical assessment. In the whole glacier region, ITS_LIVE
velocities exhibit strong correlations with COSI Corr (0.857), GIV (0.755), and ImGRAFT
(0.755), indicating a generally good agreement across methods (Fig. 6b).
In the ablation zone, ITS_LIVE shows a higher correlation with COSI Corr (0.872) and
about similar correlation with GIV (0.738) and ImGRAFT (0.753), reflecting the influence of
surface melting, crevassing, and ice deformation, which can introduce uncertainties in velocity
retrieval (Fig. 6b). In contrast, the accumulation zone presents the strongest correlations, with
ITS_LIVE aligning closely with COSI Corr (0.977), GIV (0.922), and ImGRAFT (0.964),
suggesting more stable ice dynamics in this region due to reduced surface melting and a more
consistent ice mass flow (Fig. 6b).
Fig. 6c represents the glacier velocity distribution along the central flow line from the
terminus to the accumulation zone, highlighting distinct spatial variations in ice motion. The
velocity remains relatively low near the terminus due to high frictional resistance from bedrock
and debris cover, gradually increasing in the ablation zone where ice thinning and gravitational
flow enhance movement (Nicholson et al., 2018). In the accumulation zone, the velocity
reaches its peak, attributed to the increased ice mass and steep surface gradients (Vatsal et al.,
2025b). The observed variations of the entire glacier velocity reflect the combined effects of
topography, ice thickness, and surface conditions, with the ELA emerging as a critical zone for
maximum ice movement. The strong correlation in this region suggests that all methods
effectively capture glacier dynamics in areas with consistent ice movement. These results
demonstrate the compatibility of feature tracking techniques with well-established satellite
datasets, enhancing confidence in their application for glacier velocity mapping.
As delineated in Fig. 6a, the ITS_LIVE velocity captures these patterns of decreasing
velocity from accumulation to the ablation zone, followed by ImGRAFT, which also captures
similar pattern, however, with higher velocity in some accumulation zones in contrast to
ITS_LIVE. These differences in the velocity patterns are also statistically observed in the
higher RMSE for ImGRAFT compared with ITS_LIVE. It is worth mentioning here that for
the statistical comparison, the ITS_LIVE and the COSI-Corr velocities were resampled to 30m
from 120m and 60m, respectively. The relatively higher agreement between the COSI-Corr
velocity and the ITS_LIVE product may be attributed to the coarser resolution compared with
the GIV and ImGRAFT where the source spatial resolution of the velocity product was 30m.
Fig. 6c shows that the GIV and ImGRAFT methods largely underestimate the velocity in some
zones of the accumulation region. Overall, statistically, the GIV shows marginally better
performance than ImGRAFT and henceforth was considered for further investigations.

***6.1.2.  Assessment with other studies***

The mean velocity of the Gangotri Glacier was estimated in this study as 0.095 ± 0.017
md⁻¹, which is similar to the observations in a recent study (Bhattacharjee & Garg, 2024) which




reported a mean Gangotri Glacier velocity of 0.09 ± 0.008 m d⁻¹ using Sentinel-1 data.
Gantayat et al. (2014) estimated the ice velocity in the snout region of the Gangotri Glacier to
range between 20 and 30 ma⁻¹. Saraswat et al., (2013) also highlighted the velocity of the snout
area of the Gangotri Glacier was between 24.8 ± 2.3 ma⁻¹ and 28.9 ± 2.3 ma⁻¹. The velocity
estimates derived in the present study align well with the results of these studies.
In the study period, we observed a downslope acceleration in the Gangotri Glacier's
surface velocity as per time-series investigations based on the GIV tool. However, the
distribution of the surface velocity varied across the glacier. The marginal regions of the glacier
exhibited a decreasing trend in surface velocity which may be due to debris accumulation,
which increases friction and inhibits ice flow (Fig. S2). Additionally, the presence of stagnant
ice and reduced ice thickness in marginal areas further contributes to slower movement, as the
driving stress diminishes near the margins, whereas the central trunk of the glacier displayed
an increasing trend during the study period.

### 635 *6.2. Glacier thickness*

#### 636 *6.2.1. Assessment with other studies*

Due to the lack of in-situ ice thickness measurements for the Gangotri glacier in the
public domain and as available to us, we conducted an assessment of the velocity-based ice
thickness results based on previous studies (Bhattacharjee & Garg, 2024; Bhushan et al., 2017;
Gantayat et al., 2014; Nela et al., 2023). We evaluated different ice-thickness estimation
methods for the year 2016 for the Gangotri glacier for comparative assessment. Among these,
we present the results from velocity-based (Van Wyk De Vries et al., 2022) (Fig. 7a), and
stress-based (GT2b and GT2w) (Fig. 7b & 7c) (Sinha et al., 2024) approaches. The ice
thickness uncertainty is depicted in Fig. 7d. Fig. 7e shows the correlation matrix of ice
thickness derived from different approaches for the Gangotri glacier corresponding to Farinotti
et al., (2019). Farinotti et al., (2019) estimated surface topography-based ice thickness, further
providing a global dataset, which is widely used in comparisons (Nela et al., 2023; Van Wyk
De Vries et al., 2022). The robust fit coefficient of determination between VWDV and Farinotti
estimated thickness was observed to be 0.88, which depicts a strong positive correlation where
the RMSE is also comparatively lower than other methods (Fig. 7f). Stress-based (GT2b and
GT2w) estimates of ice thickness provided a strong correlation with Farinotti's estimated ice
thickness which exhibited a highly positive correlation coefficient of 0.883 and 0.894,
respectively.
The average thickness of the Gangotri Glacier estimated by Farinotti based on surface
topography was 139.57 m (Farinotti et al., 2019). The calculated mean ice thickness using the
VWDV method was 147.32 where the uncertainty was ±18.93 m (Fig. 7d) and the ice thickness
pattern was observed to be similar to Farinotti (2019) along the central flow line of the glacier
(Fig. 4f). Differential interferometric SAR velocity-based ice thickness was estimated over
Uttarakhand state where the mean thickness of that region was ~112 m (Nela et al., 2023) where
Gangotri glacier was showing highest ice thickness was of the order of 400 m, approximately.
Another study done by (Bhusan et al., 2017) based on ice surface velocity estimated that the
thickness of the upper reaches of the glacier was around 200 m to 400 m. The maximum ice-
thickness of the results from the VWDV method was 493 m, which aligns well with these
observations.

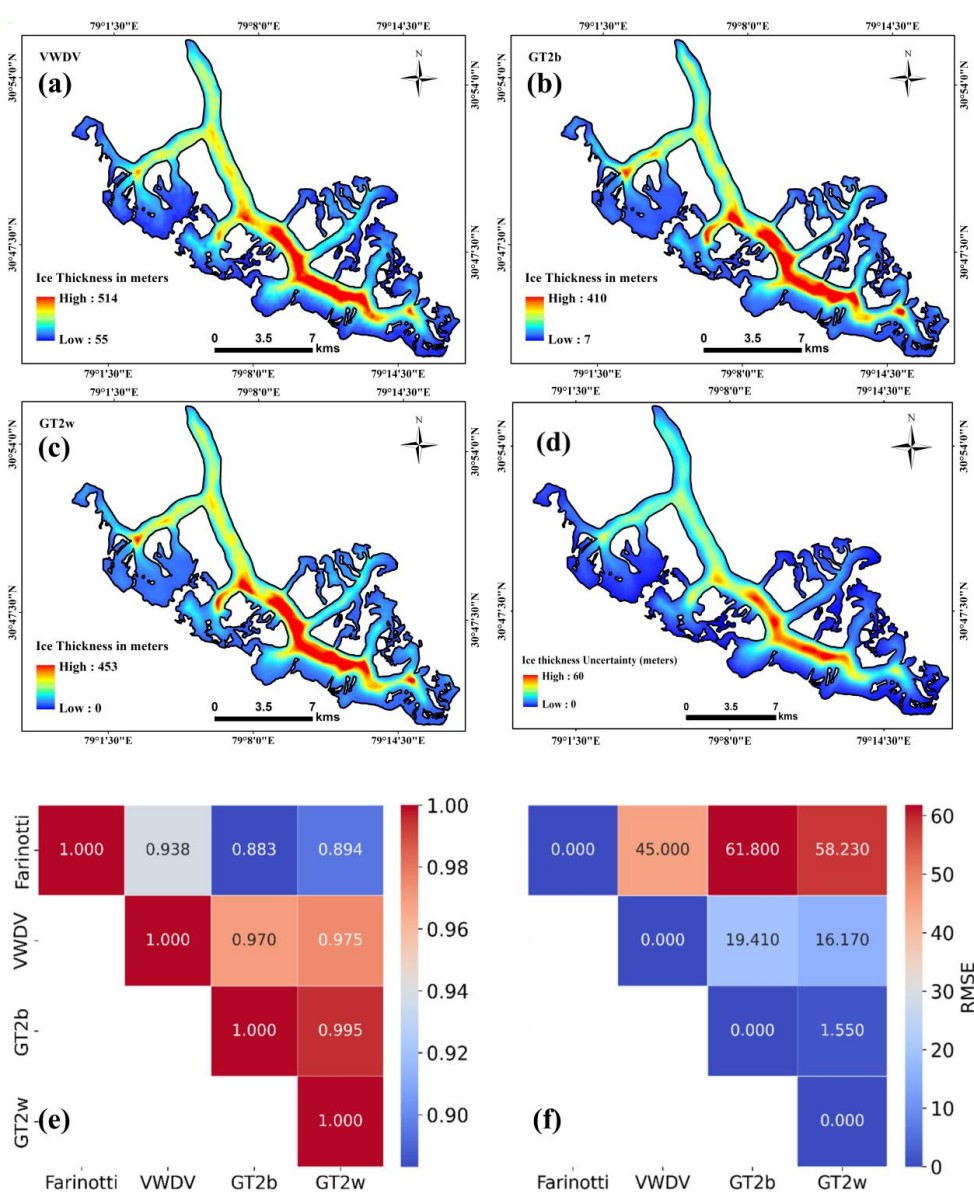

**Figure 7:** Applied three models for ice thickness estimation and their comparison: a) VWDV
is the fully distributed velocity-based inversion (Van Wyk De Vries et al., 2022); b) GT2b is
the basal-shear-stress-based basin-divided approach (Linsbauer et al., 2012b); c) GT2w is the
basal-shear-stress-based whole glacier approach (Ramsankaran et al., 2018b); d) Mean
thickness uncertainty (meters) e) correlation among all model and f) RMSE between different
model



### 6.2.2. *Assessment of Ice thinning rates*


In this study, two approaches were employed to estimate the thinning rate of the
Gangotri Glacier (Section 3.5). We relied on DEM differencing to assess glacier thinning
trends. Previous studies have reported similar estimates, providing a basis for comparison.
Bhushan et al. (2017) reported a mean thinning rate of -0.64 ± 0.49 ma$^{-1}$, with an average
elevation change of -10.78 ± 7.43m over the period 2000–2014, based on differencing SRTM
2000 and Cartosat 2014 DEM with 30 m spatial resolution. Similarly, Gardner et al. (2013)
utilized the ICESat altimetry bi-temporal dataset to estimate an elevation change rate of -0.44
± 0.20 ma$^{-1}$ for parts of the Central Himalaya during 2003–2009 (Gardner et al., 2013).
We estimated the mean thinning rate of the Gangotri Glacier using DEM differencing
between SRTM (2000) and COP-DEM (2015), yielding a rate of -0.50 ± 0.22 ma$^{-1}$ (Fig. 4c &
4e), with an average elevation change of -8.39 ± 3.31 m over 2000–2015. Furthermore, we
estimated the mean ice thickness change rate, which was found to be -1.22 ± 0.48 ma$^{-1}$, further
emphasizing substantial ice mass loss. A recent study by Bhattacharjee & Garg (2024) reported
an ice thickness change rate by subtracting estimated ice thickness data from 2017 to 2022, and
an estimated ice thickness change of -0.62 ma$^{-1}$ for the Gangotri Glacier system (including its
tributaries). Overall, our study establishes a strong agreement with previous research,
reinforcing the consistency of both thinning rate and thickness change estimates. The observed
trends indicate a persistent and significant mass loss in the Gangotri Glacier, aligning well with
the broader patterns of glacial retreat reported in the region (Thapliyal et al., 2023).

### 6.3. Assessment of the Gangotri glacier ice volume and mass balance


The Gangotri glacier ice volume was estimated by Gantayat et al. (2014) using velocity-based
determined thickness of 23.4 ± 4.2 km³ in 2009 and 2010. Haq et al. (2014) estimated the ice
volume using artificial neural networks (ANN) and perfect plasticity to be 21.559 km³. Fig. 8
illustrates the time-series volumetric changes in Gangotri ice volume (pixel area × depth) and
the corresponding equivalent water volume of the Gangotri Glacier from 2016 to 2023, as
estimated in this study. Gangotri Glacier area as per our modified glacier boundaries was
determined to be 141.9 km$^2$. Fig. S4 depicts the glacier ice volume based on glacier area and
ice depth.

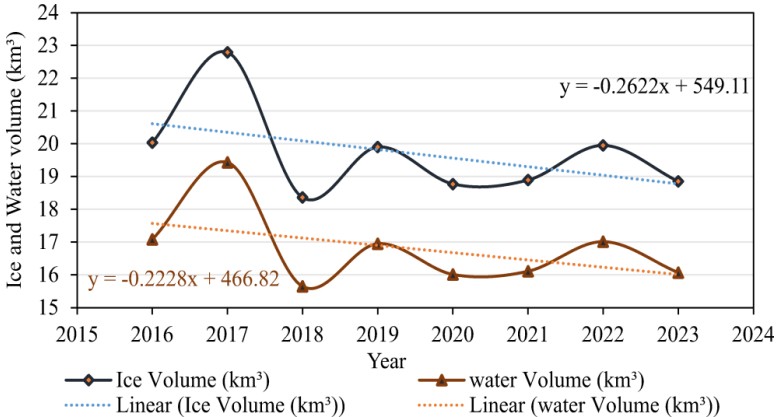







**Figure 8:** Piece-wise estimated ice volume and equivalent water volume of the Gangotri glacier
(2016 - 2023). Both curves indicate a declining trend with high variations between 2016-2018
and some undulations thereafter.

**Table 4.** Comparison of the Gangotri glacier ice volume and estimated mean thickness by (Frey
et al., 2014).

| Glacier parameters | (Chen & Ohmura, 1990) | (Bahr et al., 1997) | LIGG et al., (1988) | Slope dep. Thick (Frey et al., 2014) | GlapTop -2 (Frey et al., 2014) | HF model (Frey et al., 2014) | VWDV (This study) |
|---|---|---|---|---|---|---|---|
| **Thickness (metres)** | 177 | 218 | 224 | 91 | 145 | 149 | 147 ± 18.93 |
| **Volume (km³)** | 25.1 | 30.1 | 31.8 | 12.9 | 20.6 | 21.1 | 19.70 ± 2.64 |

A comparative analysis as exhibited in Table. 4, shows that the ice volume estimates derived
using the VWDV model with GIV-based velocities showed significant similarities with the
observations from the previous studies (Bahr et al., 1997; Chen & Ohmura, 1990; Frey et al.,
2014; LIGG et al., 1988).

Throughout the study period, the ice volume exhibited noticeable fluctuations, with a
trend indicating periods of accumulation and melting. The highest recorded ice volume was in
2017, reaching 22.799 ± 3.386 km³, suggesting significant ice accumulation or reduced melting
(Fig. 8). This trend was likely influenced by favourable climatic conditions, such as increased
snowfall or cooler temperatures, which can be attributed to the peak La Niña event that
prevailed during the accumulation period in 2017 (Kuttippurath et al., 2024). However,
following 2018, the emergence of El Niño conditions over the Niño 3.4 region contributed to
a shift in atmospheric patterns, potentially enhancing surface melting and reducing ice volume
(McPhaden, 2019). In contrast, the lowest ice volume was recorded in 2018, 2020 and 2023
indicating significant ice loss, which may be attributed to accelerated melting, reduced
snowfall, or other factors influencing glacier dynamics. The overall trend from 2017 to 2023
reveals a decline in ice volume, pointing to a negative mass balance in recent years.

Some studies estimated the mass balance of the Gangotri glacier or Gangotri glacier
system based on energy balance modelling and geodetic approach. Our estimated mass wastage
-1.01 ± 0.403 m w.e. a$^{-1}$ on Gangotri glacier show significant agreement with (Agrawal et al.,
2018) and (Hussain et al., 2022), where the mass wastage was estimated as -0.98 (1985–2005)
and -0.89 ± 0.31 (1999–2014) m w. e. a$^{-1}$ using energy balance model, respectively. Another
study estimated the mass wastage of the Gangotri glacier at -0.55 ± 0.42 m w.e. a$^{-1}$ using the
geodetic method over 1999–2014 (Bhushan et al., 2017). In Garhwal Himalayas some
benchmark glacier's (Dunagiri, Chorabari and Dokriani) mass balance (Azam et al., 2018) were
conducted with the glaciological approach which are considered for evaluating the results. The
measured mass balance of these glacier -1.04 m w.e. a$^{-1}$ (1984–1990), -0.73 m w.e. a$^{-1}$ (2003–
2010) and -0.39 m w.e. a$^{-1}$ (1997–2000) respectively (Azam et al., 2018), which align with our



mass balance results of the Gangotri glacier -1.01 ± 0.403 m w.e. a$^{-1}$ in the period of 2016–
2023.  The comparisons are detailed in Section 2 of the Supplementary Material (Table S4).

### *6.4. Assessment of influencing factors*

The glacier ice thinning is largely influenced by numerous factors, some already discussed
previously such as the climate anomalies. In the subsequent sub-sections, we attempt to
investigate the meteorological and anthropogenic factors.

#### *6.4.1. Maximum Temperature and precipitation*

Rising temperatures, particularly in high-altitude regions, accelerate glacier melt and
contribute to negative mass balance (Bhattacharjee & Chandra Pandey, 2022). The increase in
global mean surface temperature has significantly influenced the cryosphere, leading to glacier
retreat and thinning across various mountain ranges (IPCC, 2019). In this process, the freezing
altitude plays a crucial role by regulating the equilibrium line altitude (ELA), which divides
the glacier's accumulation and ablation zones (Ohmura & Boettcher, 2022). As temperature
rises, the ELA shifts upwards (towards higher altitudes), reducing the accumulation zone and
expanding the ablation area, resulting in accelerated glacier mass loss (Rabatel et al., 2013b).
Long-term temperature trends help assess glacier mass balance variability with studies showing
that warming trends in the Himalayas, Andes, and Alps have resulted in substantial reductions
in glacier volume over the past decades (Bolch et al., 2012b; Pandey et al., 2025). In most
regions globally, increased temperatures have led to significant glacier mass loss and increased
meltwater runoff, affecting downstream hydrology (Immerzeel et al., 2010b).





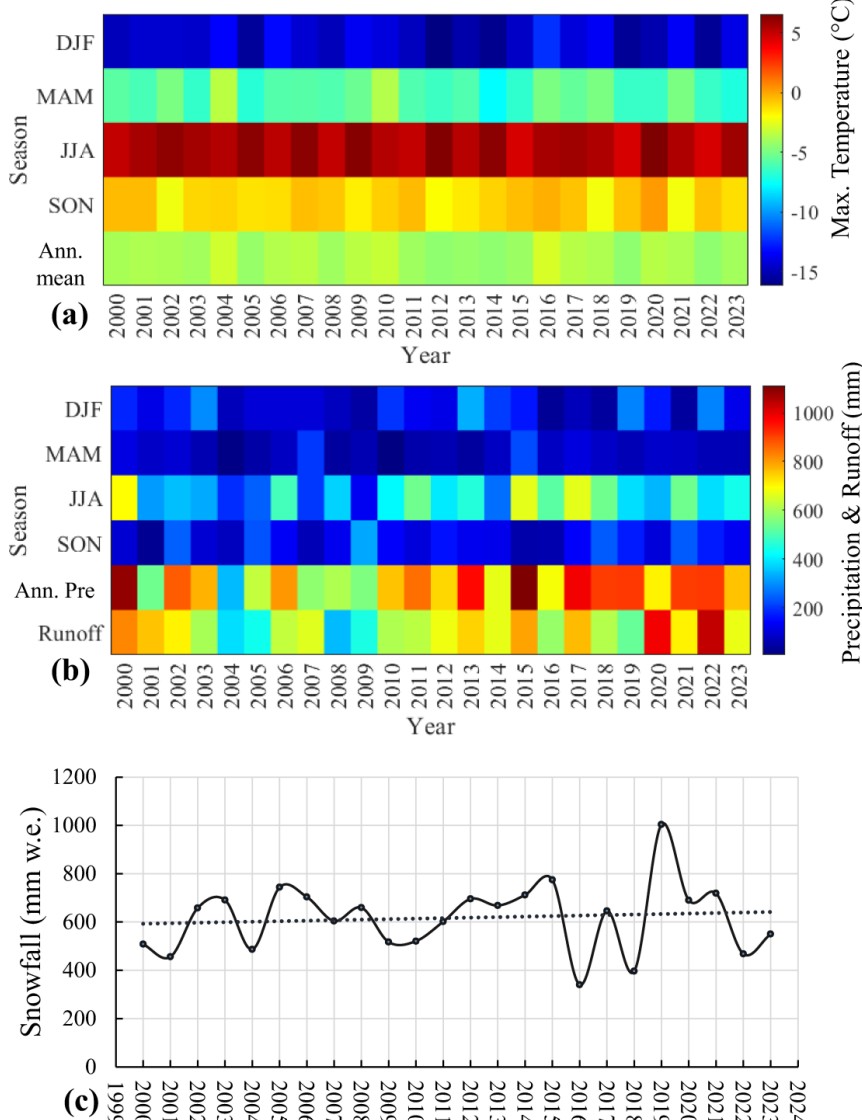

**Figure 9:** a) Seasonal Maximum Temperature (°C): This plot illustrates the seasonal variations in the maximum temperature observed at the Gangotri Glacier over time. Represents a distinct season: Winter (December–February, DJF), Pre-Monsoon (March–May, MAM), Monsoon (June–August, JJA), Post-Monsoon (September–November, SON) and annual mean. b) Seasonal and Annual Precipitation and Total Runoff (mm): This plot represents the seasonal (DJF, MAM, JJA, SON) and annual precipitation patterns, alongside the total runoff observed at the glacier. c) Annual snowfall of the Gangotri glacier derived from ERA5 reanalysis data.

Snowfall is the primary source of glacier mass accumulation, and its variability directly affects glacier growth and sustainability. Annual and seasonal precipitation trends dictate the balance between glacier accumulation and ablation (Fujita & Nuimura, 2011). In regions where





snowfall is decreasing or shifting toward rainfall due to rising temperatures, glaciers experience reduced accumulation and enhanced melting, leading to overall volumetric decline (Etter et al., 2017). Studies in the Himalayan and Andean glacier systems show that declining winter snowfall has weakened glacier regeneration processes, leading to negative mass balance trends (Shrestha et al., 2017). Changes in monsoon intensity, duration, and moisture availability significantly impact these glaciers (Kaser et al., 2010). Moreover, precipitation anomalies due to changing climate patterns, such as El Niño-Southern Oscillation (ENSO) events, can further disrupt glacier accumulation processes. ENSO-induced droughts or heavy precipitation in different regions cause glaciers to either suffer from extreme melting or benefit from excessive snowfall, influencing volumetric changes on a regional scale (Vuille, 2013). This study analysed the temperature, precipitation, and runoff trends over the Gangotri Glacier from 2000 to 2023 using the Terra climate dataset (Abatzoglou et al., 2018) to assess their potential influence on glacier dynamics. The analysis revealed that the maximum temperature trend (Fig. S5) does not exhibit a consistently increasing pattern across the entire study period (Fig. 9a). However, a slight warming trend was detected during the SON (September–October–November) season, which may contribute to enhanced late-monsoon and post-monsoon melting processes (Dobhal et al., 2013). Such seasonal variations in temperature could influence the glacier's surface energy balance, affecting the rate and timing of meltwater generation.

In contrast, both annual precipitation and runoff (Fig. S6 & S7) demonstrated an overall increasing trend throughout the study period (Fig. 9b). The rising precipitation may be attributed to changing monsoonal patterns or increased winter snowfall (Fig. 9c), potentially contributing to seasonal mass gain in certain glacier zones. However, the concurrent increase in runoff suggests that the glacier is experiencing enhanced meltwater discharge, which possibly aligns with the observed declining glacier ice volume in the Gangotri region. This increase in runoff, despite variable temperature trends, implies that other factors, such as changes in snowpack, ice dynamics, or alterations in precipitation phases (rain vs. snow), may be influencing the glacier's hydrological response. These findings reinforce the concerns about the ongoing glacial mass losses in the Gangotri glacier basin.

### 6.4.2. *Air quality and aerosol concentration*

The seasonal variation in Aerosol Optical Depth (AOD) (Fig. S8) and PM2.5 (Fig. S9) concentrations over Uttarakhand reveals critical trends that may contribute to glacier mass loss in the region (Fig. 10). The AOD values (Fig. 10a) exhibit significant interannual variability, with higher concentrations during the pre-monsoon (MAM) and monsoon (JJA) seasons. These elevated aerosol levels, primarily driven by rural biomass burning, dust transport, and vehicular emissions, increase atmospheric radiative forcing, leading to enhanced atmospheric warming and surface melt (Thanveer et al., 2024). Similarly, the PM2.5 concentrations (Fig. 10b) show a rising trend, particularly after 2015, with peak values observed in pre-monsoon months. The deposition of black carbon and fine particulate matter on glacier surfaces reduces surface albedo, thereby intensifying glacial melt rates (Hassan et al., 2023).

Our study has depicted a negative mass balance for the Gangotri Glacier, indicating sustained ice loss over recent years. The increasing aerosol loading in the region further



exacerbates this decline by enhancing surface energy absorption and accelerated meltwater
production (Thanveer et al., 2024). Given the high sensitivity of the Himalayan glaciers to
atmospheric pollution, these findings underscore the role of aerosols in glacier mass loss and
retreat. The strong seasonal dependence of AOD and PM2.5 suggests that spring and summer
months are critical periods where atmospheric deposition significantly influences glacier
dynamics (Hassan et al., 2023).

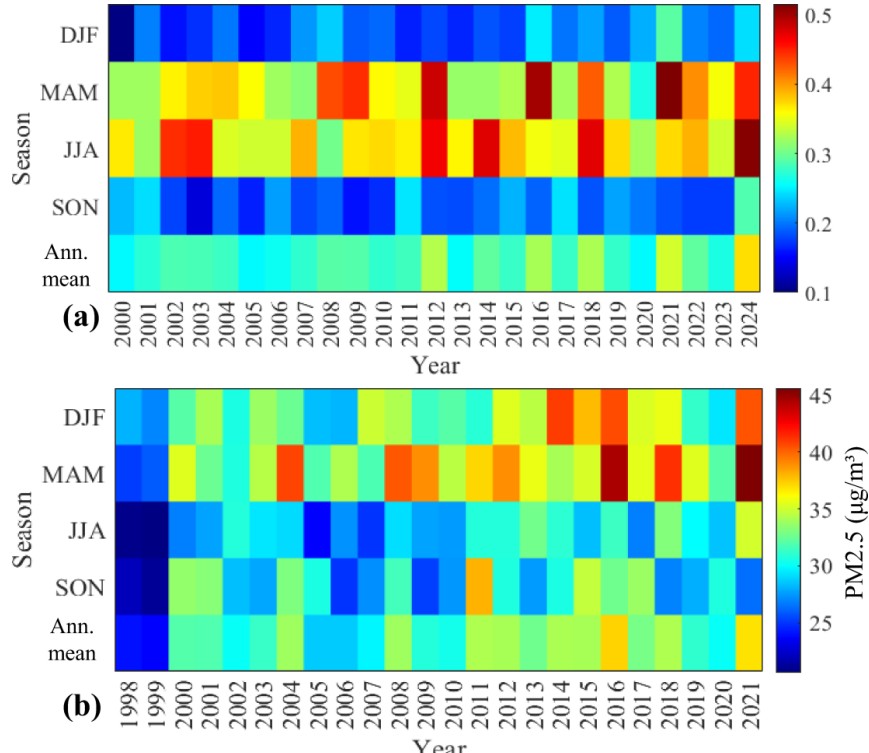


**Figure 10:**  a) Seasonal Aerosol Optical Depth (AOD) Concentration in Uttarakhand State
(DJF, MAM, JJA, SON and Annual mean): This figure represents the seasonal variations in
Aerosol Optical Depth (AOD) across Uttarakhand. AOD indicates the level of atmospheric
aerosols, which influence air quality. b) Seasonal PM2.5 (μg/m³) Concentration in Uttarakhand
State (DJF, MAM, JJA, SON and Annual mean): The seasonal variations in PM2.5
concentrations, representing fine particulate matter levels in the atmosphere. The seasonal
trends (DJF, MAM, JJA, SON) highlight variations in pollution sources, atmospheric transport,
and meteorological influences on air quality in Uttarakhand.

*6.4.3. Increasing Trend of Apparent Thermal Inertia in the Gangotri Glacier*

The intensity of diurnal melt-freeze cycles in temperate glaciers is significant in
assessing the overall mass loss over the season. Apparent Thermal Inertia (ATI) is a crucial
parameter for assessing the thermal response of glacier surfaces and is calculated using the
equation 18 (Van Doninck et al., 2011):



$$ATI = C\frac{(1-\alpha)}{\Delta T} \tag{18}$$

where $\alpha$ represents the surface albedo, and $C = 0.84$ is the solar correction factor, which
accounts for variations in incoming solar radiation calculated according to (Nicholas & Locke,
1982) and $\Delta T$ is the diurnal temperature range. This equation highlights the fundamental
relationship between surface reflectivity, thermal variation, and energy retention, making ATI
an effective indicator of glacier melt dynamics.

ATI may explain how glaciers respond to significant changes in the temperature and is
often an under looked parameter (Brenning et al., 2012). For exposed glacier ice or debris
covered ice as in the case of the Gangotri glacier, the ATI can describe how rapidly or gradually
the ice mass loss occurs under temperature fluctuations providing significant insights into the
melting processes (Fig. 11). A lower ATI of the glaciers may indicate their susceptibility to
melting, as they respond swiftly to temperature fluctuations by heating up or cooling down
rapidly, especially during summer or warmer conditions (Brenning et al., 2011). Apparently,
from equation 18, the texture and reflectivity of snow plays a significant role in determining
the ATI. Gangotri glacier has a rough texture owing to a significant debris cover which forms
critical meltwater pools in the summers (Fig. 12). The rough texture and lower albedo results
in a lower ATI for such glaciers. Further, the melt water pools behave similarly to a heat sink,
reducing the thermal inertia and accelerating mass wastage.
Satellite-based analyses indicate an increasing trend in ATI (Fig. 11) over the Gangotri
Glacier, suggesting a change in surface properties like reduced albedo, and surface
roughness/texture, leading to enhanced melt processes, and consequently, negative mass
balance. Observations from MODIS data (MCD43A3) reveals a decline in glacier albedo due
to debris cover, dust deposition, and melting ice exposure, leading to increased solar energy
absorption and a subsequent rise in ATI (Shaw et al., 2021). Furthermore, MODIS thermal
infrared (TIR) data (MOD11A1) indicate a rise in daytime LST, combined with reduced albedo
may create ripe conditions for enhanced melting.
The Gangotri Glacier has witnessed significant expansion of debris-covered areas,
which retains heat far longer and delays night-time cooling. This effect is further confirmed by
thermal anomalies detected in ASTER and Landsat TIR imagery, highlighting the thermal
buffering effect of supraglacial debris (Bhambri et al., 2011). Such changes contribute to a
more persistent melting regime, affecting overall glacier stability and mass balance. The
increasing trend in Aerosol Optical Depth (AOD) and PM2.5 concentrations in the Uttarakhand
region further amplifies the concern about glacier degradation. The AOD, which represents the
concentration of aerosols (dust, smoke, and pollutants) in the atmosphere, has shown a rising
trend, particularly during the pre-monsoon (MAM) and monsoon (JJA) seasons. Higher AOD
values indicate an increased presence of suspended and deposited particles, which can
significantly affect radiative forcing by trapping more heat and reducing surface albedo when
these aerosols settle on glaciers.
Similarly, the rising trend of PM2.5 (particulate matter with a diameter of 2.5
micrometres or smaller) suggests worsening air quality. These fine particles originate from





vehicular emissions, biomass burning, forest fire and industrial activities, and their deposition
on glacier surfaces leads to a phenomenon known as light-absorbing impurities (LAIs)
(Thanveer et al., 2024). The darkening effect caused by PM2.5 and AOD deposition increases
the absorption of solar radiation, which further reduces albedo and enhances melting rates. The
combined effect of rising AOD and PM2.5 concentrations with increasing ATI values indicates
a strong linkage between atmospheric pollution and glacier melt dynamics of the glacier. The
deposition of black carbon and other pollutants on glaciers is likely to exacerbate the warming
effect, leading to accelerated ice loss (Kinnard et al., 2022). This has significant implications
for hydrological regimes, as increased glacier melting contributes to higher runoff. Moreover,
the timely similarities between the peaks of AOD/PM2.5 concentration and sudden spikes in
ATI (2015 and 2016) suggest that air pollution plays a critical role in modifying the glacier's
thermal response.

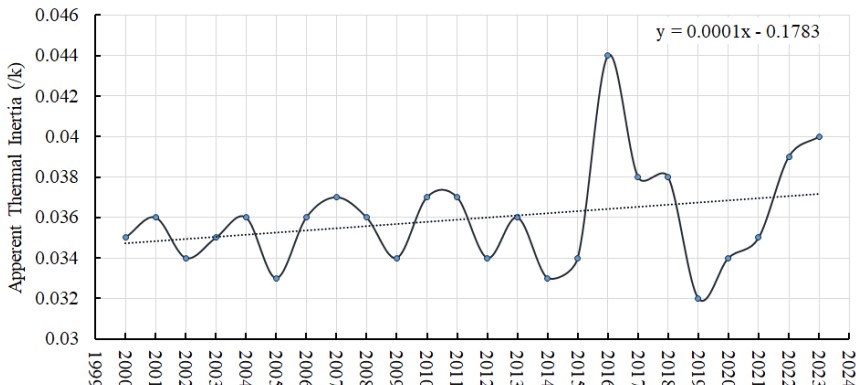


**Figure 11:** Apparent Thermal Inertia (ATI) ($K^{-1}$): This graph represents the satellite-based
annual trend of ATI for the Gangotri Glacier. ATI is derived from remote sensing observations
and indicates the glacier's thermal response, reflecting changes in surface properties, heat
retention, and potential melting patterns over time.



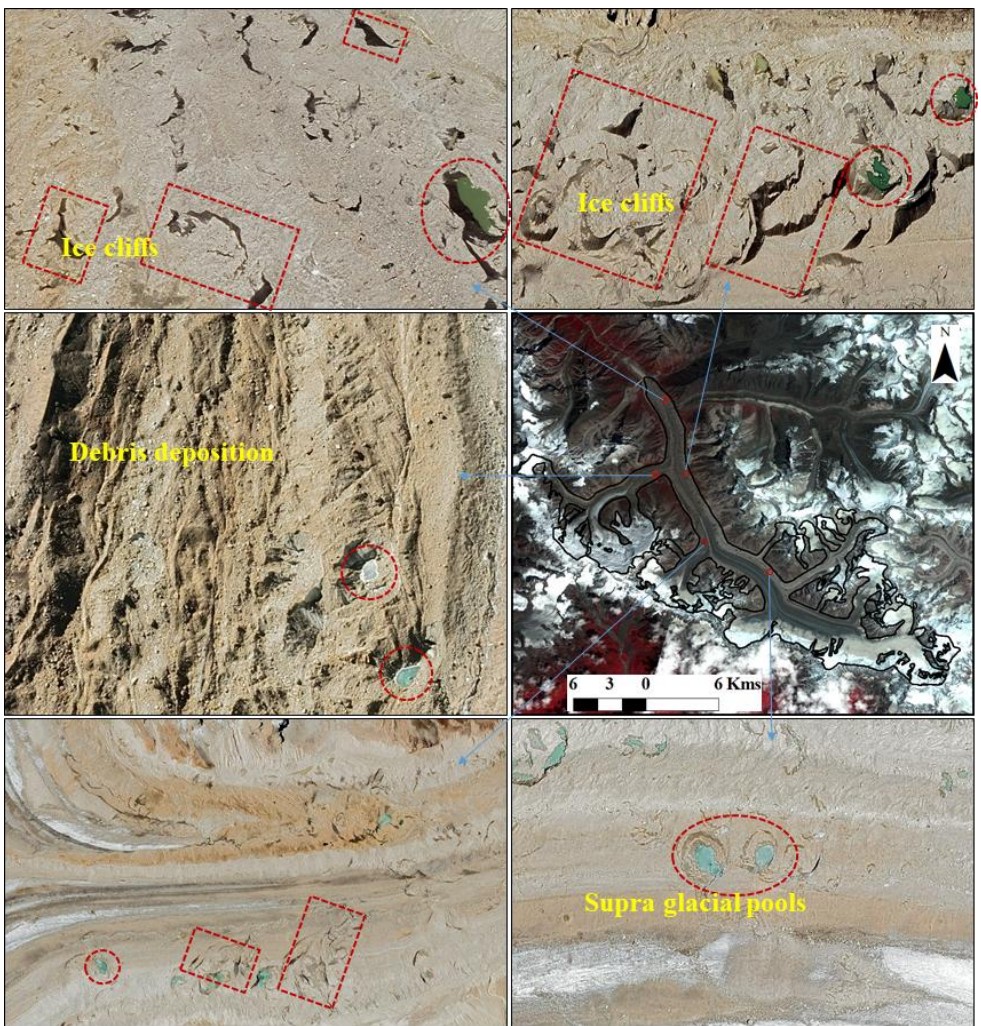


**Figure 12:** The heat-trapping hotspot zones over the Gangotri Glacier surface, where low albedo and high debris cover accelerate mass wastage. The ATI of these regions indicates rapid temperature response, leading to intensified melting. Features such as ice cliffs, supraglacial pools, and debris deposition influence thermal dynamics, intensifying glacier ice loss and enhancing surface instability. All copyrights Image ©2017 and before DigitalGlobe and after Maxar Technologies, accessed 2024.

### 6.4.4. *Anthropogenic factors*

Anthropogenic factors play a significant role in altering climatic patterns by increasing the concentration of greenhouse gases. The impacts of climate change are most prominently observed in the cryospheric regions, particularly in glaciers and permafrost. The Hindu Kush Himalaya serves as a key indicator of climate change, exhibiting increasing vulnerability to hazardous conditions over the past decades. Rapid urbanization and industrialization in the



Himalayan regions have significantly affected water storage systems, leading to drastic changes in hydrological balance. Land Use Land Cover (LULC) changes in Uttarakhand highlight the growing influence of human activities. Several studies utilizing satellite-based LULC assessments have reported an increasing trend in artificial structures (e.g., built-up areas, mining activities) and a declining trend in forest cover (Arun Kumar et al., 2019; Mani et al., 2023). The Land Use Land Cover (LULC) patterns in Uttarakhand for 2017 and 2023 indicate an expansion of built-up areas and a reduction in forest cover (Fig. S10). The results highlight a notable increase in urbanization, industrialization, and large-scale infrastructure projects, particularly due to the rapid expansion of roads, hydropower projects, and tourism-related developments as reported over the past decade (Mehta et al., 2013). Large-scale construction activities increase dust deposition on glacier surfaces, reducing albedo and accelerating ice melt (Dobhal et al., 2013). These changes may have indirectly contributed to glacier mass wastage and volumetric reduction by altering regional temperature and precipitation patterns.

### 6.5. Limitations and future scope of work

The present analysis is entire conducted based on satellite remote sensing data, and observations from the existing literature. Due to the absence of any in-situ field measurements, we were unable to verify the results and hence, uncertainties were estimated for each of the derived glacier products. While this approach ensures comparability with past studies, it also introduces potential uncertainties that must be carefully evaluated. It is worth mentioning that in various other studies on the Gangotri glacier, reliable ground-based observations have also largely been missing, as is the case for several other glaciers (Majeed et al., 2021; Paul & Linsbauer, 2012; Rashid & Majeed, 2018; Sattar et al., 2019; Shea et al., 2015).

Uncertainties in the proposed framework propagate from the velocities to the volume estimation and the mass balance derived from the ice thinning. Although we have made every effort to determine the uncertainties, several factors in practice should be considered, including such as the spatial resolution of the satellite products, especially considering that the Gangotri glacier is highly dominated by debris cover in the main trunk, subsequently, the velocities which are based on feature tracking may also have some uncertainties which may not have been fully realized. Similarly, the ice-thickness estimation also includes several assumptions in the absence of actual ground-based values, for example, the ice density. Notably, the ice density may not remain constant as considered but varies spatially. Further errors in thickness modelling can arise from multiple sources, image processing limitations due to sensor constraints, and inaccuracies in the DEM that affect elevation-dependent calculations. Additionally, uncertainties in defining the glacier boundary may also contribute to potential errors. The use of coarse resolution MODIS data for land surface temperature (used in deriving the Arrhenius creep factor) further adds to the uncertainty, as it may produce bias in maximum and minimum magnitudes of LST, potentially impacting the thickness estimation.

To enhance the reliability of glacier velocity and ice thickness estimations, future studies should incorporate field-based validation techniques. For velocity measurements, stake-based velocity estimation using GPS markers can provide high-precision results. Similarly, for ice thickness estimation, ground-penetrating radar (GPR) (Azam et al., 2012) surveys can offer direct measurements, reducing uncertainties associated with remote sensing-based approaches. Advancements in remote sensing methodologies can significantly improve



accuracy. The use of differential interferometric synthetic aperture radar (DInSAR) (Nela et
al., 2023) with very low temporal imaging gaps can refine glacier velocity estimations by
minimizing decorrelation effects. Additionally, Unmanned Aerial Vehicles (UAVs) (Bhardwaj
et al., 2016) can be utilized to capture high-resolution images, enabling more precise glacier
boundary delineation with reduced uncertainty. Integrating these advanced techniques with
remote sensing-based modelling will enhance the accuracy and reliability of glacier parameter
estimation, contributing to a better understanding of glacial dynamics and climate change
impacts.

## 7. Conclusion

Despite certain limitations, as discussed earlier, the present study provides significant insights
into glacier dynamics by utilizing a velocity measurement approach, applying different models
for ice thickness assessment, and performing a time-series analysis of volumetric changes. This
study underscores the applicability of model-based estimation of mass wastage and equivalent
water volume change in the Gangotri glacier, offering a more comprehensive understanding of
glacier's behavior.
Different feature-tracking approaches were applied to estimate the dynamics of the
Gangotri Glacier, demonstrating their robustness in capturing glacier motion. Despite
variations in algorithmic techniques, we found that optical imagery effectively captured ice
dynamics. The output resolution of each method varied, with ImGRAFT providing a finer
resolution (10m) compared to the GIV and COSI-Corr methods (60m). ImGRAFT
demonstrated a higher capability in capturing better glacier dynamics while preserving a
consistent flow pattern in the accumulation region, where the ice moves faster due to
topographic influence and extensional forces. All three methods exhibited a relatively similar
velocity structure along the central flow line, showing an increasing trend up to the ELA,
followed by a stagnation phase, and then a decreasing pattern. For ice thickness estimation, the
velocity and ice surface slope-based model (VWDV) were applied across the entire study
period, while the shear-stress-based model was used for a comparative assessment. The VWDV
model proved to be more reliable than the shear-stress-based approach, demonstrating better
consistency throughout the study period for the Gangotri glacier.
The Gangotri Glacier is a valley-type glacier in the Garhwal Himalaya. The surface,
particularly in the ablation zone, is characterized by rough textures, thick debris cover, large
crevasses, ice cliffs, and supraglacial pools, all of which influence thermal response and
melting behavior. The most significant finding of this study is the substantial down wasting of
the Gangotri Glacier during the study period, with maximum glacier mass wastage observed in
the upper ablation zone. The ongoing trend of climate warming has caused significant changes
in precipitation and temperature patterns, indicating further ice loss in the future. As one of the
most significant glaciers in Asia, continuous monitoring is essential for understanding its long-
term evolution under changing climatic conditions. Establishing a comprehensive and well-
maintained public observation facility dedicated to the Gangotri Glacier would facilitate
systematic monitoring of mass balance, ice dynamics, meltwater discharge, and glacial hazards.
Such a facility would support long-term glacio-hydrological modelling, improve climate
impact assessments, and enhance disaster risk preparedness for downstream communities
dependent on glacier-fed rivers.



*Data availability*
All datasets utilized in this study are publicly available and accessible through open-source
platforms.
*Author contributions.*
Conceptualization: AI. DV. Methodology: AI, DV. Formal analysis: AI. Investigation: AI.
Data curation: AI. Writing– original draft: AI. Writing– review and editing: all authors.
Visualization: AI, DV. Supervision: DV, AN.
*Declaration of competing interest*
The authors declare no competing interest.

*Acknowledgments*
We acknowledge the European Space Agency (ESA), the United States Geological Survey
(USGS), National Remote Sensing Center, WorldClim, and ECMWF for the provision of their
open-access data to the scientific community. We are also thankful for the availability of
research products such as the Randolph Glacier Inventory (RGI), Global Ice Thickness dataset,
and ITS_LIVE velocity dataset, which contributed to this study.

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
