# Peer review of "Understanding the Gangotri glacier dynamics: Implications from a fully distributed inversion of equivalent water-volume change"

_EGUsphere, 2025_

## Author Comment (AC1)

**RC1: 'Comment on egusphere-2025-1614', Anonymous Referee #1, 07 Jul 2025**

This manuscript uses a diverse compilation of remote sensing data sets to investigate the recent evolution of Gangotri Glacier in the Himalaya. A special focus is laid on volume changes and implied annual mass balances. The authors used optical satellite images for updating available glacier boundaries. Glacier surface velocities were determined by feature tracking, using different software solutions, from optical and NIR satellite imagery, while published digital elevation models, also from remote sensing sources, were used for glacier geometry and the long term mass change analysis. Additional data sets were used as a control of the analysis, as well as for relating the detected glacier changes to environmental conditions. Inferring the ice thickness across the glacier by different ice dynamic methods, basically using the shallow ice approximation, represents the core of the manuscript. Annual ice velocity fields were determined for this purpose and combined with additional information, like glacier surface slope, ice temperature and assumptions about geometrical constraints and basal sliding.

The basic results agree rather well with existing global data sets, demonstrating the correct application of the widely used algorithms for surface velocity determination and ice thicknesses. This is a comprehensive study of combining remote sensing data for inferring knowledge about the status of one of the best-studied glaciers in the Indian Himalaya. The data sets provide the opportunity to analyse the decadal changes of the glacier since 2000, which however is not fully exploited. Even though the manuscript presents rather interesting data about ice velocities and ice volume change, there are several problematic issues, which require a critical evaluation. The authors calculate ice thicknesses on an annual basis for determining annual thickness changes, due to the lack of available annual elevation models. Unfortunately, these results are used in a rather uncritical approach for discussing mass balance and climate related reasons for such thickness changes, even though the volume variations are not significant with respect to the error calculations provided. In addition, the error estimates only cover the uncertainties originating from the inherent data characteristics, but not from the glacier conditions in space and time. Surface velocities can be determined to a satisfying degree of accuracy in the ablation region of glaciers, but it becomes difficult in the accumulation region, where suitable features are rare and contrast is very low. The manuscript completely lacks a discussion about the quality of surface velocities across the glacier. The shape factor is assumed to be constant for the entire glacier, which is clearly not applicable in such a complicated environment. The assumption of the basal sliding component is a rather crude estimate based on a general assessment of Himalayan glaciers, while seasonal variations from the surface velocity analysis were not utilised for a more site specific approximation. The slope of the glacier was considered constant for the analysis period, based on the DEM from 2018 and there is no information about the spatial choice for slope calculations, which is crucial for velocity determinations. Even though it can be assumed that the surface slope is not changing to a large degree within the seven years, this implies an additional error for the final results. In the end a critical assessment of the determined volume variations and their limitations is not provided, especially about the validity in slow moving regions like the small, high elevation accumulation basins with poor velocity information. The small variations in glacier volume between 2018 and 2023 are not significant at all, while potential reasons for the large deviations, especially in 2017, are not discussed in this framework. It is surprising that the DEM from 2018 is not used to evaluate the volume changes from 2015 to 2018 in comparison with the volume changes from 2016 to 2018 derived from the velocity analysis. This could provide a general idea about the validity of the analysis.

We sincerely thank the reviewer for the detailed evaluation of our manuscript. We thank the reviewer for pointing out the major issues and for the suggestions in improvement of this manuscript. Below, we provide a detailed clarification and justification of the key points raised, the corresponding components from the revised manuscript are illustrated in response to the specific comments. The responses and

revisions are mentioned in different colors.

- The reviewer has aptly pointed out that Gangotri glacier being one of the highly studied glaciers
  in the literature. However, we would like the reviewer to note that no studies to the best of our
  knowledge have supplemented investigations with in-situ measurements of ice-thickness and
  surface velocities. Moreover, very few to no studies report the in-situ measured mass balance.
- We agree that the available datasets provide the opportunities to investigate from 2000. However, there is remarkable difference in the spatial resolutions of the available datasets. The earlier datasets of Landsat mission do not specifically allow to capture detailed seed points (features) for surface velocity estimation when compared with high-resolution Sentinel-2 datasets. Moreover, accounting the uncertainties from multi-sensor velocities in the framework would be highly challenging and hence, like several other studies, we opted to restrict our investigation from the earliest availability of Senitnel-2 datasets (Li et al., 2024; Nagy et al., 2019; Sinha et al., 2024; Zhang et al., 2023). Moreover, our present focus was to establish a physically consistent, fully distributed inversion of water-equivalent volume change, which required consistent surface velocity over multiple years. The methods followed are completely based on remote sensing data. Hence, we have revised the title of the manuscript (as also pointed out by Reviewer-2) to "Understanding the Gangotri glacier: Implications from a remote sensing based fully distributed inversion of equivalent water-volume change."
- Because of the lack of temporally consistent annual DEMs for the full period, we performed the
  inversion on an annual basis using the best-available elevation and velocity composites.
  Further, it must also be noted that the freely available DEMs also contain errors of the order of
  a few meters that typically propagate in geodetic modelling of ice-thickness changes, yet the
  approach is widely popular, considering the significant lack of available information, especially
  in ungauged glaciated basins (Bandyopadhyay et al., 2019; Bhattacharjee & Garg, 2024;
  Halder et al., 2025).
- We acknowledge the concern on the discussion of mass balance and climate relations. Kindly
  note that other studies have observed similar uncertainties in ice-thickness estimation and have
  followed a similar approach for the discussion of mass balance changes corresponding to
  climate change, likely due to the lack of detailed long-term in-situ observations available in the
  Indian Himalayas (Bhattacharjee & Garg, 2024; Bhushan et al., 2017; Yang et al., 2024).
- For the uncertainty estimation, we followed the standard approaches in the literature (Bhattacharjee & Garg, 2024; Bhushan et al., 2024; Gantayat et al., 2014; Nela et al., 2023; Xu et al., 2021). These approaches are popularly followed for the Himalayan glaciers due to their largely ungauged nature. Hence, exact field conditions in a consistent manner are largely unknown, especially for the Gangotri glacier which is a highly sensitive region with limited and restricted access. Subsequently, we have now revised the limitation subsection 6.5 discussing the propagated uncertainty in both spatial and temporal contexts, incorporating error contributions not only from input data (e.g., DEM, velocity) but also from glacier geometry, basal topography, and model assumptions (inversion parameters).
- We fully agree that the reliability of surface velocity retrieval varies spatially. We have revised
  the manuscript discussing this explicitly, indicating that velocity accuracy is highest in the
  ablation zone, where well-defined supraglacial features permit precise tracking, while
  uncertainties increase in the accumulation area due to limited contrast and limitations of the
  feature tracking algorithm.
- Concerning the shape factor and basal sliding component, we recognize the importance of spatial variability in these parameters. In the revised manuscript, we clarify that the shape factor (f) was initially assumed constant following previous studies for Himalayan valley glaciers (e.g., Gantayat et al., 2014; Haeberli & Hoelzle, 1995). However, to better represent the varying lateral drag and cross-sectional geometry along the Gangotri Glacier, we have varied f between

0.7 and 0.9. The same approach has also been followed by (Van Wyk De Vries et al., 2022), during their investigation of the glacier thickness and ice volume in the Andes, and by (Sinha et al., 2024) in their investigation on Chhota Shigri glacier, in Himachal Pradesh, India.

- The basal sliding factor (β) in previous studies has been applied as a constant average value (Bhattacharjee & Garg, 2024; Bhushan et al., 2017; Nela et al., 2023, 2023). Similarly, following Van Wyk De Vries et al., (2022) and Sinha et al., (2024), we have adopted a variable range of 0 to 0.4, which better represents the spatial and thermal heterogeneity of the polythermal, temperate-type glacier for estimating the ice thickness by inversion modelling approach. This adjustment accounts for variations in basal hydrological conditions and thermal regime, where higher sliding values are likely in the lower ablation region and minimal sliding in the frozen-bed accumulation zones. These refinements have been integrated into the inversion model to improve the physical consistency of ice-thickness estimation and reduce systematic bias associated with uniform parameter assumptions. We have now also performed a sensitivity analysis using a spatially varying shape factor derived from the glacier width–depth relationship.
- For the surface slope, we clarify that it was derived from the 2018 DEM at 30 m resolution, computed over 3x3 pixel moving windows to minimize local noise. The slope was assumed stable on account of non-availability of DEMs after 2018. Hence, the uncertainty propagated towards the thickness and volume calculation by propagation error principle was also assumed to remain same over the short 2016–2023 period as ±8.7% to the overall error budget.
- Regarding the slow moving regions like the small, high elevation accumulation basins with poor velocity information, we have updated the limitations section to discuss these at appropriate length.
- Regarding the small variations in glacier volume, a change of 1km³ is considerably large when observed in terms of water resource. The total ~glacier volume of the Uttarakhand state in India is 213.74 km³ (Dobhal & Pratap, 2015), comparing this with Gangotri glacier as 19.7 km³, this is approximately 9.2%. We believe changes of the order of 5% (1km³) of this equivalent volume is significant.
- The DEM selections were based on the latest availability and as per the technicalities discussed
  later in the specific comments. The geodetic approach was implemented using Copernicus
  DEM and SRTM DEM (both technically radar based DSMs) and not with Cartosat DEM (which
  is actually a DEM based on stereo-optical imagery).

In the following, I provide more detailed remarks about the sections 1-5. The further sections probably require a serious reconsideration, after implementing a realistic assessment of the methods and results.

**Specific remarks:**

**1. L. 42/43: Himalayas vs HKH, this is not the same.**

We strongly agree with the reviewer's observation and appreciate the clarification. The terms "Himalayas" and "Hindu Kush Himalaya (HKH)" indeed refer to different geographical extents and should not be used interchangeably. In the revised manuscript, we have carefully rewritten the introductory section to ensure terminological precision. The revised text now specifies the Hindu Kush Himalaya (HKH) as the focus region, emphasizing its broader geographical coverage that extends beyond the Himalayan arc to include adjacent mountain systems such as the Hindu Kush and Karakoram ranges.

**2. L. 47: the three basins cover the entire region**

We appreciate the reviewer's valuable observation and agree that the earlier phrasing could imply that the Indus, Ganges, and Brahmaputra basins encompass the entire Hindu Kush Himalaya (HKH) region, which is not strictly accurate. In the revised manuscript, we have clarified this statement to reflect the

correct spatial relationship. We have updated the sentence as follows.

"The HKH region is the source of 10 major river basins — the Indus, Ganges, Brahmaputra, Irrawaddy, Salween, Mekong, Yangtze, Yellow, Amu Darya, and Tarim — making it one of the most important water resource of the world. Among these, the Indus, Ganga, and Brahmaputra basins together contain the majority of the region's glacierized area (Mukherji et al., 2015; H. Singh et al., 2025)."

3. L. 51: what does glaciers below 5700 m mean? The entire glacier below this elevation, or only the terminus?

Thank you for your observation. The revised sentence reads as

"Glaciers that are entirely situated below 5700 m elevation are mostly sensitive to climate change, particularly those glaciers that are not covered by thick debris cover and are directly exposed to atmospheric conditions (Bajracharya et al., 2015)"

**As compared to previous**

"The glaciers below 5700m elevation are particularly sensitive to climate change, particularly when they are not covered by thick debris and are directly exposed (Bajracharya et al., 2015)."

4. L. 54/55: This statement is not supported by the references provided. Prasad et al (2009) only considers a small sample of glaciers and Bhambri & Bolch (2009) state that the length changes are highly heterogeneous, but the data basis does not provide evidence for centennial changes of the majority of the glaciers. Indeed, the largest glacier class is the class of small glaciers, which would already have disappeared with the cited retreat rates.

We thank the reviewer for this insightful comment. We agree that the previous version of the statement lacked adequate supporting evidence and that the cited references did not fully substantiate the claim of region-wide centennial-scale glacier retreat. In the revised manuscript, we have carefully updated and expanded the reference list to include more robust and representative studies that demonstrate spatially variable but regionally retreat patterns across the Himalayan glaciers. The revised statements read as follows:

"Ongoing climate change has led to widespread glacier retreat across the Himalayas in recent decades, although the rates and patterns vary significantly between regions and individual glaciers (Kulkarni & Karyakarte, 2014; Bhambri & Bolch, 2009; A. K. Prasad et al., 2009; Dobhal et al., 2004)."

5. L. 56: What does that mean "ongoing melting and stagnation"? Stagnant glaciers, very likely do not experience exceptional melt.

Thank you for your observation. We agree that the original phrasing was confusing. Our intention was to differentiate the glaciers experiencing accelerated melting. We have revised the sentence accordingly to clarify this distinction and improve accuracy.

"The retreat of Himalayan glaciers and a high and consistent melting rate may contribute significantly to proglacial lake volumes, particularly at glacier terminus posing a critical risk of glacial lake outburst floods (GLOFs) and other mountain hazards (Bhambri et al., 2020)."

6. L. 59: This sentence has no connection to the text above. What do you want to express with this statement?

We agree with the reviewer's comment, and the unrelated sentence has been removed to improve the coherence of the introduction.

7. L. 83/84: Is the inversion of ice thickness distribution a major aim of the paper? It is one application of glacier surface velocities among others, which then also could be named. Otherwise

elaborate, why you specifically mention thickness inversion and connect it to the following paragraph.

We thank the reviewer for pointing this out. The ice thickness inversion is indeed a significant process in estimating the glacier volume. We have revised the statement, "The precise estimation of glacier velocity and at better spatial resolution is critical in the inversion of glacier ice-thickness." to as follows.

"The precise estimation of glacier velocity and at better spatial resolution is critical in the inversion of glacier ice-thickness, which is a critical parameter in estimating glacier volume (Gantayat et al., 2014; Sinha et al., 2024, Van Wyk De Vries et al., 2022). Glacier volume and ice thickness are critical inputs in glacio-fluvial hydrological modelling and water resource management (Ren et al., 2023; Silwal et al., 2023; J. Singh et al., 2025)."

8. L. 98-107: This reads like a loose compilation of methods and experiments without a clear goal. There is a large difference between global estimates and detailed reconstructions of individual glaciers. The last sentence is out of context. What is the reference time for the reduction of ice volume in High Mountain Asia?

We thank the reviewer for illustrating these issues. We have reorganized and revised the paragraph. As suggested, we have excluded literature focussing on global estimates and focussed on the methods. The revised paragraph reads as follows.

"The collection of in situ observations using intrusive or extrusive methods, such as hot water drilling, seismic or radar measurements, and gravimetry, poses significant challenges on glaciers with rugged terrain and are often impractical for complete glacier surfaces (Murray et al., 2007). Some glaciers are often inaccessible or have restricted access due to their geopolitical sensitivity rendering no potential opportunities for on-site data collection (Ambinakudige, 2010). As a result, remote sensingbased techniques are widely employed. Models using digital elevation model (DEM) data, glacier boundary, and boundary of ice-flow catchments—such as the mass conservation approach by (Farinotti et al., 2009; Huss & Farinotti, 2012)—have become particularly prominent for estimation of glacier ice thickness. Other methods are based on surface slope, velocity and basal shear stress (Bhushan et al., 2017; Gantayat et al., 2014, 2017; Linsbauer et al., 2012; Michel et al., 2013; Zorzut et al., 2020). An extension of the ice-velocity based approach based on SIA (VWDV) was proposed by (Van Wyk De Vries et al., 2022), which showed significant glacier ice-thickness estimates compared with in-situ measurements. Sinha et al., (2024) proposed an ensemble modelling approach to account for overestimation and underestimation of different ice-thickness inversion models. In practical applications, these modelling techniques effectively determine ice thickness for most glaciers, though they tend to exhibit greater uncertainties when applied to small glaciers unlike the Gangotri glacier with gentle topography (Linsbauer et al., 2012; Rabatel et al., 2018)."

9. L. 108-117: This is again a loose collection of methods missing a clear context. The existence of different methods for ice thickness inversion is already mentioned at lines 85 ff. It would be better to prepare a concise paragraph about the different methods and the application of these methods in a structured way.

We thank the reviewer for pointing this out. We have merged the two paragraphs and revised the statements as per the suggestion focusing on the methods.

10. L. 144-152: The general description of the Gangotri Glacier setting is rather unclear. ELA estimation is based on observations at Dokrani Glacier, missing information how this was done. The temperature range of the Gangotri Glacier region is not plausible. There is a seasonal variation and an altitude variation. It is unlikely that there are positive temperatures at 7000 m elevation. What are the given values, daily means, monthly means? The same is true for the humidity level, which rather likely varies considerably during the seasons (dry winter and humid monsoon). The

precipitation magnitudes are very likely far too low, as measurements at high altitudes do not exist. What do the two values represent, seasonal or altitudinal variability? What is the period?

We thank the reviewer for their remarks pointing out these mistakes. We have revised the section as follows, accordingly.

"The Gangotri Glacier is typical valley-type glacier of the Garhwal Himalaya in the upper Bhagirathi catchment (Fig. 1). Situated in Uttarakhand, India, the Gangotri Glacier spans latitudes 30°43′22″N to 30°55′49″N and longitudes 79°4′41″E to 79°16′34″E, with an approximate length of 30 km and an area of 137 km² (as per this study). The glacier's surface elevation varies significantly, ranging from around 3970 m to 7000 m. The major tributaries of Gangotri Glacier are Kirti, Chaturangi, and Raktvarna. Ganga which is the largest river of India flows from the Gangotri glacier. Bhushan et al., 2017 reported the equilibrium line altitude (ELA) for the Gangotri Glacier and its tributaries to be approximately 5100 m as a first order estimate considered from the observations of nearby Dokriani glacier (Dobhal et al., 2008) in case of inadequate field data (O. King et al., 2017). Climatically, the Gangotri Glacier region exhibits significant seasonal variability. The melting period of the glacier is May to October (Dobhal et al., 2013). The near Gangotri glacier region exhibits precipitation ranging from 131-394 mm with summer temperatures varying between 5-15°C and a summer relative humidity of about 68% (P. Singh & Singh, 2001)."

11. L. 161-163: This sentence is not comprehendible.

We thank the reviewer for pointing out the lack of clarity in this sentence. The sentence has been rewritten to improve readability. The revised text now reads as follows:

"To derive the glacier surface slope, we used the latest Version 3 Cartosat-1 digital elevation model (DEM) from 2018, with a 30-meter spatial resolution. This DEM product provides relatively better elevation accuracy, making it suitable for glacier slope and ice thickness modelling during the 2016–2023 study period (Talchabhadel et al., 2021)."

12. L. 166: This is not the vertical accuracy; it is the mean vertical error which has a rather large RMSE of almost 15 m. Therefore, it depends very much on the individual case, how well the true elevation is represented.

We thank the reviewer for noting and reporting this error. We have revised the sentence as follows.

"The vertical accuracy of SRTM DEM varies significantly depending on terrain. Rodríguez et al., (2006) reported a general vertical error of less than 9 meters for the SRTM DEM in mountainous regions. Kolecka & Kozak, (2014) observed a mean vertical error of 4.31 meters, and root mean square error (RMSE) of ±14.09 meters in mountainous terrain."

13. L. 168/169 This is not comprehendible.

We thank the reviewer for highlighting this issue. The sentence has now been revised for clarity and rewritten to ensure that the intended meaning is clearly conveyed. This reads as follows now.

"The Copernicus DEM which is based on the TanDEM-X mission data, with a spatial resolution of 30 meters and an absolute vertical accuracy better than 4 meters (90% linear error) was used for year 2015 (European Space Agency & Airbus, 2022). The Copernicus DEM was used in combination with the SRTM DEM (data year 2000) to assess surface elevation changes between 2000-2015 over the Gangotri Glacier after penetration depth correction based on Landsat-7 satellite images with normalized difference snow index (NDSI) (Millan et al., 2015; Purinton & Bookhagen, 2018; Carturan et al., 2020; Guillet & Bolch, 2023). The selection of the Copernicus DEM and the SRTM DEM stems from the fact that both these are technically DSMs generated from radar remote sensing data, as compared to the Cartosat DEM which is generated from stereo-optical imagery."

14. L. 170/171: What is the reason for using the Cartosat DEM for the slope, but the Copernicus DEM for the DEM differencing?

We thank the reviewer for their suggestion. It is worth noting that Cartosat DEM is generated from stereo optical imagery and the Copernicus DEM is based on X-band radar, while the SRTM DEM is generated from C-band radar data. Technically, the radar-based terrain products are DSMs compared to the Cartosat product, which is a DEM. Below is the difference of the Cartosat DEM (2018) and Copernicus DEM (2015) for reference, which shows unrealistic changes. The following sentence is added to the datasets section.

"The selection of the Copernicus DEM and the SRTM DEM stems from the fact that both these are technically DSMs generated from radar remote sensing data, as compared to the Cartosat DEM which is generated from stereo-optical imagery."

Figure R1. Elevation change between 2015 and 2018 (Copernicus DEM and Cartosat DEM)

**15. L. 171-174: Why did you have to adapt the glacier boundaries?**

We appreciate the reviewer's comment. The glacier boundaries from the RGI 6.0 didn't match exactly and hence, the boundaries had to be adapted from high resolution imagery. The glacier boundary description has been revised for clarity as follows.

"The Gangotri glacier boundary was derived from the RGI 6.0 dataset (RGI Consortium, 2017) and subsequently refined using high-resolution satellite imagery in Google Earth Pro. On-screen digitization during snow-free periods allowed for accurate updates of the terminus and lateral boundaries, especially where RGI 6.0 outlines were inconsistent compared with recent imagery."

16. Table 1: The table presents more data sets as are described in the data section. What were the missing data used for?

We thank the reviewer for pointing out these issues. We have revised, reorganized and added in the dataset section the ancillary data used for climatic variability trend analysis. We also split the paragraph now as it was too long. The revised paragraphs with data descriptions now reads as follows.

"The Gangotri glacier boundary was derived from the RGI 6.0 dataset (RGI Consortium, 2017) and subsequently refined using high-resolution satellite imagery in Google Earth Pro. On-screen digitization during snow-free periods allowed for accurate updates of the terminus and lateral boundaries, especially where RGI 6.0 outlines were inconsistent compared with recent imagery. ITS LIVE velocity data (Gardner et al., 2022), with a resolution of 120 meters, was employed to enable a comparative assessment with the ice velocity datasets derived in this study. This comparison provides an independent reference to evaluate the consistency of the estimated velocities, helping to identify potential deviations, while the widely used ice thickness dataset by Farinotti et al., (2019), with a 50-meter resolution, was used for comparative analysis with thickness model outputs.

To assess the climatic and environmental variability influencing the Gangotri Glacier, multiple datasets were utilized. MODIS (Yu et al., 2022) Land Surface Temperature (LST) data (MOD11A1) from 2016 to 2023, at a 500-meter resolution from Aqua and Terra satellites, was applied to derive ice surface temperature (IST). The MODIS MCD43A3 dataset provided surface albedo estimates essential for analysing the apparent thermal inertia (ATI) of the glacier ice surface. TerraClimate data was employed to examine maximum temperature, precipitation, and runoff trends across the glacier basin. ERA5-Land reanalysis data supplied estimates of snowfall parameter. The MODIS MCD19A2 product was used for aerosol optical depth (AOD) analysis, contributing to the assessment of radiative forcing impacts. Further, the Global Annual Weighted PM2.5 dataset (GWRPM25) was used to evaluate long-term atmospheric pollution trends over the region and their potential effects on glacier melt dynamics. The MODIS products have been widely used in the literature for climatic investigations in mountainous regions (Hassan et al., 2023; Thanveer et al., 2024; Varade et al., 2023; Varade & Dikshit, 2020). Table 1. summarizes the various datasets used in this study for modelling, comparative assessment and complementary analysis."

Table 1 has been revised as follows.

Table 1. Satellite and other ancillary data used in this study.

[revised manuscript text omitted]

|           |             |           | estimation      |                             |  |
|-----------|-------------|-----------|-----------------|-----------------------------|--|
|           |             |           | Aerosol Optical | (Lyanuatia & Wang           |  |
| MCD19A2   | 2000 - 2024 | 500 m     | Depth (AOD)     | (Lyapustin & Wang,
2018) |  |
|           |             |           | estimation      |                             |  |
| GWRPM25   | 1998 - 2021 | 1.13 km   | PM 2.5 trend    | (Van Donkelaar et           |  |
| GWRPIVI25 | 1996 - 2021 | 1.13 KIII | analysis        | al., 2021)                  |  |

**17. L. 188: What does "basal shear stress of glacier surface" mean?**

Thank you for pointing out the mistake. The phrase "basal shear stress of glacier surface" was indeed misleading. We have revised the sentence as follows.

"The framework for the estimation of the ice thinning rate, ice volume, mass balance and equivalent water volume change of the Gangotri Glacier involves integrating multiple datasets and methods using glacier surface topography, ice movement, shape, elevation change, and the basal shear stress, as depicted in Fig. 2."

18. Section 3.1 is rather cumbersome to read, as you predominantly write about the future steps (GIV analysis) instead of presenting details about the pre-processing of the images and the techniques used for glacier delineation.

We thank the reviewer for this valuable observation. Section 3.1 has been merged with Section 3.2 considering the objective velocity estimation from satellite images. Section 3.2 (new) has been revised to present a detailed description of the techniques used for glacier delineation and velocity estimation.

**"3.1. Glacier ice velocity estimation**

A total of 152 Sentinel-2 RGB images, spanning the years 2016 to 2023, were used for Glacier Image Velocimetry (GIV)-based velocity estimation (Van Wyk De Vries & Wickert, 2021). The RGB images were processed in Google Earth Engine (GEE) with a cloud-masking filter to enhance feature-tracking accuracy. Additionally, for comparison single-band near infrared (NIR) grayscale images, free of clouds and snow were utilized for velocity estimation using the Co-registration of Optically Sensed Images and Correlation (COSI-Corr) tool (Leprince et al., 2007) and Image georectification and feature tracking toolbox (ImGRAFT) with time-series image pairs from 2016 to 2023. Sentinel-2 NIR single-band images for the snow-free months of July to October, were downloaded from the Sentinel Data Hub for velocity estimation using Cosi-Corr and ImGRAFT tools. The three methods used for glacier surface velocity estimation rely on tracking persistent surface features between multi-temporal satellite images using different correlation algorithms and windowing strategies, were employed to cross-validate results and assess the consistency and reliability of velocity estimations derived from different algorithms. The resulting velocity maps were carefully refined by applying a glacier mask based on the manually derived glacier outline based on high resolution imagery and RGI 6.0 glacier boundaries.

Sentinel-2 RGB images were used for a pair-wise velocity estimation based on the GIV tool using multi pass feature tracking frequency domain image correlator (Van Wyk De Vries & Wickert, 2021). Image pairs with temporal intervals ranging from 9 days to 1 year were used to derive annual surface displacement and velocity fields of the glacier. Total pairs were generated for this glacier with all years 151 for this study covering an 8-year time period (2016 - 2023). For each Sentinel-2 RGB image pair, displacements were estimated iteratively using a multipass template matching following the standard GIV workflow. The GIV approach follows an iterative multi-pass approach in which displacement is first estimated using larger window sizes, followed by smaller windows in subsequent passes to refine the estimated velocity. We used the standard GIV reference window sizes of 400 m, 200 m, and 100 m and a 50% window overlap, resulting in a final glacier surface velocity map with a spatial resolution of 30 m (Van Wyk De Vries & Wickert, 2021). The overlap refers to the percentage of each template window that overlaps with adjacent windows during feature matching in the Glacier Image Velocimetry (GIV) process, where a 50% overlap is used to significantly reduces noise in the resulting velocity field and ensure consistency. Signal to noise ratio lower than 5 and peak ratio less than 1.3 were considered

during multi-pass template matching with sub-pixel estimator (a correlation refinement technique used to improve the accuracy of displacement measurements beyond the spatial resolution of the input imagery) (Van Wyk De Vries & Wickert, 2021). To enhance the quality of estimated displacements, three pre-processing filters were applied including, the Contrast Limited Adaptive Histogram Equalization (CLAHE), a high-pass filter, and the Near-Anisotropic Orientation (NAO) Filter (Van Wyk De Vries & Wickert, 2021). The CLAHE filter was applied to the satellite images to enhance local contrast in the images aiding in feature tracking. The high pass Sobel filter was applied to the images for improved identification of features for tracking, and the NAO filter was applied to the resulting displacement map. Further, to remove outliers arising from mismatches or noisy correlations in feature tracking, an upper velocity threshold of 200 m a-1 was applied to the resulting displacement map and thresholded pixels were interpolated. The threshold was selected based on prior studies in the region and is consistent with expected velocity ranges for Gangotri glacier (Bhushan et al., 2017b; Gantayat et al., 2014). The resultant displacement map represents the annual velocity maps with velocity values expressed in meters per annum (ma-1).

Through the Cosi-Corr, a single-pass feature-tracking approach was implemented to evaluate glacier surface displacement using Sentinel-2 NIR band imagery, primarily acquired during July-October (2016 to 2023) to ensure minimal cloud cover over the Gangotri Glacier (Table S1). The surface displacement was estimated for consecutive study intervals (2016-2017, 2017-2018, ..., 2022-2023)(Leprince et al., 2007). The COSI correlation module provides two correlation algorithms frequency-based and statistical approach. This study used the frequency method, which is better for reliable results using an optical dataset (Bhushan et al., 2017). The correlation window, commencing with an initial window dimension of 64 x 64, progressively decreases to a final size of 32 x 32, employing a step size of 2, as specified by Bhushan et al., 2017. First, the displacement output is derived at resolution at 160 meters, containing displacements in the east-west (EWD) and north-south (NSD) direction, as well as the associated signal-to-noise ratio (SNR). Pixels with low correlation are eliminated by applying a limit value of SNR

23. L. 244/245: In lines 2010/2011 you mention that you use the NIR band images from October and November. Here you state that you use the images from July-October?

We thank the reviewer for identifying this inconsistency. The correct period is July to October, and this has been corrected in the revised manuscript.

24. L. 251-264: It seems a bit strange that you provide so many details about the COSI-Corr algorithm, while ImGraft and GIV is presented rather briefly. It would be much better to shortly describe the core principle of all three algorithms and then provide some details about the specific application to your data set (sample size, iteration steps, overlaps, multi-band samples, etc.) followed by a short description of the resulting products.

We thank the reviewer for pointing this out. We have merged section 3.1 and section 3.2 and updated the methods to provide more clarity and detail as suggested. The revised section has been shown previously.

25. L. 284/295: and the potential bed deformation. The formula in Cuffey and Paterson assume a hard bed.

We thank the reviewer for bringing attention to this point. We acknowledge that the Cuffey & Paterson, (2010) formulation assumes a rigid (hard) bed, and therefore does not explicitly account for bed deformation. In the revised manuscript, we have clarified this limitation in the context of our inversion, noting that the applied equation reflects hard-bed conditions while basal deformation may still occur in certain parts of the glacier. The following changes are made in the revised manuscript.

The original lines from 282-285 have been revised as follows to clarify these changes.

"This approach combines the principles of glacier flow with the shallow-ice approximation to estimate ice thickness under the assumption of a hard, non-deforming bed (Cuffey & Paterson, 2010; Farinotti et al., 2019; Frey et al., 2014; Maussion et al., 2019; Millan et al., 2022; Nela et al., 2023). The SIA method utilizes basal shear stress for defining the glacier motion as compared to the full driving stresses, and assumes that the local stresses inducted are much greater than the stresses arising from the lateral coupling between adjacent columns (Cuffey & Paterson, 2010). Many glaciers in the Himalaya and Andes are characterized by relatively thin ice in their upper reaches and moderate to steep surface slopes. These conditions exhibit the dominance of basal shear stress within the total driving stress, thereby justifying the application of the SIA for estimating ice thickness and glacier dynamics (Schotterer et al., 2003; Thouret et al., 2007). Several studies employed the glacier surface velocity based approach (Farinotti et al., 2017; Gantayat et al., 2014; Van Wyk De Vries & Wickert, 2021), while others utilize the basal shear stress approach for the retrieval of glacier ice thickness (Farinotti et al., 2017, 2019; Haeberli & Hoelzle, 1995; Kumari et al., 2021; Linsbauer et al., 2012). These methods were carefully selected to provide comprehensive estimates of glacier ice thickness by leveraging different glaciological principles."

26. L. 300/301: Equation 2 shows the glacier surface velocity as a combination of the contribution from internal deformation and basal sliding.

We thank the reviewer for pointing out these mistakes. Lines 294-301 now can be read as follows.

"The glacier ice surface velocity u is a result of two component glacier internal deformation  $u_d$  and basal velocity  $u_b$  (Cuffey & Paterson, 2010), as shown in equation 2.

$$u(H) = u_d(H) + u_h \tag{2}$$

where, the internal deformation  $u_d$  and glacier ice surface velocity are functions of the ice-thickness H that are evaluated at the ice surface."

27. Eq. 4: this equation is only correct for beta=0. Therefore, I assume some mistake in the formulation.

We thank the reviewer for pointing out this mistake. Equation 4 has been corrected as follows.

$$(1 - \beta)u(H) = u_d(H) = \frac{2A_c}{n+1}\tau_b^n H$$
 (4)

28. L. 313/314: I am not convinced that you can apply a general estimate for a specific study, as the Gangotri Glacier does not represent a "mean" glacier of the Himalaya. It would be much better to try and discriminate between annual velocities and winter velocities, in order to estimate the magnitude of sliding. In addition, basal sliding is not constant across the entire glacier. Therefore, it very much depends on the purpose of your thickness inversion if a constant value is reasonable.

We agree with reviewers concern that applying a general estimate for basal sliding may not accurately represent the Gangotri Glacier, as it does not reflect the characteristics of an average Himalayan glacier. We acknowledge that basal sliding is variable and not uniform across the glacier surface. However, estimation of spatially distributed basal sliding is yet not possible with remotely sensed data, and following these limitations, several studies have considered various assumptions. We have revised

these lines to highlight the different assumptions of  $\beta$  considered in the literature and in our methodology. We have also reorganized and revised the section and it now reads as follows.

**"3.2.1. Velocity-based ice thickness**

Glaciers primarily move as the ice deforms under the force of gravity. This flow occurs through three main mechanisms: internal ice deformation, sliding at the base, and deformation of the subglacial bed (Hambrey & Glasser, 2012). The glacier ice surface velocity u is a result of two component glacier internal deformation  $u_d$  and basal velocity  $u_b$  (sum of basal sliding and subglacial sediment deformation) (Cuffey & Paterson, 2010), as shown in equation 2.

$$u(H) = u_d(H) + u_b \tag{2}$$

where, the internal deformation  $u_d$  and glacier ice surface velocity are functions of the ice-thickness H that are evaluated at the ice surface. The deformation velocity is related to the basal shear stress and the ice thickness as follows (Cuffey & Paterson, 2010).

$$u_d(H) = \frac{2A_c}{n+1} \tau_b^n H \tag{3}$$

where  $A_c$  is the Arehenius creep parameter,  $\tau_b$  represents basal shear stress, n is the Glen's flow exponent (n=3) (Glen, 1958). The glacier surface velocity as shown in equation 2 can then be represented as shown in equation 4 (Glen, 1958; Van Wyk De Vries et al., 2022).

$$(1 - \beta)u(H) = u_d(H) = \frac{2A_c}{n+1}\tau_b^n H$$
 (4)

where,  $\beta$  is the basal sliding correction factor (Van Wyk De Vries et al., 2022). The laminar flow law (C. A. M. King, 1983) accounts for both surface and basal velocities of the glacier. However, determining basal sliding velocity through remote sensing is not feasible.

"Some benchmark studies assumed the subglacial deformation to be negligible and have considered basal sliding to represent the major contributions to the glacier's surface velocity (Farinotti et al., 2009, 2017; Linsbauer et al., 2012). In other studies, the basal velocity was typically considered as one-fourth (25%) of the surface velocity of the glacier (Bhushan et al., 2017; Gantayat et al., 2014; Nela et al., 2023; Remya et al., 2019). In contrast, van Wyk et al., (2022) employed a basal sliding correction factor to account for the fraction of glacier motion corresponding to basal sliding. The basal shear stress,  $\tau_b$ , is expressed in terms of measurable parameters following equation 5 (Frey et al., 2014; Linsbauer et al., 2012; Van Wyk De Vries et al., 2022).

$$\tau_b = f \rho_i g H \sin(\alpha) \tag{5}$$

where f is the shape factor (Gantayat et al., 2014; Haeberli & Hoelzle, 1995),  $\rho_i$  is the snow/firn/ice density, g is acceleration due to gravity,  $\alpha$  is the ice-surface slope angle (derived from the DEM) and H is ice-thickness. We follow the velocity-based ice thickness estimation by Van Wyk De Vries et al., (2022), as shown in equation 6.

$$H = \left(\frac{n+1}{2(f\rho_i g)^n A_c^* \exp\left(\frac{Q_c}{R} \left[\frac{1}{T} - \frac{1}{T^*}\right]\right)}\right)^{\frac{1}{n+1}} \left(\frac{u(H)(1-\beta)}{\sin(\alpha)^n}\right)^{\frac{1}{n+1}}$$
(6)

where the Arrhenius creep constant  $A_c$  as defined in Cuffey & Paterson, (2010), was determined using glacier ice surface temperature based on the following temperature-dependent relation.

$$A_c = A_c^* \exp\left(\frac{Q_c}{R} \left[ \frac{1}{T} - \frac{1}{T^*} \right] \right) \tag{7}$$

where the constants being  $A_c^*=2.4\cdot 10^{-24}$ ,  $Q_c=115~{\rm kJ~mol^{-1}}$ ,  $R\approx 0.0083145$  (the ideal gas constant), and  $T^*=273~{\rm K}$  (Cuffey & Paterson, 2010). To estimate the ice surface temperature T of the Gangotri Glacier, MODIS MOD11A1 LST data was utilized. Further details regarding the temperature extraction and processing are provided in Table S2 and S3, and Fig S1.

The glacier ice thickness inversion as described in equation 6, involves several parameters typically assumed as constant such as g,  $\beta$ ,  $\rho_i$ , f,  $A_c$  and n. In equation 6, the spatially varying components include  $\alpha$ , u(H), and  $A_c$ . The parameter  $A_c$  varies spatially corresponding to the ice surface temperature determined by MODIS LST data,  $\alpha$  as per the DEM and the surface velocity u(H) as defined in the previous sub-section. Recent studies (Sinha et al., 2024; Van Wyk De Vries et al., 2022; Yang et al., 2024) as compared to previous literature (Bhattacharjee & Garg, 2024; Frey et al., 2014; Nela et al., 2023; Ramsankaran et al., 2018) vary the constant parameters based on a Monte Carlo simulation for improving the ice-thickness estimates. Following these recent studies, to account for the spatial and thermal variability of the Gangotri Glacier, which exhibits polythermal and temperate characteristics, the basal sliding correction factor  $\beta$  was varied from 0 to 0.4 in the inversion simulations as per previous studies in a Monte Carlo simulation framework. This approach provides a more realistic representation of basal motion compared to the conventional assumption of a constant sliding coefficient. the shape factor f was varied between 0.7 and 0.9 instead of being held constant, to account for the influence of valley geometry and lateral drag effects across different sections of the glacier (ablation and accumulation) (Gantayat et al., 2014; Linsbauer et al., 2012; Sinha et al., 2024; Van Wyk De Vries et al., 2022). This range reflects the realistic variability between narrow, deep valley segments and wider glacier, thereby improving the physical representation of ice flow in the inversion modelling. The snow-firn-ice density was varied between 850 and 917 kg m-3 to account for glacier surface (Sinha et al., 2024; Van Wyk De Vries et al., 2022; Yang et al., 2024)."

**29. L. 314/315: How do you discriminate between driving stress and basal shear stress, based on your available data?**

Thank you for the pointing out on this concern. Due to the limitations of remote sensing data and techniques, no information is derivable on the driving stresses and the basal shear stress can also only be approximated as consistently done in the literature. Lines 311-312 in the original submission state as follows.

"However, determining basal sliding velocity through remote sensing is not feasible."

**30. L. 316/317: This sentence is not comprehendible. Where do you get the ice temperature from?**

We thank the reviewer for pointing out the issues here. The ice surface temperature is approximated from the land surface temperature acquired from the MODIS MOD11A1 product. The sentence has been revised as follows.

"We follow the velocity-based ice thickness estimation by Van Wyk De Vries et al., (2022), as shown in equation 6.

$$H = \left(\frac{n+1}{2(f\rho_i g)^n A_c^* \exp\left(\frac{Q_c}{R} \left[\frac{1}{T} - \frac{1}{T^*}\right]\right)}\right)^{\frac{1}{n+1}} \left(\frac{u(H)(1-\beta)}{\sin(\alpha)^n}\right)^{\frac{1}{n+1}}$$
(6)

where the Arrhenius creep constant  $A_c$  as defined in Cuffey & Paterson, (2010), was determined using glacier ice surface temperature based on the following temperature-dependent relation.

$$A_c = A_c^* \exp\left(\frac{Q_c}{R} \left[ \frac{1}{T} - \frac{1}{T^*} \right] \right) \tag{7}$$

where the constants being  $A_c^* = 2.4 \cdot 10^{-24}$ ,  $Q_c = 115 \text{ kJ mol}^{-1}$ ,  $R \approx 0.0083145$  (the ideal gas constant), and  $T^* = 273 \text{ K}$  (Cuffey & Paterson, 2010). To estimate the ice surface temperature T of the Gangotri Glacier, MODIS MOD11A1 LST data was utilized. Further details regarding the temperature extraction and processing are provided in Table S2 and S3, and Fig S1."

**31. Eq. 6: f is not "measureable" and you do not have information about H.**

We agree with the concerns raised. The shape factor f varies per glacier shape and geometry. For the valley type glaciers such as the Gangotri glacier, we varied the f from 0.7 to 0.9 in the Monte Carlo Simulations as specified by van Wyk De Vries et al., 2022. We have revised the section and added a discussion to highlight these simulations under varying ranges of commonly considered constant parameters as per the literature. The discussed is added below again for reference.

"The glacier ice thickness inversion as described in equation 6, involves several parameters typically assumed as constant such as g,  $\beta$ ,  $\rho_i$ , f,  $A_c$  and n. In equation 6, the spatially varying components include  $\alpha$ , u(H), and  $A_c$ . The parameter  $A_c$  varies spatially corresponding to the ice surface temperature determined by MODIS LST data,  $\alpha$  as per the DEM and the surface velocity u(H) as defined in the previous sub-section. Recent studies (Sinha et al., 2024; Van Wyk De Vries et al., 2022; Yang et al., 2024) as compared to previous literature (Bhattacharjee & Garg, 2024; Frey et al., 2014; Nela et al., 2023; Ramsankaran et al., 2018) vary the constant parameters based on a Monte Carlo simulation for improving the ice-thickness estimates. Following these recent studies, to account for the spatial and thermal variability of the Gangotri Glacier, which exhibits polythermal and temperate characteristics, the basal sliding correction factor  $\beta$  was varied from 0 to 0.4 in the inversion simulations as per previous studies in a Monte Carlo simulation framework. This approach provides a more realistic representation of basal motion compared to the conventional assumption of a constant sliding coefficient. the shape factor f was varied between 0.7 and 0.9 instead of being held constant, to account for the influence of valley geometry and lateral drag effects across different sections of the glacier (ablation and accumulation) (Gantayat et al., 2014; Linsbauer et al., 2012; Sinha et al., 2024; Van Wyk De Vries et al., 2022). This range reflects the realistic variability between narrow, deep valley segments and wider glacier, thereby improving the physical representation of ice flow in the inversion modelling. The snow-firn-ice density was varied between 850 and 917 kg m-3 to account for glacier surface (Sinha et al., 2024; Van Wyk De Vries et al., 2022; Yang et al., 2024)."

**32. L. 326: Why do you use a density of 850 kg/m³ for ice?**

Thank you for pointing this out. Please see the response to the previous comment. The density was varied in the simulations between 850-917 kg/m³ to consider the possible range of values.

33. Eq. 7: Can you explain how you get rid of u\_b? Refer to the formulation of beta in eq. 4.

There was a mistake in equation 4. The corrected equation is as follows.

$$(1 - \beta)u(H) = u_d(H) = \frac{2A_c}{n+1}\tau_b^n H$$
 (4)

This equation can be derived from eqns 2 and 3.

$$u(H) = u_d(H) + u_b \tag{2}$$

$$u_d(H) = u(H) + u_h$$

Now considering that the basal velocity is some fraction of the surface velocity determined by the basal sliding correction factor  $\beta$ , we have as follows.

$$(1 - \beta)u(H) = u_d(H)$$

34. Section 3.3.2: This is a rather crude estimate of ice thicknesses for calculating glacier volumes.

Yes, we agree, the GlabTop approach only uses the approximation of ice-thickness from the DEM and hence, is completely dependent upon the DEM characteristics. Yet, it is widely used in the literature (Linsbauer et al., 2012; Majeed et al., 2021; Pandit & Ramsankaran, 2020; Ramsankaran et al., 2018; Romshoo et al., 2023).

We used this model primarily for comparative analysis with the velocity-based ice thickness estimations (VWDV) in the year of 2016.

35. L. 355: It seems a bit odd that you state a study period of 2000-2015, while the velocity fields are determined for 2016-2023. This is probably not the study period, but the period of thickness change analysis.

We thank the reviewer for pointing out this mistake. The period has been corrected in the revision.

36. L. 364: What do you want to say with this sentence? What is CRS?

We thank the reviewer for noting this ambiguity. CRS referred to Coordinate Reference System. We have revised this sentence to improve clarity and readability. The revised sentences now reads as follows.

"The geodetic method is based on the differencing of bi temporal DEMs, where understandably the two DEMs should be perfectly overlapping, which is ensured through co-registration. Prior to this, however, it is necessary to ensure that both DEMs are correspond to a common geographic or projected system to ensure consistent spatial resolution, alignment, and compatibility for shape and transform. Here, the two DEMs were reprojected to UTM Zone 44N (EPSG: 32644) and then co-registered following the method by Nuth & Kääb, (2011) to correct horizontal and vertical offsets between DEMs during the rectification process, as shown in Equation 10."

37. L. 370: How can a publication from 2000 reference a publication from 2011? Or do you refer to a method, developed in 2000 and used be the authors from in the 2011 publication?

We thank the reviewer for pointing out this ambiguity. We have removed the reference. Nuth and Kaab 2011 adapted the method of Etzelmueller 2000. However, interestingly we found the reference of Etzelmueller 2000 in the list but not in the main article in Nuth and Kaab 2011.

38. Eq. 10: this equation relates to the correction of the DEM during the rectification process. This should be made clear.

We thank the reviewer for their comment. We have revised the sentences and incorporated the suggestions in them. The revised sentences can be read as follows.

"The geodetic method is based on the differencing of bi temporal DEMs, where understandably the two DEMs should be perfectly overlapping, which is ensured through co-registration. Prior to this, however, it is necessary to ensure that both DEMs are correspond to a common geographic or projected system to ensure consistent spatial resolution, alignment, and compatibility for shape and transform. Here, the two DEMs were reprojected to UTM Zone 44N (EPSG: 32644) and then co-registered following the method by Nuth & Kääb, (2011) to correct horizontal and vertical offsets between DEMs during the rectification process, as shown in Equation 10. The same approach has also been followed in Bhushan et al. 2017 for the geodetic mass loss estimation of Gangotri glacier."

39. L. 381: Why do you not discriminate between the penetration depth of snow and ice?

We thank the reviewer for pointing out these concerns. We first discuss the reasons for ignoring penetration depth correction for ice.

It is quoted from Nuth and Kääb 2011 as follows.

"Corrections for depth penetration are hardly used for the SRTM data, and is extremely difficult to correct for as knowledge of the snow conditions at the time of acquisition is required yet hardly available. We do not consider radar wave penetration in this study"

The corrections here are similar to Bhushan et al., 2017 that are approximated based on Kääb et al., 2012. Quoted from Kääb et al., 2012

"Consequently, extrapolation of ICESat-derived glacier elevation trends (Fig. 2) back to the SRTM acquisition date of February 2000 reveals first-order C-band penetration estimates of several metres, largest for firn/snow and smallest for debris-covered ice, and largest for KK and smallest for HP and NB"

Indicating smaller penetration estimates for debris covered ice, as observed predominantly in the Gangotri glacier. Based on this study, Bhushan et al., 2017 used the average C Band penetration values of 2.3 and 1.7 for snow/firn and clean ice, respectively. However, we applied only the corrections for snow/firn, considering the study by Yousuf et al., 2024 that shows dominant distribution of debris and mixed debris compared with clean ice.

Figure R2. Glacier facies classification (Yousuf et al., 2024)

The sentences (380-385 in original submission) in the revision are modified as follows.

"For penetration depth correction of the SRTM DEM, we apply average C Band penetration values of  $2.3 \pm 0.6$  m for snow, as specified by (Kääb et al., 2012) and also employed by Bhushan et

al. 2017 for the penetration correction of the C-band SRTM data. Yousuf et al., (2024) illustrated the glacier facies evolution showing a major dominance of ice-mixed debris and supraglacial debris at Gangotri glacier. According to Kääb et al., 2012, the penetration estimates are the largest for snow/firn and smallest for debris covered ice. Following these deductions, we only applied penetration corrections for snow/firn. To identify the snow/firn areas, we used a Landsat - 7 image from September 2000, with the NDSI calculated using the green and short-wave infrared (SWIR) bands and a threshold value of 0.23."

40. L. 394: Can you specify, why you need a Monte Carlo approach for the mean glacier-wide ice thickness? It this included in your approach according to section 3.3.2?

The Monte Carlo approach has been defined by van Wyk de Vries et al. (2022) to account for the uncertainties in ice thickness inversion and their incorporation in the ensemble (final) ice thickness map. The approach is directly followed from van Wyk de Vries et al. (2022) and is now cited in the revised version as follows, where equation 11 has also been updated.

$$V_i = H.A_a \tag{11}$$

where  $V_i$  is the glacier ice volume, H is Monte-Carlo-derived mean ice thickness, as described by van Wyk de Vries et al. (2022), and  $A_q$  denotes area of the glacier."

41. L. 397: what is the meaning of the mean ice thickness for a single pixel?

We thank the reviewer for this observation. Each pixel represents one grid cell of the glacier, and the ice volume for that cell is computed as ice thickness multiplied by pixel area. Since the thickness is estimated through Monte Carlo simulations, the "mean ice thickness" refers to the average thickness value for that pixel obtained from all simulation runs. The total glacier volume is then calculated by integrating over all pixels using Equation 12. The following changes are made in the revised manuscript.

"The mean ice thickness refers to the pixelwise average ice-thickness from all iterations of the Monte Carlo runs." This sentence has been added after L397-398 of the original submission.

42. L. 404/405: Why did you not utilize the DEM from 2018 for an elevation change analysis? This you enable you to check the validity of your annual elevation changes at least for the period 2016-2018.

We thank the reviewer for their remarks. We have addressed this guery in response to comment no 14.

43. Eq. 13: You cannot use the ice density for the density conversion to water equivalent, because you neglect the effect of the firn body. Either r is the pixel area, or you need to write r2.

We thank the reviewer for pointing out this blunder. We have corrected the equation 13 and in the writeup accordingly, as follows.

$$B = \frac{\Delta V}{S} \times \frac{\rho_{ice}}{\rho_{\text{water}}} = \Delta \overline{H} \times \frac{\rho_{ice}}{\rho_{\text{water}}};$$

$$\Delta V = \sum_{i=1}^{N} r^2 \Delta \overline{h}_i; S = \sum_{i=1}^{N} r^2$$
(13)

where B is given in m.w.e,  $\Delta h_i$  is the mean ice thickness change of the ith pixel,  $\Delta H$  is the mean glacier ice thickness change in meters,  $\rho_{ice}$  is the density of glacier ice,  $\rho_{water}$  is the density of water, r is the pixel resolution in meters,  $S = \sum_{j=1}^{n} r^2$  is the average area of glacier during the study period, N is the number of pixels covering the glacier ice at its maximum extent. The density of ice was considered as 850 kgm-3 following previous studies (Cogley, 2011; Hubbard et al., 2000; Zemp et al.,

2013). A standard water density of 997 kg m-3 was used to convert the derived ice volume changes into m w.e., following commonly adopted glaciological practices (Huss, 2013; Zemp et al., 2013)."

$$B = \frac{1}{N} \sum_{i=1}^{N} \Delta h_{i} \times r^{2} \times \frac{\rho_{ice}}{\rho_{water}} = \frac{\Delta V}{S} \times \frac{\rho_{ice}}{\rho_{water}}$$

$$B = \frac{\rho_{ice}}{\rho_{water}} \times \frac{1}{N} \sum_{i=1}^{N} \Delta h_{i} \times \frac{1}{N} \sum_{i=1}^{N} r^{2} = \frac{\Delta V}{S} \times \frac{\rho_{ice}}{\rho_{water}}$$

$$B = \frac{\rho_{ice}}{\rho_{water}} \times \frac{1}{N} \sum_{i=1}^{N} \Delta h_{i} \times S = \frac{\Delta V}{S} \times \frac{\rho_{ice}}{\rho_{water}}$$

$$B = \frac{\rho_{ice}}{\rho_{water}} \times \frac{1}{N} \sum_{i=1}^{N} \Delta h_{i} = \frac{\rho_{ice}}{\rho_{water}} \times \Delta H$$

44. L. 421: The method is not limited by cloud boundaries, but cloud cover.

We thank the reviewer for notifying this mistake. The sentence has been revised accordingly to accurately reflect the limitation related to cloud cover.

"Compared to microwave imagery, the main drawback of optical image-based feature tracking is that it is limited by cloud cover since it lacks the ability to penetrate clouds."

45. L. 425: what about the accumulation area?

We have revised the sentence as follows.

"In this study, to reduce inaccuracies caused by snow and cloud cover, only images with minimal snow and cloud cover were selected over both the accumulation and ablation areas of the glacier."

46. L. 428: the 25° is the maximum slope of the stable ground? It is not clear how you define the velocity uncertainty in the end.

We thank the reviewer for their remarks. We have revised the paragraph to improve clarity in the methods, as follows.

"In this study, to reduce inaccuracies caused by snow and cloud cover, only images with minimal snow and cloud cover were selected over both the accumulation and ablation areas of the glacier. To estimate the uncertainty of glacier surface velocity using the feature tracking methods GIV, COSI-Corr, and ImGRAFT, we identified stable ground areas (representing no movement) adjacent to the glacier boundaries that are free of both snow and cloud cover (Bhattacharjee & Garg, 2024; Bhushan et al., 2017; H. Singh et al., 2023; Sinha et al., 2024). These areas were selected based on gentle terrain slopes (≤ 20°), which are generally assumed to represent stable, non-glacierized ground with minimal motion. In such zones, the expected displacement should ideally be zero; thus, any detected movement is interpreted as error or noise and is used to quantify the uncertainty in velocity estimates (Bhattacharjee & Garg, 2024; Bhattacharya et al., 2016; Saraswat et al., 2013). The mean values of the modelled surface velocity in these non-glaciated zones is used to represent the uncertainty in the velocity as per different studies (Bhushan et al., 2017; Gantayat et al., 2014; H. Singh et al., 2023; Sinha et al., 2024)."

47. L. 438-442: Several of the values used seem rather crude estimates (e.g. for beta). Some more justifications are required for their choice.

We thank the reviewer for their remarks. We agree that the values are crude estimates, however in the absence of glacier specific measurements; these have been widely used in the literature. We have reinforced these sentences with more references where these values have been used.

The uncertainties for  $A_c$  and  $\beta$  which were not informed in the previous submission have been detailed in the revisions. Equation 14 has also been corrected. The revisions read as follows.

66

$$\frac{dH}{H} = \sqrt{\left(\frac{1}{4}\frac{du(H)}{u(H)}\right)^2 + \left(\frac{3}{4}\frac{df}{f}\right)^2 + \left(\frac{3}{4}\frac{dA_c}{A_c}\right)^2 + \left(\frac{3}{4}\frac{d\rho}{\rho}\right)^2 + \left(\frac{3}{4}\frac{\sin\alpha}{\sin\alpha}\right)^2 + \left(\frac{3}{4}\frac{d\beta}{\beta}\right)^2}$$
(14)

Velocity uncertainty u(H) was estimated from the stable terrain, as described in the previous subsection. The uncertainty values for the different parameters such as df,  $d\rho$ ,  $dsin\alpha$ ,  $dA_c$  and  $d\beta$  were considered based on previous literature. The shape factor was considered by Linsbauer et al. (2012) as 0.8 as an average between 0.7 and 0.9 for ablation and accumulation zones, respectively. They assessed the impact of uncertainty in different parameters where a ±12% uncertainty in the model results was observed for shape factor. The uncertainty in density in the absence of quantitative values of density is typically assumed. We selected the uncertainty in ice density according to Bhattacharjee and Garg (2024), as  $60 \text{kg/m}^{-3}$ . Uncertainties in the slope, i.e.  $sin(\alpha)$  arise primarily due to vertical inaccuracies in the digital elevation model (DEM). In the absence of ground elevation measurements, we rely on reported vertical root mean square (RMS) errors for the Cartosat DEM as approximately ±8.7% (Maanya et al., 2016; Remya et al., 2022; Snehmani et al., 2013). These assumptions have also been made in various other studies on glacier ice thickness and mass balance estimation (Bhattacharjee & Garg, 2024; Maanya et al., 2016; V. Prasad et al., 2019). The uncertainties for parameters  $A_c$  and  $\beta$  have been ignored in previous studies (Guillet & Bolch, 2023; Steidl et al., 2025), where  $\beta$  itself is often ignored (Bhattacharjee & Garg, 2024; Bhushan et al., 2017; Gantayat et al., 2017; Nela et al., 2023; Ramsankaran et al., 2018; Wu et al., 2020). In this study, we approximate the uncertainties in both these parameters using the standard error approach (Lee et al., 2015), where random samples (N) under continuous uniform distribution were extracted from the minimum and maximum values of the two parameters. These values for Ac were derived from minimum and maximum annual LST values and for  $\beta$ , were based on the range specified (0, 0.4). The standard error ( $\epsilon$ ) was computed per law of propagation of error as  $\epsilon = \sigma/\sqrt{N}$ , where  $\sigma$  is the standard deviation of the random samples (Lee et al., 2015)."

48. Fig. 4: It is not clear what it the time basis for the individual sub-plots. I guess that a) is the mean over the 2016-2023 period. The thinning rate in c) is the mean of the annual thickness changes, d) represents the total thickness change between 2000 and 2015, e) is the mean like in c) but for the central flow line, while d) is equal to c) but scaled for w.eq.

We thank the reviewer for their observation. We have revised the Figure captions as follows to improve clarity and readability.

"Figure 4: Spatial and temporal analysis of ice thickness and elevation changes of the Gangotri Glacier. a) Mean ice thickness (in meters) derived using the fully distributed velocity-based (VWDV) inversion method (Van Wyk De Vries et al., 2022), averaged over the period 2016–2023; b) Ice thickness profile along the central flow line of the Gangotri Glacier depicted by transect A–A'; c) Mean annual ice thinning rate (in m a-1), computed as the average of yearly thickness changes across the glacier surface during the study period; d) Thickness (elevation) change (in m) between 2000 and 2015 derived from DEM differencing; e) Thickness (elevation) change from 2000 and 2015 (in m) profile along the central flow line, illustrating declining trends; f) Equivalent water volume loss (in m³ a-1), calculated from the thinning rate shown in (c), representing the annual glacier mass loss in terms of equivalent water volume."

49. L. 523: positive thinning rate relating to thickening is rather unfortunate. Please consider to reformulate.

We thank the reviewer for noting this mistake. The sentence has been corrected as follows.

"In contrast, the left-hand side (LHS) tributaries (Kirti Bhamak, Ghanohim, Sumeru) and right-hand side (RHS) tributaries (Swachanand and Maiand glacier) of the Gangotri glacier exhibited ice accumulation, with rates ranging from 0 to  $1.5 \pm 0.592$  m a-1."

50. L. 527/528: How do you calculate a total mean thinning of -8.39 m, while the mean thinning rate is 0.5 m/a? I would expect that the mean thinning rate is in fact the mean thinning divided by the time.

We thank the reviewer for this comment. Indeed, the mean thinning rate is calculated as the total mean thinning divided by the time period (15 years). The total mean thinning of -8.39 is observed as the mean of the difference of the ice-thickness change by DEM differencing from 2000-2015.

51. Table 3: These results do not reflect variations in the glaciers' mass balance, they just reflect the uncertainty inherent of the approach. The variability is considerably smaller than the error. In column 3 it is not ice volume, but glacier volume, as this volume also contains firn and snow.

We thank the reviewers for their remarks. We agree that the variability and the uncertainty are of similar orders. However, the purpose of the time-series analysis is to highlight the overall trend of ice-thickness and glacier volume which is generally declining. It also indicates the changes of the glacier ice thickness and volume are gradual and significant departures can be better observed from decadal changes. However, the same has its own challenges in harmonizing long-term multi-sensor data and then processing the same through these models.

Notably, the uncertainty we have reported is of similar orders as have been noted of from the literature covering not only the Gangotri glacier but also several other Himalayan glaciers using remote sensing based approaches (Bhattacharjee & Pandey, 2023; Bhattacharya et al., 2016; Bhushan et al., 2017; Gantayat et al., 2014; Nela et al., 2023; Ramsankaran et al., 2018; Thakur et al., 2023). Understandably, there is no replacement for in-situ data, yet in the absence of the same, remote sensing provides crucial information which would otherwise be not available at all.

We have replaced the term "ice volume" with "glacier volume" to correctly indicate that the estimated volume includes snow, firn, and glacier ice in Table 3.

52. L. 536-538: These numbers seem much too small to be detectable. How would you resolve 500 m3/a?

We agree with your remarks. However, this is not a summation over the entire area, but rather a range of pixel values calculated from the mean ice thickness change raster, considering the densities of water, ice, and firn. The mean value of volume wastage is -276.23 m³ w.e.  $a^{-1}$  per pixel. Suppose we have 10 such pixels, then the total volume wastage would be  $(10 \times -276.23) = -2762.3$  m³ w.e.  $a^{-1}$ . However, the total volume wastage derived from the entire raster is -0.403 km³ w.e.  $a^{-1}$  during the study period 2016–2023.

We are grateful for the reviewer for this comment, as it enabled us to track a major blunder here in the uncertainties. These have been revised and updated now. The revised sentences now read as follows.

"The estimated equivalent water volume change is shown in Fig. 4f. Over the main tributaries, the equivalent water volume change ranges from  $\sim 500 \pm 70$  to  $\sim 200 \pm 28$  m³ w.e. a⁻¹, with the upper accumulation area showing a positive mean equivalent water volume change of approximately  $66.44 \pm 9.30$  m³ w.e. a⁻¹. The mean equivalent water volume change for the Gangotri Glacier is approximately  $\sim 276.23 \pm 38.67$  m³ w.e. a⁻¹, while the main trunk exhibits a change of approximately  $\sim 450 \pm 63$  m³ w.e. a⁻¹ (Fig. 4f)."

53. L. 544: This is not in line with your results in Table 3, which rather shows a very variable year to year potential mass balance. Even though, as noted above, this is not a significant result due to the involved errors.

We have revised equation 13 to highlight that the specific mass balance was estimated from the mean ice thickness change in meter (Figure 4c), which was converted to mass balance in m.w.e by scaling with the density ratio of ice and water, which translated to Figure 4f. Figure 5 was derived from Figure 4f by interpolating at 500 m elevation bands.

The mass balance in L544 is derived as the mean of  $\left(\frac{\rho_{ice}}{\rho_{water}} \times \Delta \overline{h}_i\right)$ , where  $\Delta H$  represents the mean glacier ice thickness change in meters. We checked the calculations again from Table 3 it matches the mass balance values.

| Year | Mean ice thickness (m) $ar{h}_i$ | Uncertainty $\sigma_h(m)$ | % Uncertainty $\sigma_h$ (%) | Thickness Change (m) $\Delta \bar{h}_i$ | Specific Mass Balance (m.w.e) $B = \Delta \overline{h}_l \times \frac{\rho_{ice}}{\rho_{water}}$ |
|------|----------------------------------|---------------------------|------------------------------|-----------------------------------------|--------------------------------------------------------------------------------------------------|
| 2016 | 145.45                           | 17.88                     | 12.29288                     |                                         | $ \rho_{ice} \\ = 850  Kg \\ /m^3 $                                                              |
| 2017 | 164.8                            | 24.42                     | 14.81796                     | 19.35                                   | $ \rho_{ice} \\ = 997  Kg \\ /m^3 $                                                              |
| 2018 | 133.31                           | 18.04                     | 13.53237                     | -31.49                                  |                                                                                                  |
| 2019 | 144.32                           | 18.09                     | 12.53465                     | 11.01                                   |                                                                                                  |
| 2020 | 136.75                           | 18.43                     | 13.47715                     | -7.57                                   |                                                                                                  |
| 2021 | 136.44                           | 20.16                     | 14.77573                     | -0.31                                   |                                                                                                  |
| 2022 | 144.28                           | 21.15                     | 14.659                       | 7.84                                    |                                                                                                  |
| 2023 | 136.85                           | 18.39                     | 13.43807                     | -7.43                                   |                                                                                                  |
|      |                                  |                           | 13.69097                     | -1.22857                                | -1.04743                                                                                         |

A recent study by (Halder et al., 2025) reported continued down wasting of  $-1.21 \pm 0.68$  m a-1 during 2020–2024, based on two ASTER DEMs differencing, however they reported an uncertainty of ~60%.

54. Section 6.1.1. This is the wrong discussion. If you compare remote sensing products between each other, it does not tell anything about the reliability of the results, but only about the comparability of the methods. It is common knowledge that velocity inversions are especially affected by errors in the accumulation zone, which is also demonstrated by the strong variability of the individual results in Fig. 6 c. The discussion should therefore focus on the potential reliability of the results for different regions of the glacier. The velocities might represent the real velocities on the glacier tongue reasonably well, while the velocities in the accumulation zone rather likely need to be considered with care. By the way, which periods are compared in this analysis? Cover the ITS-Live date the same period as in your analysis?

We thank the reviewer for this valuable comment. We agree that comparing remote sensing products mainly reflects the comparability of the methods rather than the absolute reliability of the results. In the revised Section 6.1.1, we have focused the discussion on the potential reliability of the derived velocities

in different regions of the glacier. Specifically, our analysis indicates that velocities on the glacier tongue are likely well represented, whereas velocities in the accumulation zone should be interpreted with caution due to higher sensitivity to inversion errors, consistent with the strong variability shown in Fig. 6c. Regarding the comparison periods, we have clarified in the revised manuscript that the ITS-Live velocities cover the same period as our analysis to ensure a meaningful comparison. This has been explicitly stated to avoid ambiguity. The following changes are made in the revised manuscript.

**6.1.1. Comparative assessment with ITS\_LIVE**

This study investigates the ice surface velocity of the Gangotri Glacier using three different feature-tracking approaches: GIV, COSI-Corr, and ImGRAFT. To evaluate the internal consistency of these methods, we compared their outputs with the Inter-mission Time Series of Land Ice Velocity and Elevation (ITS\_LIVE) dataset (Gardner et al., 2022) over a common period (2016–2023). While this comparison does not validate the absolute accuracy of the methods—since all are remote-sensing-based and lack in situ ground-truth data—it provides insights into the relative spatial variability of different approaches across various glacier zones. It is important to note that such inter-comparisons primarily reflect the comparability of methodologies, not their absolute correctness. To ensure a robust assessment, we analysed velocity estimates along the central flow line and in glacier sub-zones (ablation, equilibrium line, accumulation), minimizing lateral variability (Fig. 3e & 6a). Further, recent Sentinel-1-based velocity data (Bhattacharjee & Garg, 2024) and earlier feature-tracking results (Gantayat et al., 2014; Saraswat et al., 2013) were used for contextual comparison. Statistical comparisons were performed using a Least Absolute Residuals (LAR) fit with 95% prediction bounds. The ITS\_LIVE data were resampled from their native 120 m resolution to 30 m to match the resolution of GIV, ImGRAFT, and resampled COSI-Corr (originally 60 m) Fig 6.

Across the whole glacier, ITS\_LIVE velocities showed strong statistical correlations with COSI-Corr (r = 0.857), GIV (r = 0.755), and ImGRAFT (r = 0.755), indicating overall consistency (Fig. 6a & 6b). In the ablation zone, velocity agreement was highest with COSI-Corr (r = 0.872). At the same time, GIV and ImGRAFT exhibited slightly lower correlations (0.738 and 0.753, respectively), likely due to localized effects such as surface melt, debris cover, and ice deformation. In the accumulation zone, while high correlations were also observed (e.g., COSI-Corr = 0.977), these must be interpreted cautiously, as this region is prone to featureless snow cover and frequent image decorrelation. These issues are known to reduce tracking accuracy, and the stronger agreement here may reflect limited feature variability rather than true velocity accuracy.

Figure 6: a) Illustrates the correlation between different glacier velocity estimation approaches for the Gangotri Glacier across various zones; b) Root Mean Square Error (EMSE) of the ice velocity between different approach and c) Depicts the distribution of glacier velocity along the central flow line, extending from the glacier terminus to the accumulation peak (A–A').

In the ablation zone, ITS\_LIVE shows a higher correlation with COSI Corr (0.872) and about similar correlation with GIV (0.738) and ImGRAFT (0.753), reflecting the influence of surface melting, crevassing, and ice deformation, which can introduce uncertainties in velocity retrieval (Fig. 6b). In contrast, the accumulation zone presents the strongest correlations, with ITS\_LIVE aligning closely with COSI Corr (0.977), GIV (0.922), and ImGRAFT (0.964), suggesting more stable ice dynamics in this region due to reduced surface melting and a more consistent ice mass flow (Fig. 6b).

Fig. 3e & 6c represents the glacier velocity distribution along the central flow line from the terminus to the accumulation zone, highlighting distinct spatial variations in ice motion. The velocity remains relatively low near the terminus due to high frictional resistance from bedrock and debris cover, gradually increasing in the ablation zone where ice thinning and gravitational flow enhance movement (Nicholson et al., 2018). In the accumulation zone near the ELA, the velocity reaches its peak, attributed to the increased ice mass and steep surface gradients (Vatsal et al., 2025). The observed variations of the entire glacier velocity reflect the combined effects of topography, ice thickness, and surface conditions, with the ELA emerging as a critical zone for maximum ice movement. The strong correlation suggests that all the remote sensing methods effectively capture glacier dynamics with consistent ice movement along the central flow line. These results demonstrate the compatibility of remote sensing

based feature tracking techniques with well-established satellite datasets. As delineated in Fig. 6a, the ITS\_LIVE velocity captures these patterns of decreasing velocity from accumulation to the ablation zone, followed by ImGRAFT, which also captures similar pattern, however, with higher velocity in some accumulation zones in contrast to ITS\_LIVE. These differences in the velocity patterns are also statistically observed in the higher RMSE for ImGRAFT compared with ITS\_LIVE (Fig. 6b). It is worth mentioning here that for the statistical comparison, the ITS\_LIVE and the COSI-Corr velocities were resampled to 30m from 120m and 60m, respectively. The relatively higher agreement between the COSI-Corr velocity and the ITS\_LIVE product may be attributed to the coarser resolution compared with the GIV and ImGRAFT where the source spatial resolution of the velocity product was 30m. Fig. 6c shows that the GIV and ImGRAFT methods largely underestimate the velocity in some zones of the accumulation region along the central flow line.

While the velocity estimates are broadly consistent across all remote sensing methods, this comparison emphasizes spatial reliability more than validation. The methods appear most robust in the ablation zone, where clear surface features persist throughout the season. In contrast, estimates in the accumulation zone carry higher uncertainty, and results in this region should be interpreted conservatively. These findings highlight the importance of considering glacier zone-specific reliability when applying and comparing remote sensing velocity products."

55. L. 620-622: Why do you use such a weird unit, m/d? Please be consistent.

We have revised accordingly as "32.85 ± 2.25 m a-1 (Bhattacharjee & Garg, 2024)".

56. L. 620-626: Comparing mean glacier wide velocities has no real value, as dynamic changes can happen, even if the glacier wide mean velocity remains almost constant. If you want to compare your results to other investigations, you should focus on specific regions, the main flow line or other representative units.

We thank the reviewer for this observation. We agree that glacier-wide mean velocities can mask important spatial variations in ice dynamics. We have addressed this limitation by shifting the comparison toward spatially representative zones, including the main flowline and key glacier sectors, rather than relying solely on glacier-wide averages. As in-situ measurements are not available for Gangotri Glacier, we continue to compare our remotely sensed velocity fields with previous studies for consistency and validation, but now with emphasis on region-specific patterns rather than mean values. The revised sentences read as follows.

"Due to the harsh climatic conditions in the Gangotri Glacier region, in-situ field-based velocity measurements through stake installations remain challenging. Therefore, we compared our velocity estimates with existing studies based on optical and microwave remote sensing techniques. To ensure meaningful comparison, we focused on specific glacier zones—particularly the snout and ablation areas—where earlier studies have discussed glacier surface velocity quantitatively. In this study, the surface velocity (2019) near the central flowline in the near the terminus zone was estimated to range between  $18 \pm 3.6$  and  $31 \pm 5.78$  m a-1, which aligns well with the findings of (Thakur et al., 2023; Gantayat et al., 2014), who reported velocities between 20 and 30 m a-1 in period of 2019. Similarly, (Saraswat et al., 2013) documented velocities between  $24.8 \pm 2.3$  and  $28.9 \pm 2.3$  m a-1 in the snout region in the period of (2004-2010). A more recent study by Bhattacharjee & Garg, (2024), based on Sentinel-1 offset tracking for 2017–2022, estimated a mean glacier velocity of  $32.85 \pm 2.25$  m a-1, closely matching the average values along the central flow line observed in our results. Our results identified the mean surface velocity along the central flow line as  $30.20 \pm 5.60$  m a-1 for 2017-2022."

57. Section 6.1.2: again this analysis only shows how well the different algorithms agree, but not if this results in a more reliable ice thickness distribution. It would be rather interesting to discuss the influence of the different parameters on the ice thickness distribution, like the unknown ice temperature for different parts of the glacier, or the uncertainties in the slope, especially for steep areas, where remote sensing methods are affected by larger errors.

We agree with the reviewer and show aligning interests. However, detailed and precise investigation of ice-thickness would require long term or at least a full season spatially distributed in-situ data which is highly challenging.

We now tested the influence of key parameters—such as ice density, shape factor, Arrhenius creep factor (temperature-dependent), and surface slope—during the thickness inversion. These parameters were varied within realistic ranges (Chen et al., 2022; James & Carrivick, 2016; Pang et al., 2023; Patel et al., 2022; Pieczonka et al., 2018). The analysis shows that while these parameters do affect the spatial distribution of the estimated thickness, the resulting variation in overall ice thickness remains limited, and does not significantly change the glacier-scale thickness estimate. Therefore, the uncertainties associated with these parameters have only a minor effect on the total ice-thickness uncertainty, although local variations—especially in steep terrain—can still occur. Glacier shape factor and surface slope depicts highly sensible inversion parameters. In the revised manuscript we quantified and clarified the parameters sensitivity as follows.

"Figure R3. Sensitivity analysis of the input physical parameters of the VWDV model (where 'dh', 'f', 'Ac' depicts ice thickness uncertainty, shape factor and Arrhenius creep factor)"

The sensitivity analysis for glacier ice thickness and associated uncertainty (dh) to various physical parameters highlights the relative influence of each factor of the VWDV model (Chen et al., 2022; James & Carrivick, 2016; Pang et al., 2023; Patel et al., 2022; Pieczonka et al., 2018). The shape factor (f), when varied from 0.70 to 0.90, shows a change in dh of approximately  $\pm 2.2\%$ , indicating that the model is sensitive to variations in f. A change of 0.1 in shape factor results in measurable changes in uncertainty of about  $\pm 6.5\%$ . For ice density ( $\rho$ ), when varied 200 kg/m³, the change in dh is about  $\pm 0.5\%$ , depicting that dh is less sensitive to changes in ice density, although density is still an important parameter in ice flow models. The slope ( $\alpha$ ) has a strong effect on glacier thickness uncertainty itself. As the slope change 1°, thickness change significantly  $\pm 3.5\%$ , reflecting a high sensitivity of ice thickness to slope. A 30 m resolution Cartosat DEM with a vertical error of  $\pm 8$  m results in a slope uncertainty of about 19% in flat terrain (worst case) and about 2.5% in steep slopes. An uncertainty of 20° in slope (worst case) results in ~18m uncertainty in ice thickness compared to mean glacier ice thickness of 147m, i.e. about  $\pm 12.2\%$ . The creep factor (Ac), when varied from 1.55 × 10-23 to 1.30 × 10-23 Pa-3 s-1, results in a change of approximately  $\pm 1.1\%$  in dh. This demonstrates a lesser

sensitivity, yet Ac influences the ice deformation rate through Glen's flow law. The slope and shape factor exert the most significant influence on the uncertainty of the glacier ice thickness, followed by creep factor, with ice density showing the least sensitivity for VWDV model."

58. Section 6.2.2: The long term mass loss of Gangotri Glaciers seems a robust result of this analysis. This seems to be in accordance with other studies. However, the comparison could be more focussed on the temporal evolution (still taking the errors in consideration), as most of the studies cover different periods. Also, the comparison of the DEM-differencing results from 2000-2015 and the thickness changes from 2016-2023 could be compared in a more detailed way. Again there raises the question, why the DEM from 2018 is not used to combine the two periods.

We thank the reviewer for their valuable observation regarding the comparison of temporal evolution across different studies and datasets. However, we submit that long-term mass loss evaluations must be investigated at a larger time interval considering that the variability annually is often not significantly large. As suggested, we analysed the mass-balance across different timeframes from various studies in the following chart.

Figure R4. The trend of the specific mass balance of the Gangotri glacier estimated by different study from 1985 to 2024.

We have already responded regarding the DEM selections, we sincerely request you to consider the same.

59. Fig. 8: considering the inherent errors of the method, this evolution of the ice volume rather likely cannot be used for any analysis of volume change. It is also necessary to show the errors in the graph.

We agree with the reviewer considering the uncertainties, the evolution of the ice volume may not be precise, however it maybe noticeable in the absence of any other approaches, yet. Similar approach has been followed widely in the literature (Bhattacharjee & Garg, 2024; Cook et al., 2023; Halder et al., 2025; Labe et al., 2018; Nela et al., 2020; Patel et al., 2022; S & M, 2025; J. Singh et al., 2023). Please consider the following updates for Figure 8.

"Figure 8: Piece-wise estimated glacier volume of the Gangotri glacier (2016 - 2023). The curve indicates a declining trend with high variations between 2016-2018 and some undulations thereafter."

60. L. 716-727: This is an interesting attempt to relate the volume changes to climatic variations. But I doubt that the results are robust enough. At least this needs to be discussed in the manuscript.

We thank the reviewer for this insightful remark. We agree that linking inter-annual ice volume fluctuations to large-scale climatic oscillations requires caution. Although these fluctuations show a reasonable correspondence with ENSO-driven climatic anomalies, we acknowledge that the robustness of this linkage remains constrained by the relatively short observation window and the indirect nature of volume–climate attribution. Glacier mass and volume changes are governed by a complex interplay of local processes—including monsoon strength, precipitation phase and distribution, surface albedo, debris cover, and turbulent heat fluxes—which cannot be fully isolated using reanalysis datasets alone. Therefore, the ENSO interpretation presented here should be regarded as a first-order indication of potential climatic influence rather than a definitive causal relationship. The following changes are made in the revised manuscript.

"Although the 2017 ice volume peak shows a clear response to favourable climatic conditions, this interpretation should be viewed with caution, as the annual fluctuations are relatively small compared to their uncertainty range. Therefore, the apparent climatic signal may be partly amplified by uncertainty in the volume estimates. While the model provides a robust first-order trend, the limited observation period and the absence of in-situ mass-balance data restrict our ability to attribute the 2017 peak solely to climate variability."

61. Fig. 9: The source of the data is only mentioned in the text, but not in the caption. Maximum temperature is not a valuable parameter, as it does not provide any information about the seasonal melt potential. How is the runoff calculated in this data set?

We thank the reviewer for this valuable comment. The caption of Fig. 9 has now been updated. In addition, Table 1 and Section 2.2 have been revised to clearly describe the climate datasets used in our analysis, including the parameters relevant for melt and mass-balance variability. The Runoff data was extracted from TerraClimate products, in the absence of field data or any other source of data.

---

## Author Comment (AC2)

**RC2: 'Comment on egusphere-2025-1614', Anonymous Referee #2, 01 Oct 2025**

**General comments:**

This study presents comprehensive work done on the Gangotri glacier in Himalaya.

The study has three to four branches:

- 1. Comparing different methods for velocity mapping
- 2. Calculating ice thickness using two different inversion approaches one based on a known DEM and velocity field and one on a stress-based approach
- 3. Calculating thinning rates based on dem differencing
- 4. Apparent thermal inertia: but this is not well described

While a lot of solid work clearly forms the basis of the manuscript, I found the 40-page manuscript quite hard to follow, and a major revision of the text as well as a discussion of the velocity maps is required. I agree with Referee #1 on all comments, and I hope my general and line-by-line comments below can help identify the main issues to address before it can be considered for publication.

We sincerely thank the reviewer for their valuable time and thoughtful comments on our manuscript "Understanding the Gangotri glacier dynamics: Implications from a fully distributed inversion of water equivalent volume change." The comments and suggestions provided have been significantly beneficial in improving the clarity, rigor, and overall quality of the work. We have carefully considered each comment and intend to revise the manuscript accordingly while addressing the concerns and enhancing the presentation of our findings. The **responses** and revisions are mentioned in different colors in the following.

**Main concerns:**

The title: Understanding the Gangotri glacier dynamics: Implications from a fully distributed inversion of water equivalent volume change.

I think that the title might refer to glacier changes and not actual dynamics, since it is the changes in mass/volume and not the glacier dynamical changes that are in focus of the paper.

We thank the reviewer for their insightful remarks. We have accordingly revised the title as follows:

"Understanding the Gangotri glacier: Implications from a remote sensing based fully distributed inversion of equivalent water-volume change"

There is a strong focus on the ice volume calculations and as far as I understand the velocity mapping is done with the purpose to create an input for the thickness and volume change inversions. Maybe with the focus of ending up with mass balance estimates, the velocity mapping descriptions could be shortened throughout or maybe moved to supplementary, making room for a stronger discussion of the quality of the velocity maps and the different time periods.

We thank the reviewer for their remarks. The glacier surface velocity is one of the critical inputs in the velocity based inversion of ice-thickness and the focussed model VWDV. Additionally, we compared different tools for velocity inversion from remote sensing data. Following this, we believe that sufficient depth to details on velocity estimation, and associate discussion is needed in the manuscript as conducted. We agree that in the original write-up the clarity was lacking. To account for this, we have

merged section 3.1. Image Processing and Ice Masking and section 3.2. Glacier velocity estimation as a single section 3.1. Glacier velocity estimation. We have also added further details to clarify the data selections and methods. We have also revised section 6.1. to improve the discussion on comparison of velocities as per glacier sub-zones, different methods and other studies. These two sections can now be read as follows.

**"3.1. Glacier ice velocity estimation**

A total of 152 Sentinel-2 RGB images, spanning the years 2016 to 2023, were used for Glacier Image Velocimetry (GIV)-based velocity estimation (Van Wyk De Vries & Wickert, 2021). The RGB images were processed in Google Earth Engine (GEE) with a cloud-masking filter to enhance feature-tracking accuracy. Additionally, for comparison single-band near infrared (NIR) grayscale images, free of clouds and snow were utilized for velocity estimation using the Co-registration of Optically Sensed Images and Correlation (COSI-Corr) tool (Leprince et al., 2007) and Image georectification and feature tracking toolbox (ImGRAFT) with time-series image pairs from 2016 to 2023. Sentinel-2 NIR single-band images for the snow-free months of July to October, were downloaded from the Sentinel Data Hub for velocity estimation using Cosi-Corr and ImGRAFT tools. The three methods used for glacier surface velocity estimation rely on tracking persistent surface features between multi-temporal satellite images using different correlation algorithms and windowing strategies, were employed to cross-validate results and assess the consistency and reliability of velocity estimations derived from different algorithms. The resulting velocity maps were carefully refined by applying a glacier mask based on the manually derived glacier outline based on high resolution imagery and RGI 6.0 glacier boundaries.

Sentinel-2 RGB images were used for a pair-wise velocity estimation based on the GIV tool using multi pass feature tracking frequency domain image correlator (Van Wyk De Vries & Wickert, 2021). Image pairs with temporal intervals ranging from 9 days to 1 year were used to derive annual surface displacement and velocity fields of the glacier. Total pairs were generated for this glacier with all years 151 for this study covering an 8-year time period (2016 - 2023). For each Sentinel-2 RGB image pair, displacements were estimated iteratively using a multipass template matching following the standard GIV workflow. The GIV approach follows an iterative multi-pass approach in which displacement is first estimated using larger window sizes, followed by smaller windows in subsequent passes to refine the estimated velocity. We used the standard GIV reference window sizes of 400 m, 200 m, and 100 m and a 50% window overlap, resulting in a final glacier surface velocity map with a spatial resolution of 30 m (Van Wyk De Vries & Wickert, 2021). The overlap refers to the percentage of each template window that overlaps with adjacent windows during feature matching in the Glacier Image Velocimetry (GIV) process, where a 50% overlap is used to significantly reduces noise in the resulting velocity field and ensure consistency. Signal to noise ratio lower than 5 and peak ratio less than 1.3 were considered during multi-pass template matching with sub-pixel estimator (a correlation refinement technique used to improve the accuracy of displacement measurements beyond the spatial resolution of the input imagery) (Van Wyk De Vries & Wickert, 2021). To enhance the quality of estimated displacements, three pre-processing filters were applied including, the Contrast Limited Adaptive Histogram Equalization (CLAHE), a high-pass filter, and the Near-Anisotropic Orientation (NAO) Filter (Van Wyk De Vries & Wickert, 2021). The CLAHE filter was applied to the satellite images to enhance local contrast in the images aiding in feature tracking. The high pass Sobel filter was applied to the images for improved identification of features for tracking, and the NAO filter was applied to the resulting displacement map. Further, to remove outliers arising from mismatches or noisy correlations in feature tracking, an upper velocity threshold of 200 m a-1 was applied to the resulting displacement map and thresholded pixels were interpolated. The threshold was selected based on prior studies in the region and is consistent with expected velocity ranges for Gangotri glacier (Bhushan et al., 2017; Gantayat et al., 2014). The resultant displacement map represents the annual velocity maps with velocity values expressed in meters per annum (ma-1).

Through the Cosi-Corr, a single-pass feature-tracking approach was implemented to evaluate glacier surface displacement using Sentinel-2 NIR band imagery, primarily acquired during July–October (2016 to 2023) to ensure minimal cloud cover over the Gangotri Glacier (Table S1). The surface

displacement was estimated for consecutive study intervals (2016–2017, 2017–2018, ..., 2022–2023)(Leprince et al., 2007). The COSI correlation module provides two correlation algorithms frequency-based and statistical approach. This study used the frequency method, which is better for reliable results using an optical dataset (Bhushan et al., 2017). The correlation window, commencing with an initial window dimension of  $64 \times 64$ , progressively decreases to a final size of  $32 \times 32$ , employing a step size of 2, as specified by Bhushan et al., 2017. First, the displacement output is derived at resolution at 160 meters, containing displacements in the east-west (EWD) and north-south (NSD) direction, as well as the associated signal-to-noise ratio (SNR). Pixels with low correlation are eliminated by applying a limit value of SNR

Figure 6: a) Illustrates the correlation between different glacier velocity estimation approaches for the Gangotri Glacier across various zones; b) Root Mean Square Error (EMSE) of the ice velocity between different approach and c) Depicts the distribution of glacier velocity along the central flow line, extending from the glacier terminus to the accumulation peak (A–A').

In the ablation zone, ITS\_LIVE shows a higher correlation with COSI Corr (0.872) and about similar correlation with GIV (0.738) and ImGRAFT (0.753), reflecting the influence of surface melting, crevassing, and ice deformation, which can introduce uncertainties in velocity retrieval (Fig. 6b). In contrast, the accumulation zone presents the strongest correlations, with ITS\_LIVE aligning closely with COSI Corr (0.977), GIV (0.922), and ImGRAFT (0.964), suggesting more stable ice dynamics in this region due to reduced surface melting and a more consistent ice mass flow (Fig. 6b).

Fig. 3e & 6c represents the glacier velocity distribution along the central flow line from the terminus to the accumulation zone, highlighting distinct spatial variations in ice motion. The velocity remains relatively low near the terminus due to high frictional resistance from bedrock and debris cover, gradually increasing in the ablation zone where ice thinning and gravitational flow enhance movement (Nicholson et al., 2018). In the accumulation zone near the ELA, the velocity reaches its peak, attributed to the increased ice mass and steep surface gradients (Vatsal et al., 2025). The observed variations of the entire glacier velocity reflect the combined effects of topography, ice thickness, and surface conditions, with the ELA emerging as a critical zone for maximum ice movement. The strong correlation suggests that all the remote sensing methods effectively capture glacier dynamics with consistent ice movement along the central flow line. These results demonstrate the compatibility of remote sensing based feature tracking techniques with well-established satellite datasets. As delineated in Fig. 6a, the ITS LIVE velocity captures these patterns of decreasing velocity from accumulation to the ablation zone, followed by ImGRAFT, which also captures similar pattern, however, with higher velocity in some accumulation zones in contrast to ITS\_LIVE. These differences in the velocity patterns are also statistically observed in the higher RMSE for ImGRAFT compared with ITS\_LIVE (Fig. 6b). It is worth mentioning here that for the statistical comparison, the ITS\_LIVE and the COSI-Corr velocities were resampled to 30m from 120m and 60m, respectively. The relatively higher agreement between the COSI-Corr velocity and the ITS LIVE product may be attributed to the coarser resolution compared with the GIV and ImGRAFT where the source spatial resolution of the velocity product was 30m. Fig. 6c shows that the GIV and ImGRAFT methods largely underestimate the velocity in some zones of the accumulation region along the central flow line.

While the velocity estimates are broadly consistent across all remote sensing methods, this comparison emphasizes spatial reliability more than validation. The methods appear most robust in the ablation zone, where clear surface features persist throughout the season. In contrast, estimates in the accumulation zone carry higher uncertainty, and results in this region should be interpreted conservatively. These findings highlight the importance of considering glacier zone-specific reliability when applying and comparing remote sensing velocity products.

**6.1.2. Assessment with other studies**

Due to the harsh climatic conditions in the Gangotri Glacier region, in-situ field-based velocity measurements through stake installations remain challenging. Therefore, we compared our velocity estimates with existing studies based on optical and microwave remote sensing techniques. To ensure meaningful comparison, we focused on specific glacier zones—particularly the snout and ablation areas—where earlier studies have discussed glacier surface velocity quantitatively. In this study, the surface velocity (2019) near the central flowline in the near the terminus zone was estimated to range between  $18 \pm 3.6$  and  $31 \pm 5.78$  m  $a^{-1}$ , which aligns well with the findings of (Thakur et al., 2023; Gantayat et al., 2014), who reported velocities between 20 and 30 m  $a^{-1}$  in period of 2019. Similarly, (Saraswat et al., 2013) documented velocities between  $24.8 \pm 2.3$  and  $28.9 \pm 2.3$  m  $a^{-1}$  in the snout region in the period of (2004-2010). A more recent study by Bhattacharjee & Garg, (2024), based on Sentinel-1 offset tracking for 2017–2022, estimated a mean glacier velocity of  $32.85 \pm 2.25$  m  $a^{-1}$ , closely matching the average values along the central flow line observed in our results. Our results identified the mean surface velocity along the central flow line as  $30.20 \pm 5.60$  m  $a^{-1}$  for 2017-2022.

In the study period, we observed a downslope acceleration in the Gangotri Glacier's surface velocity as per time-series investigations based on the GIV tool. However, the distribution of the surface velocity varied across the glacier. The marginal regions of the glacier exhibited a decreasing trend in surface velocity which may be due to debris accumulation, which increases friction and inhibits ice flow (Fig. S2). Additionally, the presence of stagnant ice and reduced ice thickness in marginal areas further

contributes to slower movement, as the driving stress diminishes near the margins, whereas the central trunk of the glacier displayed an increasing trend during the study period."

**Abstract:**

The abstract needs to be revised and some of the details left out. See line by line comments for details.

We thank the reviewer for noticing the issues in the abstract. We have revised the same accordingly as follows.

"Evidence of rapidly increasing temperatures with climate change is clearly visible in the concerning mass changes of Himalayan glaciers. Subsequently, monitoring glacier volumes is critical for managing regional water resources and predicting glacier dynamics. The Gangotri Glacier remains a subject of scientific debate due to limited field data on its dynamics, ice thickness, volume, and mass balance, leading to uncertainties in understanding its behavior and changes. The Gangotri glacier, a significant water resource for northern India, is experiencing significant changes due to climate change. The ice thickness and mass balance of Gangotri glacier during 2016-2023 was analyzed using a velocity and shear stress-based framework. A laminar flow-based approach is applied to determine the ice thickness of the Gangotri Glacier, while glacier surface velocity was estimated from Sentinel-2 multispectral imagery using three independent feature-tracking methods. The thickness change of the study period is used to estimate the mass balance and equivalent water volume change of the glacier. The analysis revealed a pronounced spatial variability in glacier flow and ice thickness, with faster movement and greater ice depth in the accumulation and upper ablation zone that gradually diminished toward the snout. The Gangotri Glacier exhibited a consistent thinning trend and negative mass balance, reflecting significant ice loss during the study period. We observed the volumetric change is a declining pattern of the study period 2017 to 2023 gradually. The climatic parameters observed an increasing trend over the last two decades. We also found that the Apparent Thermal Inertia (ATI) increased which determined the debris accumulation over the ablation zone significantly from the side wall and transported from the accumulation and emerges in the ablation zone of the glacier due to fluctuation of the temperature differences (Thaw-freezing). These changes denote a significant reduction in the water storage capacity of the Gangotri Glacier."

**Description of methods and data:**

The Apparent Thermal Inertia is mentioned in the abstract, but no information can be found about this in the introduction, method and data or results section.

We are grateful for this observation. We have included in the introduction literature focusing on the critical influences of climate forcing on Himalayan glaciers.

"Glacier dynamics are critically governed by interconnected climatic forcing that drive surface ablation, ice flow acceleration, and irreversible thinning (Immerzeel et al., 2020). Declining winter precipitation and snowfall reduce accumulation zones, lowering albedo and shifting melt-runoff timing, with snowmelt contributions dropping in High Mountain Asia (Jouberton et al., 2025; Kraaijenbrink et al., 2021). Enhanced runoff from intensified summer melt alters downstream hydrology, amplifying flood risks while depleting late-season baseflow (Pritchard, 2019). The thermal properties as identified from the ATI of snow, ice, and debris of a glacier play a vital role in determining its thermal resistance and its melting processes. ATI quantifies the resistance to temperature change in the upper layers of ice/snow, revealing enhanced ablation on debris-free tongues where diurnal temperature shifts are high and consequentially low ATI facilitates frequent melt–freeze cycles (Brenning et al., 2012). Thus, the combined effects of changing precipitation, runoff, ATI, and aerosol-induced albedo reduction collectively drive enhanced glacier surface lowering across the Himalaya (Bolch et al., 2019)."

A new sub-section 3.6. Analysis with key influencing parameters is added in the methodology section.

**"3.6. Analysis with key influencing parameters**

The Gangotri glacier ice thinning was analysed corresponding climatic factors such as precipitation, runoff, snowfall, ATI, PM2.5 concentrations, and aerosol optical depth. The various satellite and reanalysis products used for the extraction of these factors are listed in Table 1, and these were used at their native spatial resolution to derive mean annual magnitudes to assess their impact on Gangotri glacier thinning. One of the critical parameters used in this study, is the ATI, which is primarily based on the diurnal melt-freeze cycles is derived from MODIS LST (MOD11A1) and MODIS Albedo (MCD43A3). The diurnal melt-freeze cycles in temperate/polythermal glaciers exert a strong influence on the seasonal pattern of surface mass loss, as daily fluctuations in surface temperature directly control melting and refreezing intensity (Irvine-Fynn et al., 2011; Steiner et al., 2021). The ATI is a key parameter used to evaluate the thermal response and heat retention capacity of glacier surfaces, reflecting variations in surface composition, moisture content, and debris thickness (Foster et al., 2012; Van Doninck et al., 2011). ATI was calculated using Equation (18) following the formulation of Van Doninck et al., (2011), where it is derived from the combination of diurnal temperature amplitude and the albedo to represent the apparent resistance of the surface to temperature change. The ATI was analysed to infer thermal heterogeneity and surface energy exchange characteristics across the glacier, thereby linking surface thermodynamics with observed mass-balance variations.

$$ATI = C \frac{(1 - \alpha)}{\Lambda T} \tag{18}$$

where  $\alpha$  represents the surface albedo, and C = 0.84 is the solar correction factor, which accounts for variations in incoming solar radiation calculated according to (Nicholas & Locke, 1982) and  $\Delta T$  is the diurnal temperature range. This equation highlights the fundamental relationship between surface reflectivity, thermal variation, and energy retention, making ATI an effective indicator of glacier melt dynamics."

In particularly, the velocity data used as input needs to be better described to understand the effect of potentially different input data to the velocity mapping. And an analysis of the seasonal and year to year evolution of the surface velocity would be both interesting and aid the understanding of uncertainties in the thickness calculations.

We thank the reviewer for their observation. We have revised the sections to add more details and clarity to velocity inversion process and datasets and their discussion.

Although it would be interesting to compare the seasonal velocities and annual velocities, due to limited availability of cloud-free satellite images, seasonal velocity estimation is not possible with optical remote sensing data, particularly in very high elevation glaciers.

Using the Shallow Ice Approximation on a valley glacier needs to be justified, maybe with references to other studies? Also, I would be really interested in seeing the seasonal evolution of the velocity (even though that might be out of the scope for this study).

We thank the reviewer for their remarks. Reviewer-1 had also suggested improving clarity and justifications for the same. We have revised the section as follows.

**"3.2. Glacier Ice thickness estimation**

Ice thickness is determined using the glacier surface velocity and the ice surface slope through the shallow ice approximation (SIA) method (Cuffey & Paterson, 2010; Hutter & Morland, 1984; Le Meur et al., 2004). The laminar flow-based SIA method is commonly used globally for estimating glacier ice thickness. This approach combines the principles of glacier flow with the shallow-ice approximation to estimate ice thickness under the assumption of a hard, non-deforming bed (Cuffey & Paterson, 2010; Farinotti et al., 2019; Frey et al., 2014; Maussion et al., 2019; Millan et al., 2022; Nela et al., 2023). The SIA method utilizes basal shear stress for defining the glacier motion as compared to the full driving stresses, and assumes that the local stresses inducted are much greater than the stresses arising from

the lateral coupling between adjacent columns (Cuffey & Paterson, 2010). Many glaciers in the Himalaya and Andes are characterized by relatively thin ice in their upper reaches and moderate to steep surface slopes. These conditions exhibit the dominance of basal shear stress within the total driving stress, thereby justifying the application of the SIA for estimating ice thickness and glacier dynamics (Schotterer et al., 2003; Thouret et al., 2007). Several studies employed the glacier surface velocity based approach (Farinotti et al., 2017; Gantayat et al., 2014; Van Wyk De Vries & Wickert, 2021), while others utilize the basal shear stress approach for the retrieval of glacier ice thickness (Farinotti et al., 2017, 2019; Haeberli & Hoelzle, 1995; Kumari et al., 2021; Linsbauer et al., 2012). These methods were carefully selected to provide comprehensive estimates of glacier ice thickness by leveraging different glaciological principles.

**3.2.1. Velocity-based ice thickness**

Glaciers primarily move as the ice deforms under the force of gravity. This flow occurs through three main mechanisms: internal ice deformation, sliding at the base, and deformation of the subglacial bed (Hambrey & Glasser, 2012). The glacier ice surface velocity u is a result of two component glacier internal deformation  $u_d$  and basal velocity  $u_b$  (sum of basal sliding and subglacial sediment deformation) (Cuffey & Paterson, 2010), as shown in equation 2.

$$u(H) = u_d(H) + u_b \tag{2}$$

where, the internal deformation  $u_d$  and glacier ice surface velocity are functions of the ice-thickness H that are evaluated at the ice surface. The deformation velocity is related to the basal shear stress and the ice thickness as follows (Cuffey & Paterson, 2010).

$$u_d(H) = \frac{2A_c}{n+1} \tau_b^n H \tag{3}$$

where  $A_c$  is the Arehenius creep parameter,  $\tau_b$  represents basal shear stress, n is the Glen's flow exponent (n=3) (Glen, 1958). The glacier surface velocity as shown in equation 2 can then be represented as shown in equation 4 (Glen, 1958; Van Wyk De Vries et al., 2022).

$$(1 - \beta)u(H) = u_d(H) = \frac{2A_c}{n+1}\tau_b^n H$$
 (4)

where,  $\beta$  is the basal sliding correction factor (Van Wyk De Vries et al., 2022). The laminar flow law (King, 1983) accounts for both surface and basal velocities of the glacier. However, determining basal sliding velocity through remote sensing is not feasible.

Some benchmark studies assumed the subglacial deformation to be negligible and have considered basal sliding to represent the major contributions to the glacier's surface velocity (Farinotti et al., 2009, 2017; Linsbauer et al., 2012). In other studies, the basal velocity was typically considered as one-fourth (25%) of the surface velocity of the glacier (Bhushan et al., 2017; Gantayat et al., 2014; Nela et al., 2023; Remya et al., 2019). In contrast, van Wyk et al., (2022) employed a basal sliding correction factor to account for the fraction of glacier motion corresponding to basal sliding. The basal shear stress,  $\tau_b$ , is expressed in terms of measurable parameters following equation 5 (Frey et al., 2014; Linsbauer et al., 2012; Van Wyk De Vries et al., 2022).

$$\tau_b = f \rho_i g H \sin(\alpha) \tag{5}$$

where f is the shape factor (Gantayat et al., 2014; Haeberli & Hoelzle, 1995),  $\rho_i$  is the snow/firn/ice density, g is acceleration due to gravity,  $\alpha$  is the ice-surface slope angle (derived from the DEM) and H is ice-thickness. We follow the velocity-based ice thickness estimation by Van Wyk De Vries et al., (2022), as shown in equation 6.

$$H = \left(\frac{n+1}{2(f\rho_i g)^n A_c^* \exp\left(\frac{Q_c}{D} \left[\frac{1}{T} - \frac{1}{T^*}\right]\right)}\right)^{\frac{1}{n+1}} \left(\frac{u(H)(1-\beta)}{\sin(\alpha)^n}\right)^{\frac{1}{n+1}}$$
(6)

where the Arrhenius creep constant  $A_c$  as defined in Cuffey & Paterson, (2010), was determined using glacier ice surface temperature based on the following temperature-dependent relation.

$$A_c = A_c^* \exp\left(\frac{Q_c}{R} \left[ \frac{1}{T} - \frac{1}{T^*} \right] \right) \tag{7}$$

where the constants being  $A_c^* = 2.4 \cdot 10^{-24}$ ,  $Q_c = 115$  kJ mol-1,  $R \approx 0.0083145$  (the ideal gas constant), and  $T^* = 273$  K (Cuffey & Paterson, 2010). To estimate the ice surface temperature T of the Gangotri Glacier, MODIS MOD11A1 LST data was utilized. Further details regarding the temperature extraction and processing are provided in Table S2 and S3, and Fig S1.

The glacier ice thickness inversion as described in equation 6, involves several parameters typically assumed as constant such as g,  $\beta$ ,  $\rho_i$ , f,  $A_c$  and n. In equation 6, the spatially varying components include  $\alpha$ , u(H), and  $A_c$ . The parameter  $A_c$  varies spatially corresponding to the ice surface temperature determined by MODIS LST data,  $\alpha$  as per the DEM and the surface velocity u(H) as defined in the previous sub-section. Recent studies (Sinha et al., 2024; Van Wyk De Vries et al., 2022; Yang et al., 2024) as compared to previous literature (Bhattacharjee & Garg, 2024; Frey et al., 2014; Nela et al., 2023; Ramsankaran et al., 2018) vary the constant parameters based on a Monte Carlo simulation for improving the ice-thickness estimates. Following these recent studies, to account for the spatial and thermal variability of the Gangotri Glacier, which exhibits polythermal and temperate characteristics, the basal sliding correction factor  $\beta$  was varied from 0 to 0.4 in the inversion simulations as per previous studies in a Monte Carlo simulation framework. This approach provides a more realistic representation of basal motion compared to the conventional assumption of a constant sliding coefficient, the shape factor f was varied between 0.7 (ablation) and 0.9 (accumulation) instead of being held constant, to account for the influence of valley geometry and lateral drag effects across different sections of the glacier (ablation and accumulation) (Gantayat et al., 2014; Linsbauer et al., 2012; Sinha et al., 2024; Van Wyk De Vries et al., 2022). This range reflects the realistic variability between narrow, deep valley segments and wider glacier, thereby improving the physical representation of ice flow in the inversion modelling. The snow-firn-ice density was varied between 850 and 917 kg m-3 to account for glacier surface (Sinha et al., 2024; Van Wyk De Vries et al., 2022; Yang et al., 2024)."

Regarding the seasonal velocities, we agree it would be very interesting, yet it is not possible with optical remote sensing due to insufficient cloud-free images.

Mass balance estimates from thickness change, require some consideration of density which is not described to a high enough detail.

We thank the reviewer for pointing out this issue. We have revised sentences in the section 3.5. Mas balance estimation. The revisions are as follows.

"The mass balance and the change in the total volume  $\Delta V$  was determined by summing the change in the ice thickness  $\Delta h_i$  at an individual pixel using the equation 13.

$$B = \frac{\Delta V}{S} \times \frac{\rho_{ice}}{\rho_{\text{water}}} = \Delta \overline{H} \times \frac{\rho_{ice}}{\rho_{\text{water}}};$$

$$\Delta V = \sum_{i=1}^{N} r^2 \Delta \overline{h}_i; S = \sum_{i=1}^{N} r^2$$
(13)

where B is given in m.w.e,  $\Delta h_i$  is the mean ice thickness change of the ith pixel,  $\Delta H$  is the mean glacier ice thickness change in meters,  $\rho_{ice}$  is the density of glacier ice,  $\rho_{water}$  is the density of water, r is the

pixel resolution in meters,  $S = \sum_{j=1}^{n} r^2$  is the average area of glacier during the study period, N is the number of pixels covering the glacier ice at its maximum extent. The density of ice was considered as 850 kgm-3 following previous studies (Cogley, 2011; Hubbard et al., 2000; Zemp et al., 2013). A standard water density of 997 kg m-3 was used to convert the derived ice volume changes into m w.e., following commonly adopted glaciological practices (Huss, 2013; Zemp et al., 2013)."

**Line by line comments**

**Abstract:**

I would move the first sentence two sentences down to keep the structure natural: First general trends then Gangotri specifically

Thank you for your suggestion. We have moved and revised the sentence as follows.

"The Gangotri Glacier remains a subject of scientific debate due to limited field data on its dynamics, ice thickness, volume, and mass balance, leading to uncertainties in understanding its behavior and changes."

Line 21 - 25: Restructure for better reading flow.

Thank you for the suggestion. We have revised the sentences as follows for improving readability.

"The ice thickness and mass balance of Gangotri glacier during 2016–2023 was analyzed using a velocity and shear stress-based framework. A laminar flow-based approach is applied to determine the ice thickness of the Gangotri Glacier, while glacier surface velocity was estimated from Sentinel-2 multispectral imagery using three independent feature-tracking methods."

27 - 29: Leave the specific numbers on thinning out of the abstract and mention only the interpretation of these numbers and the final mass wastage

Thank you for the comment. We have revised the sentences as follows.

"The analysis revealed a pronounced spatial variability in glacier flow and ice thickness, with faster movement and greater ice depth in the accumulation and upper ablation zone that gradually diminished toward the snout. The Gangotri Glacier exhibited a consistent thinning trend and negative mass balance, reflecting significant ice loss during the study period."

31: leave out the sentence with the volume number

We thank the reviewer for their remarks. We have now removed the sentence as suggested.

**Introduction:**

43: What is HKH?

Thank you for your observation. The abbreviation HKH refers to the Hindu Kush Himalaya, which has now been added. The sentences now read as follows.

"The Hindu Kush Himalaya (HKH), also referred to as the "Water Tower of Asia," are home to one of the largest mountain glacier networks on Earth (Bolch et al., 2012). The HKH region hosts a total of 54,252 glaciers covering approximately 60,054 km² and holding an estimated ice reserve of 6,127 km³ (Bajracharya et al., 2015)."

45: What is meant by largely significant?

We thank the reviewer for the observation. The phrase has now been omitted for clarity, and a new sentence has been added as follows.

"The HKH region is the source of 10 major river basins — the Indus, Ganges, Brahmaputra, Irrawaddy, Salween, Mekong, Yangtze, Yellow, Amu Darya, and Tarim — making it one of the most important water resource of the world. Among these, the Indus, Ganga, and Brahmaputra basins together contain the majority of the region's glacierized area (Mukherji et al., 2015; Singh et al., 2025)."

52: What is meant by when? Is it the glaciers that are not covered in debris so: Particularly those glaciers that are not covered...

We have revised the sentence as follows.

"Glaciers that are entirely situated below 5700 m elevation are mostly sensitive to climate change, particularly those glaciers that are free from thick debris cover and are directly exposed to atmospheric conditions (Bajracharya et al., 2015)."

63 – 64: does ice velocity provide any insights into retreat or advance of glaciers in general?

We thank the reviewer for pointing this out. We have revised the sentence as follows.

"It provides insights into glacier movement patterns and indirectly glacier mass change (Herman et al., 2011)."

In general, the ice velocity provides valuable insights into glacier dynamics. Increased velocities may indicate mass gain and possible glacier advance, whereas decreasing velocity may reflect mass loss and glacier retreat (Herman et al., 2011). However, understandably, glacier behaviour is influenced by a complex interplay of climatic, geological, topographical and environmental factors (Dobreva et al., 2017; Nagai et al., 2014).

64: I don't think it is correct to say that velocity provides any predicting factor for GLOFs. As far as I understand the reference, it is surging that poses an increased risk of GLOFs.

We thank you for your insightful observation. We agree, Glacier velocity itself does not directly predict GLOF occurrence. Instead, surging glaciers, which exhibit episodic increases in velocity, can destabilize ice-dammed or moraine-dammed lakes and thereby increase the risk of GLOFs. We have revised the sentence as follows.

"In the case of surging glaciers (velocity spiking glacier), glacier velocity can highlight periods of increased instability which connected with a dammed lake may potentially increase GLOF risk. (Bhambri et al., 2020; Scherler et al., 2011)."

88-89: "It has been generally observed..." By whom?

We thank the reviewer for their comment. We were referring to Sinha et al. 2024. However, we have revised this paragraph (for lines 85-117 in original submission) following also the comments by Reviewer-1 as follows.

"The collection of in situ observations using intrusive or extrusive methods, such as hot water drilling, seismic or radar measurements, and gravimetry, poses significant challenges on glaciers with rugged terrain and are often impractical for complete glacier surfaces (Murray et al., 2007). Some glaciers are often inaccessible or have restricted access due to their geopolitical sensitivity rendering no potential opportunities for on-site data collection (Ambinakudige, 2010). As a result, remote sensingbased techniques are widely employed. Models using digital elevation model (DEM) data, glacier boundary, and boundary of ice-flow catchments—such as the mass conservation approach by (Farinotti et al., 2009; Huss & Farinotti, 2012)—have become particularly prominent for estimation of glacier ice thickness. Other methods are based on surface slope, velocity and basal shear stress (Bhushan et al., 2017; Gantayat et al., 2014, 2017; Linsbauer et al., 2012; Michel et al., 2013; Zorzut et al., 2020). An extension of the ice-velocity based approach based on SIA (VWDV) was proposed by (Van Wyk De Vries et al., 2022), which showed significant glacier ice-thickness estimates compared with in-situ measurements. Sinha et al., (2024) proposed an ensemble modelling approach to account for overestimation and underestimation of different ice-thickness inversion models. In practical applications, these modelling techniques effectively determine ice thickness for most glaciers, though they tend to exhibit greater uncertainties when applied to small glaciers unlike the Gangotri glacier with gentle topography (Linsbauer et al., 2012; Rabatel et al., 2018)."

107: What is meant by "less"?

We thank the reviewer; we have removed the sentence following the comments by Reviewer-1.

121: With theoretical knowledge – do you perhaps actually mean observations?

We thank you for your for pointing out this mistake. The purpose of the sentence was to highlight the intercomparison of velocity and shear stress based ice-thickness estimates. However, in the absence of field data it is not exactly possible. Hence, we have removed this phrase. The sentences now read as follows.

"This study aims to conduct a comprehensive investigation on understanding the ice-thickness and mass balance changes of the Gangotri glacier through a comparative integration of model-based analysis with remote sensing data. **The ice-thickness is modelled using both velocity and shear stress-based approaches.** We have also investigated the annual ice-thinning rates compared with the geodetic method and their implications in retrieving the equivalent water volume variations."

128-137: It is a good idea to have an overview of the study aims, they are just not easy to understand

We thank you for your suggestion. We have revised the study aims to improve clarity and readability as follows.

"In summary, the current study aims to perform:

- 1) an intercomparison of different glacier surface velocity determination methods using timeseries Sentinel-2 imagery;
- 2) a comparison of the ice-velocity and basal shear stress-based ice-thickness estimates;
- 3) estimation of the glacier ice thinning rate based on the annual ice thickness changes;
- 4) estimation of the mass balance and the equivalent water volume changes of the Gangotri glacier using annual ice thinning data for the study period 2016-2023;
- 5) an assessment of the influence of different climatic parameters such as precipitation, runoff, snowfall, etc. on glacier thinning and surface mass balance."

**Study area and data**

158: You should cite Sentinel data properly.

Cited in the revised text as follows

"The datasets used in this study primarily include remote sensing satellite optical imagery, particularly Sentinel-2 (European Space Agency, 2018) RGB and Near-Infrared band imagery (2016 – 2023) with a spatial resolution of 10 meters, which was essential for estimating glacier velocity and capturing temporal trends in glacier motion."

176: Describe why are you using ITS LIVE data at 120m resolution, when you have your own velocity maps from this study and what is velocity model output?

We thank you for the comment. We used the ITS LIVE data for independent comparison and considering that it is derived from a combination of optical and SAR remote sensing, where the latter is unaffected by cloud cover. We have revised as follows.

"ITS LIVE velocity data (Gardner et al., 2022), with a resolution of 120 meters, was employed to enable a comparative assessment with the ice velocity datasets derived in this study. This comparison provides an independent reference to evaluate the consistency of the estimated velocities, helping to identify potential deviations, while the widely used ice thickness dataset by Farinotti et al., (2019), with a 50-meter resolution, was used for comparative analysis with thickness model outputs. The ITS LIVE velocity data is based on an integrated combination of optical and SAR remote sensing data, where the latter provides gap-free all weather capabilities for velocity estimation, the drawback of the integration, however, is the reduced spatial resolution of 120 m."

204-205: This subsection is not about the ice velocity mapping

We thank the reviewer for pointing this out. We have merged section 3.1 and 3.2 as single section 3.1. glacier velocity estimation (provided in response to your comments under **Main Concerns**). Approximately, the entire section has been revised.

242: I don't understand what is meant by comprehensive. How many IV maps do you end up with? Only 1 per year? And have you considered variability in flow from year to year?

We thank the reviewer for pointing out these issues. We have revised the section as mentioned earlier. For the 8 years (2016-23), we generate annual velocity maps. The objective of the ice-velocity estimation was to gather the needed inputs for velocity based inversion of ice thickness hence, we have not considered the variability in glacier surface velocity for discussion, and focus on the mean velocity for further analysis. However, our discussion now focuses on the spatial variability of the surface velocity in different glacier zones (provided in response to your comments under **Main Concerns**).

Section 3.2: It is unclear to me what the purpose of applying these three different velocity mapping methods is.

We thank the reviewer for their comment. The three methods are used for intercomparison, where all three use different feature tracking approaches. Further details of the methods are given in the revised section 3.2. glacier velocity estimation (provided in response to your comments under **Main Concerns**).

234: Here the term sub-pixel estimator is used but not explained, only with a reference to the same study that we know the whole approach is based on. Please explain what is meant, this is also the case in line 236 and 240.

We thank the reviewer for notifying this. The sub-pixel estimator refers to correlation at sub-pixel level leading to improved displacement accuracy and provides robust sub-pixel correlation and error minimization procedures for glacier motion detection by optical imageries. We have revised the sentence as follows to improve clarity (Van Wyk De Vries & Wickert, 2021).

"Signal to noise ratio lower than 5 and peak ratio less than 1.3 were considered during multi-pass template matching with sub-pixel estimator (a correlation refinement technique used to improve the accuracy of displacement measurements beyond the spatial resolution of the input imagery) (Van Wyk De Vries & Wickert, 2021)."

244: It is difficult to understand what this paragraph is going to be about. Please add a sentence to explain what is coming.

We thank the reviewer for pointing this out. We have revised the entire section for improved readability and clarity (provided in response to your comments under **Main Concerns**).

Paragraph 251 to 264: You state just above that algorithms are explained in the reference, and then you add this whole paragraph explaining part of it. It is unclear what the purpose if this is?

We thank the reviewer for pointing this out. We have revised the entire section for improved readability and clarity (provided in response to your comments under **Main Concerns**).

3.3 Glacier ice thickness estimation

Please add a paragraph and references to why the SIA approach is suitable for this.

We thank the reviewer for pointing this out. We have revised the entire section for improved readability and clarity (provided in response to your comments under **Description of Methods and Data**).

300: What is meant by corresponds?

We thank the reviewer for pointing this out. It should have been "refers to". We have revised the entire section for improved readability and clarity (provided in response to your comments under **Description of Methods and Data**).

228-2029: I don't understand what is meant by this sentence. Is it a reference to the velocity maps described earlier?

We thank the reviewer for pointing this out. Feature tracking algorithms use multi-date satellite images for determining surface displacement (referred to as velocity). As per the GIV algorithm a minimum of 9 day gap and a maximum of 1 year (365 days) comprise the boundary conditions for the algorithm to estimate displacements. Cosi-Corr and ImGRAFT use only bi-temporal images, and hence for them we use images at a year difference. To avoid snow, typically, images are only considered between July-October, where often fewer images are available. Hence, the comparison with ITS LIVE also becomes critical, which is considered as a robust velocity product in the scientific community.

We have revised the entire section for improved readability and clarity (provided in response to your comments under **Main Concerns**).

334-336: I don't understand what is meant here at all.

We thank the reviewer for pointing this out. We have revised the entire section for improved readability and clarity (provided in response to your comments under **Description of Methods and Data**).

357: Use proper data citation

Cited as (OpenTopography, 2013).

383: cite Landsat correctly (Should probably be mentioned in the data section)

We thank the reviewer for pointing this out. The datasets section is revised as follows.

**"2.2. Datasets**

The datasets used in this study primarily include remote sensing satellite optical imagery, particularly Sentinel-2 (European Space Agency, 2018) RGB and Near-Infrared band imagery (2016 – 2023) with a spatial resolution of 10 meters, which was essential for estimating glacier velocity and capturing temporal trends in glacier motion. To derive the glacier surface slope, we used the Cartosat digital elevation model (DEM) v3 from 2018, with a 30-meter spatial resolution. This DEM product provides relatively better elevation accuracy, making it suitable for glacier slope and ice thickness modelling during the 2016–2023 study period (Talchabhadel et al., 2021). The SRTM DEM used in this study

(data year 2000), with a spatial resolution of 30 meters, was derived from C-band radar interferometry data and has an accuracy of ±16 meters (Farr et al., 2007). The vertical accuracy of SRTM DEM varies significantly depending on terrain. Rodríguez et al., (2006) reported a general vertical error of less than 9 meters for the SRTM DEM in mountainous regions. Kolecka & Kozak, (2014) observed a mean vertical error of 4.31 meters, and root mean square error (RMSE) of ±14.09 meters in mountainous terrain. The Copernicus DEM which is based on the TanDEM-X mission data, with a spatial resolution of 30 meters and an absolute vertical accuracy better than 4 meters (90% linear error) was used for year 2015 (European Space Agency & Airbus, 2022). The Copernicus DEM was used in combination with the SRTM DEM (data year 2000) to assess surface elevation changes between 2000-2015 over the Gangotri Glacier after penetration depth correction based on Landsat-7 satellite images with normalized difference snow index (NDSI) (Millan et al., 2015; Purinton & Bookhagen, 2018; Carturan et al., 2020; Guillet & Bolch, 2023). The selection of the Copernicus DEM and the SRTM DEM stems from the fact that both these are technically DSMs generated from radar remote sensing data, as compared to the Cartosat DEM which is generated from stereo-optical imagery.

The Gangotri glacier boundary was derived from the RGI 6.0 dataset (RGI Consortium, 2017) and subsequently refined using high-resolution satellite imagery in Google Earth Pro. On-screen digitization during snow-free periods allowed for accurate updates of the terminus and lateral boundaries, especially where RGI 6.0 outlines were inconsistent compared with recent imagery. ITS LIVE velocity data (Gardner et al., 2022), with a resolution of 120 meters, was employed to enable a comparative assessment with the ice velocity datasets derived in this study. This comparison provides an independent reference to evaluate the consistency of the estimated velocities, helping to identify potential deviations, while the widely used ice thickness dataset by Farinotti et al., (2019), with a 50-meter resolution, was used for comparative analysis with thickness model outputs. The ITS LIVE velocity data is based on an integrated combination of optical and SAR remote sensing data, where the latter provides gap-free all weather capabilities for velocity estimation, the drawback of the integration, however, is the reduced spatial resolution of 120 m.

To assess the climatic and environmental variability influencing the Gangotri Glacier, multiple datasets were utilized. MODIS (Yu et al., 2022) Land Surface Temperature (LST) data (MOD11A1) from 2016 to 2023, at a 500-meter resolution from Aqua and Terra satellites, was applied to derive ice surface temperature (IST). The MODIS MCD43A3 dataset provided surface albedo estimates essential for analysing the apparent thermal inertia (ATI) of the glacier ice surface. TerraClimate data was employed to examine maximum temperature, precipitation, and runoff trends across the glacier basin. ERA5-Land reanalysis data supplied estimates of snowfall parameter. The MODIS MCD19A2 product was used for aerosol optical depth (AOD) analysis, contributing to the assessment of radiative forcing impacts. Further, the Global Annual Weighted PM2.5 dataset (GWRPM25) was used to evaluate long-term atmospheric pollution trends over the region and their potential effects on glacier melt dynamics. The MODIS products have been widely used in the literature for climatic investigations in mountainous regions (Hassan et al., 2023; Thanveer et al., 2024; Varade et al., 2023; Varade & Dikshit, 2020). Table 1. summarizes the various datasets used in this study for modelling, comparative assessment and complementary analysis.

Table 1. Satellite and other ancillary data used in this study.

| Data                | Time period | Spatial
Resolution | Purpose                        | Source                           |
|---------------------|-------------|-----------------------|--------------------------------|----------------------------------|
| Sentinel – 2        | 2016 - 2023 | 10 m                  | Glacier velocity estimation    | (European Space
Agency, 2018) |
| Cartosat – 1
DEM | 2018        | 30 m                  | Slope and thickness estimation | (Muralikrishnan et al., 2013)    |
| SRTM DEM            | 2000        | 30 m                  | DEM
differencing            | (OpenTopography
2013)         |

| Copernicus
DEM                                     | 2015          | 30 m    | DEM
differencing                                                                                      | (European Space
Agency & Airbus,
2022)                           |
|-------------------------------------------------------|---------------|---------|----------------------------------------------------------------------------------------------------------|------------------------------------------------------------------------|
| RGI 6.0                                               | 2017          |         | Glacier ice masking                                                                                      | (RGI Consortium, 2017)                                                 |
| MODIS LST
(MOD11A1)
(Aqua and
Terra dataset) | 2016 - 2023   | 500 m   | Derived ice surface temperature                                                                          | (Yu et al., 2022)                                                      |
| ITS_LIVE                                              | 2016 - 2023   | 120 m   | Correlation with
velocity model
output                                                             | (Gardner et al.,
2022)                                              |
| Global Ice
Thickness
Dataset                    | 2019          | 50 m    | Correlation with
thickness model
output                                                            | (Farinotti et al., 2019)                                               |
| Google Earth Pro                                      |               |         | Glacier outline modification                                                                             | -                                                                      |
| Landsat – 7
and 8                                  | 2000 and 2022 | 30 m    | NDSI for snow
cover delineation
and IST
determination for
validation of MODIS
derived LST | (Earth Resources
Observation and
Science (EROS)
Center, 2013) |
| MODIS
(MCD43A3)                                    | 2000 - 2023   | 500 m   | Ice Surface
Albedo                                                                                    | (Schaaf & Wang,
2021)                                               |
| TerraClimate                                          | 2000 - 2023   | 4.56 km | Max. temperature,
Precipitation,
Runoff                                                            | (Abatzoglou et al., 2018)                                              |
| ERA 5 Land                                            | 2000 - 2023   | 9 km    | Snowfall estimation                                                                                      | (C3S, 2018)                                                            |
| MCD19A2                                               | 2000 - 2024   | 500 m   | Aerosol Optical
Depth (AOD)
estimation                                                             | (Lyapustin & Wang, 2018)                                               |
| GWRPM25                                               | 1998 - 2021   | 1.13 km | PM 2.5 trend analysis                                                                                    | (Van Donkelaar et al., 2021)                                           |

388: What is meant by each period?

Thank you for your observation. The phrase "each period" referred to the different years between 2016 and 2023. However, to avoid confusion, we have omitted this phrase from the revised manuscript for clarity.

"This study employed two approaches to estimate the glacier ice volume. The first method calculates the volume by multiplying the glacier area by its estimated ice thickness for 2016 - 2023, as expressed in Equation 11.

$$V_i = H.A_g \tag{11}$$

where  $V_i$  is the glacier ice volume, H is Monte-Carlo-derived mean ice thickness, as described by van Wyk de Vries et al. (2022), and  $A_g$  denotes area of the glacier."

472: I think some considerations on what seasons the velocity data covers might be appropriate here. Maybe the difference is not due to different methods, but simply different timing or inability to get velocities over certain time periods.

Thank you for your observation. As already mentioned that in one of the previous responses, the duration for acquiring images for velocity estimation is from July-Oct. GIV requires multiple images (minimum 4) for velocity retrieval while Cosi-Corr and ImGRAFT require minimum 2 images. From the image dataset for GIV, we supplied the images at a year gap to the other two methods.

523: I don't think we need abbreviations for left/right hand side.

Removed as suggested.

541: I don't think I have read anywhere anything about density considerations.

We thank the reviewer for pointing this out. We have now illustrated the computation of mass balance in the Methodology section (details provided in response to similar comment in **Study Area and Data**)

694-695: Volume estimates from other studies need to be discussed in the relation to this study's estimate – here it is only area mentioned?

Thank you for your observation we have revised the text and indicated the glacier ice volume with area during the study period 2016 - 2023.

"The Gangotri glacier ice volume was estimated by Gantayat et al. (2014) using velocity-based determined thickness of  $23.4 \pm 4.2$  km³ in 2009 and 2010. Haq et al. (2014) estimated the ice volume using artificial neural networks (ANN) and perfect plasticity to be 21.559 km³. Fig. 8 illustrates the time-series volumetric changes in Gangotri ice volume (pixel area × depth) and the corresponding equivalent water volume of the Gangotri Glacier from 2016 to 2023. In the present study, the Gangotri Glacier area, delineated from the updated glacier boundaries, was determined to be 141.9 km², with a mean ice volume of 19.70  $\pm$  2.64 km³ for the period 2016–2023. This estimate shows good agreement with the values reported in earlier studies, reflecting consistency in the glacier's overall volumetric characteristics despite methodological differences and temporal variations."

Table 4: It is quite important to state what period the volume estimate is based on in this table. Then it becomes clear that volume has decreased since the first observation maybe?

Thank you for your valuable suggestion. We have revised the table captions as follows.

"Table 4. Comparison of the Gangotri Glacier ice volume and estimated mean thickness as reported by Frey et al., (2014) for the period 2000–2010, and as obtained in the present study 2016–2023."

724-727: you have the observations to explain this if you want. At least you could say something about which zone the thinning was more severe.

Thank you for your insightful observation. In section 5.3. we have described the zonal characteristics in terms of thinning and equivalent water volume change. Here, we have added a sentence as follows.

"As described in section 5.3., the particularly dominant thinning in the ablation zone may also be assessed as increased equivalent water loss in this zone. In contrast, in the accumulation zones particularly in the tributary glaciers, there is an increase in the equivalent water volume."

729-731. so, what you are basically saying is that the mass wastage rate has been the same since 1985 to present?

Thank you for your insightful comment.

While the comparison shows similar magnitudes of mass wastage from 1985 to the present, we do not imply that the rate has remained constant over time. The consistency in the average values primarily reflects the long-term negative mass balance trend of the Gangotri Glacier under persistent climatic forcing. However, short-term variations in melt intensity, accumulation, and climatic conditions likely

exist between different study periods. We have revised the manuscript to clarify that the apparent similarity indicates a sustained pattern of glacier mass loss rather than a constant rate through time.

The following lines have been added and revised to the end of this section

"These comparisons collectively indicate a persistent negative mass balance trend in the region over the past several decades, although short-term variability in melt intensity and climatic forcing likely occurred between different study periods. The overall consistency in long-term averages highlights sustained glacier mass loss rather than a uniform rate through time. Details of these comparisons are provided in Section 2 of the supplementary material (Table S4)."

**Data availability:**

It would be of great interest to the community if also the datasets produced in this study were published.

We thank the reviewer for their remarks. In the next phase of the peer review process, we will provide the georeferenced raster maps through open source repositories like Zenodo.